# Exploring the ability of the variable-resolution CESM to simulate cryospheric-hydrological variables in High Mountain Asia

René R. Wijngaard[1,*], Adam R. Herrington[2], William H. Lipscomb[2], Gunter R. Leguy[2], and Soon-Il An[1,3]

[1]Irreversible Climate Change Research Center, Yonsei University, Seoul, South Korea
[2]Climate and Global Dynamics Laboratory, National Center for Atmospheric Research, Boulder CO, USA
[3]Climate Theory Lab, Department of Atmospheric Sciences, Yonsei University, Seoul, South Korea
[*]Now at: Institute for Marine and Atmospheric Research Utrecht, Utrecht University, Utrecht, the Netherlands

*Correspondence to*: Soon-Il An (sian@yonsei.ac.kr) and René R. Wijngaard (r.r.wijngaard.uu@gmail.com)

**Abstract.** Earth System Models (ESMs) can help to improve the understanding of climate-induced cryospheric-hydrological impacts in complex mountain regions, such as High Mountain Asia (HMA). Coarse ESM grids, however, have difficulties in representing cryospheric-hydrological processes that vary over short distances in complex mountainous environments. Variable-resolution (VR) ESMs can help through targeted grid refinement to overcome these limitations. This study investigates the ability of the VR-Community Earth System Model (VR-CESM) to simulate cryospheric-hydrological variables such as glacier surface mass balance (SMB) over HMA. To this end, a new VR grid is generated with regional grid refinement up to 7 km over HMA. Two coupled atmosphere-land simulations are run for the period 1979–1998. The second simulation is performed with an updated glacier-cover dataset and includes snow and glacier model modifications. Comparisons are made to gridded outputs derived from a globally uniform 1° CESM grid, observation-, reanalysis-, and satellite-based datasets, and a glacier model forced by a regional climate model (RCM). Climatological biases are generally reduced compared to the coarse-resolution CESM grid, but glacier SMB is too negative relative to observation-based glaciological and geodetic mass balances as well as RCM-forced glacier model output. In the second simulation, the SMB is improved but is still underestimated due to cloud-cover and temperature biases, missing model physics, and incomplete land-atmosphere coupling. The outcomes suggest that VR-CESM could be a useful tool to simulate cryospheric-hydrological variables and to study climate change in mountainous environments, but further developments are needed to better simulate the SMB of mountain glaciers.

## 1 Introduction

High Mountain Asia (HMA) encompasses the South and Central Asian Mountain ranges (e.g., the Himalayas, Karakoram, Pamir, and Tien Shan) and the highlands of the Tibetan Plateau. The region, also known as the "Asian Water Tower", hosts the largest reserves of snow and glacier ice outside the polar regions and provides essential water resources for millions of

people living in the region and surrounding lowlands (Yao et al., 2012; Immerzeel et al., 2020). By modulating the release of meltwater into rivers, the glaciers and snow reserves can sustain seasonal water availability and are therefore important contributors to the water supply for irrigation, drinking water, hydropower, industry, and ecosystem services (Nie et al., 2021; Lutz et al., 2022).

In recent decades, ongoing climate change has reduced snow volumes and driven the widespread retreat of glaciers worldwide (Hock et al., 2019). Similar trends have been observed in HMA over the last few decades, although the response to climate change is regionally dependent. Differences are linked, in part, to spatial and seasonal contrasts in precipitation patterns that are associated with the complex interplay between topography, the westerlies, and the Indian and East Asian monsoon systems (Yao et al., 2012). Westerlies drive precipitation during winter/spring in the western part of HMA,

whereas the Indian and East Asian monsoon systems are dominant to the east and south and deliver large amounts of precipitation during the monsoon season (June-September) (Wijngaard et al., 2017; Lutz et al., 2019). In response to climate change, rising temperatures and a weakened summer monsoon have led to negative snowfall trends and reduced snow water equivalent (SWE) in most regions under the influence of the monsoon systems (Yao et al., 2012; Smith and Bookhagen, 2018). Also, many HMA glaciers have lost mass since the end of the Little Ice Age (~1300–1850) with accelerated mass

losses over the last few decades (Brun et al., 2017; Zemp et al., 2019; Shean et al., 2020). This is in contrast to the western part of HMA (i.e., Pamir, Kunlun Shan, Tien Shan, Karakoram) where westerlies have strengthened over the last few decades (Cannon et al., 2015), possibly leading to increased winter snowfall and SWE (Cannon et al., 2015; Smith and Bookhagen, 2018). Furthermore, glaciers in a few regions (e.g., Karakoram, western Kunlun Shan, and eastern Pamir) have stayed in balance or gained mass in recent decades (Brun et al., 2017).

In the future, it is projected that glaciers in HMA will continue to lose mass and that snow cover and volume will decline further (Viste and Sorteberg, 2015; Kraaijenbrink et al., 2017). These changes will impact seasonal water availability, negatively affecting downstream populations and infrastructure (Nie et al., 2021; Li et al., 2022). Furthermore, the risk for natural hazards, such as glacial lake outburst floods (GLOFs) and riverine floods, might increase. Downstream climate in East and South Asia might be affected, and global mean sea level will rise (Frey et al., 2010; Wijngaard et al., 2017;

Marzeion et al., 2020; Li et al., 2022). For this reason, we need to better understand the potential impacts of glacier retreat and changing snow regimes, which requires more accurate estimates of glacier surface mass balance (SMB) and snow conditions (e.g., snow cover and snow depth).

    To derive the present and future state of glaciers and snow conditions in HMA, a variety of modelling approaches have been used, ranging from glacier models (Kraaijenbrink et al., 2017; Marzeion et al., 2020) to regional climate models

(RCMs). RCMs have been applied with one-way or two-way coupling to glacier models (Mölg and Kaser, 2011; Collier et al., 2013; Bonekamp et al., 2019; Rahimi et al., 2019; de Kok et al., 2020). Glacier evolution models are usually forced by statistically downscaled global climate models (GCMs) or by datasets based on observations or reanalysis. These models are computationally efficient, which enables them to be applied at fine horizontal resolution (varying from ~100 m to 50 km) or across many glaciers over a long period of time (Marzeion et al., 2020). One limitation, however, is that glacier models use

one-way coupling with the atmosphere without including feedbacks between the glacier or land surface and the atmosphere (Collier et al., 2013). Also, uncertainties are introduced due to the scale mismatch between coarse-gridded climate models and datasets, and small-scale mountain glaciers embedded in complex topography (Mölg and Kaser, 2011; Collier et al., 2013). To overcome these limitations and better understand the atmospheric drivers and their feedbacks with glaciers and snow, RCMs can be suitable tools, since they can be applied at high spatio-temporal resolution with horizontal grid spacing

as fine as 1 km (Bonekamp et al., 2019). RCMs can better resolve the complex topography, while also providing detailed meteorological fields to compare with in-situ observations and providing more information about land-atmosphere interactions (van Kampenhout et al., 2019). RCMs, however, need to be forced by GCMs or global reanalysis products, which disables a two-way interaction between the region of interest and the global domain.

GCMs can be more appropriate modelling tools since they allow two-way interactions, avoiding inconsistencies between

RCMs and GCMs in terms of dynamics and physics parameterizations (Huang et al., 2016). Until now, GCMs have generally not been considered as suitable for simulating SMB or snow conditions due to insufficient spatio-temporal resolution, model biases, and unresolved snow/ice physical processes, such as meltwater refreezing and snow-albedo feedbacks (Vizcaino, 2014). Recent improvements, however, have increased the suitability of some GCMs or Earth System Models (ESMs) for simulating SMB and snow conditions. These improvements include: 1) multilayer snow models with

parameterizations for snow densification, refreezing, and albedo (Flanner and Zender, 2005; van Kampenhout et al., 2017), 2) elevation tiles or classes to account for subgrid variability and downscaling SMB with altitude (Lipscomb et al., 2013; Shannon et al., 2019), and 3) surface-energy-balance schemes to simulate snow and ice melt (Vizcaíno et al., 2013). With these improvements, GCMs and ESMs such as the Community Earth System Model (CESM) have simulated increasingly realistic SMB over the Greenland and Antarctic Ice Sheets (Vizcaíno et al., 2013; Lenaerts et al., 2016; Sellevold et al.,

2019; van Kampenhout et al., 2020). This is not the case, however, for the SMB of glaciers in mountainous regions, such as HMA, which require high-resolution grids to capture complex and highly variable topography. This requires large computational resources that are often not available, and thus the application of GCMs or ESMs in mountainous regions has been limited.

Variable-resolution CESM (VR-CESM; Lauritzen et al., 2018) is a potential alternative for applications in complex

mountainous terrains. VR-CESM is a hybrid between regional and global climate models, applying regional grid refinement to the atmosphere over a region of interest within a coarse-gridded global domain (Rhoades et al., 2016). In this way, a high-resolution grid can be applied regionally without a prohibitive computational cost for the global model. For example, Huang et al. (2016) conducted Atmospheric Model Intercomparison Project (AMIP) simulations with VR-CESM over the western USA at refined spatial resolution of 0.25° and 0.125° and reduced the computational cost by factors of 10 and 25,

respectively, compared to globally uniform high-resolution grids. VR-CESM has been used to study the impacts of regional refinement on, for instance, global circulation and climatology (Gettelman et al., 2018), and the sensitivity of underlying physics to changing model resolution (Gettelman et al., 2018; Herrington and Reed, 2020). VR-CESM has also been applied over mountainous areas. In the western USA and the Chilean Andes, it has been used with regional refinements up to 7 km

to simulate regional climate and snowpack (Huang et al., 2016; Rhoades et al., 2016, 2018; Bambach et al., 2021; Xu et al., 2021). Also, over the Tibetan Plateau, South Asia, and East Asia, it has been applied to study the regional climate and snow characteristics (Rahimi et al., 2019; Xu et al., 2021; Jiang et al., 2023). The application of VR-CESM to simulate glacier SMB has been limited, thus far, to the Greenland Ice Sheet (van Kampenhout et al., 2019; Herrington et al., 2022).

The main aim of this study is to investigate the ability of the VR-CESM v2.1 to simulate SMB and snow conditions over High Mountain Asia. We also evaluate the simulation of other climatic and cryospheric-hydrological variables, such as precipitation and temperature. To this end, a VR grid is generated with horizontally refined grid spacings of 7 km (0.0625°) over HMA. The VR grid is used to run two transient model simulations covering a 20-year period, 1979–1998. The second simulation uses an updated glacier-cover dataset, snow and glacier model modifications, and revised cloud tunings. We compare the model results with gridded outputs derived from a globally uniform 1° CESM run, reanalysis- and satellite-based datasets, and a glacier model forced by the Weather Research and Forecasting (WRF) model. Compared to previous VR-CESM studies conducted for HMA (Rahimi et al., 2019), this study has several novelties. The VR-grid includes one more factor of refinement compared to Rahimi et al. (2019), who refined the grid to 14 km over the Tibetan Plateau and southern parts of HMA (i.e., excluding the Tien Shan). Also, this study is the first ESM application that investigates the simulation of mountain glacier SMB in High Mountain Asia, which is important for understanding present and future impacts of glacier changes on freshwater availability.

This paper is organized as follows. Section 2 highlights the methods and data and briefly describes the model. Section 3 presents and discusses the main outcomes, including their uncertainties and limitations, which will inform future model development. Section 4 gives conclusions.

## 2. Data and Methods

### 2.1 CESM Overview

We use the Community Earth System Model version 2.1 (CESM2; Danabasoglu et al., 2020), a global Earth system modelling framework consisting of several components (including atmosphere, ocean, ocean waves, land surface, river transport, sea ice, and land ice) that can be run in partially or fully coupled mode. With partial coupling, the active components are replaced by external data or stub components. In this study, we apply CESM in a partially coupled mode with active prognostic atmosphere and land surface components and prescribed monthly sea ice and surface temperature (i.e., replacing the active ocean and sea ice components). This configuration follows the AMIP protocol (Gates et al., 1999; Hurrell et al., 2008).

The atmosphere component of CESM2 is the Community Atmosphere Model version 6 (CAM6) with a spectral-element dynamical core enabling VR capabilities (CAM6-SE) and a dry-mass vertical coordinate with 32 vertical levels and a model top at ~40 km (Zarzycki et al., 2014; Lauritzen et al., 2018; Gettelman et al., 2019a). CAM6 physics parameterizations include the Beljaars orographic drag parameterization scheme (Beljaars et al., 2004); a shallow convection, turbulence, and

cloud macrophysics scheme (CLUBB; Bogenschutz et al., 2013); a deep convection scheme (Zhang and McFarlane, 1995); a cloud microphysics scheme with prognostic treatment of precipitation (MG2; Gettelman and Morrison, 2015); and a modified modal aerosol module (MAM4; Liu et al., 2016). More detailed information about CAM6-SE can be found in Lauritzen et al. (2018) and Gettelman et al. (2019a).

The land surface component is the Community Land Model version 5 (CLM5; Lawrence et al., 2019), which is applied with satellite phenology (CLM5-SP). CLM5 simulates the surface energy balance, hydrology, biogeochemical cycles, and their interactions with the atmosphere. Compared to previous CLM versions (e.g., Oleson et al., 2013), CLM5 includes several new and updated parametrizations, e.g., for snow, glaciers, and surface characterization (Lawrence et al., 2019). CLM5 has a multilayer snow model with up to 12 layers of snow. By default, the maximum allowed snow depth ($H_{max}$) is 10

m water equivalent (w.e.), allowing for meltwater refreezing and firn formation. The snow model includes modifications for wind- and temperature-dependent fresh snow density and snow compaction (van Kampenhout et al., 2017). A snowpack radiative heating model (SNICAR) simulates snow albedo and two-way radiative transfer in each snow layer based on effective snow grain size, incoming solar radiation, and aerosol deposition (Flanner and Zender, 2005; Flanner et al., 2007).

CLM grid cells can be represented by five different land units (vegetated, urban, lakes, crops, and glaciers) that can be

subdivided into columns and patches. Glacier land units contain multiple independent columns for elevation classes (ECs) to account for large topographic gradients over glaciers. This scheme is used to downscale several atmospheric variables required to calculate the surface energy balance (SEB) and surface mass balance (SMB) in each EC (Lipscomb et al., 2013; Sellevold et al., 2019). Among the atmospheric variables, near-surface (2m) temperature and (optionally) downwelling longwave radiation are downscaled from the mean CLM grid cell elevation to the EC elevation based on uniform

environmental lapse rates of 6 K km$^{-1}$ and 32 W m$^{-2}$ km$^{-1}$, respectively (Lipscomb et al., 2013; Van Tricht et al., 2016). The downscaled longwave radiation in each EC is bounded by 0.5–1.5 times the grid cell mean value and is normalized to conserve the grid cell mean (Van Tricht et al., 2016). Other atmospheric variables, such as specific humidity, are downscaled by assuming a constant relative humidity with altitude (Lipscomb et al., 2013). In this study, we use 36 ECs instead of the default 10 ECs to better resolve the glacier elevation distribution over High Mountain Asia. The 36 ECs are also applied over

the Greenland and Antarctic glacier regions (glacier regions in CLM where EC downscaling is applied by default). There are 35 ECs at intervals of 200 m ranging from 0 m to 7000 m, and an additional EC representing glaciers above 7000 m. To each EC, we assign a weight based on the glacier area in that elevation range relative to the total glacier area in the grid cell. Weights are derived from a high-resolution topography dataset (Wijngaard et al., 2023; also see Supplement Section S1). ECs with zero weight are considered "virtual" and do not contribute to the CLM grid cell mean that is coupled to CAM, but

they do compute an SEB and SMB for diagnostic purposes.

CLM calculates the SEB using

$$MHF = SW_{net} + LW_{net} + SHF + LHF + GHF \qquad\qquad (1)$$

where MHF is the melt heat flux, $SW_{net}$ is the net solar radiation, $LW_{net}$ is the net longwave radiation, SHF is the sensible heat flux, LHF is the latent heat flux, and GHF is the conductive or ground heat flux, all with units of W m$^{-2}$. All terms are defined as positive downward (from the atmosphere toward the surface), except for GHF, which is defined as positive upward (from the subsurface to the surface).

The SMB in CLM is defined as the total mass flux resulting from capping excess snow, minus the total ice melt and sublimation (van Kampenhout et al., 2020). When the snowpack is thicker than the maximum allowed depth, the excess snow is transformed into ice, resulting in a positive SMB. When snowpack is absent, the SMB can become negative due to bare ice that is melting or sublimating. In all other cases, the SMB is equal to 0. The CLM definition of the SMB differs from the glaciological definition in that it does not include variations in snow depth. The SMB corresponding to the glaciological definition can be computed as (Lenaerts et al., 2019; van Kampenhout et al., 2019, 2020):

$$SMB = PR - RU - SU_{sfc} - SU_{ds} - ER_{ds} \tag{2}$$

where PR is the total rainfall and snowfall, RU is the runoff, $SU_{sfc}$ is the sublimation/evaporation at the surface, and $SU_{ds}$ and $ER_{ds}$ are the sublimation and erosion as a result of drifting snow, all with units of m w.e. yr$^{-1}$. Since CESM does not include the last two terms, these terms are assumed here to be zero.

The SMB and temperature simulated in CLM can be coupled to the Community Ice Sheet Model (CISM; Lipscomb et al., 2019), the CESM component that simulates the dynamics of glaciers and ice sheets (Muntjewerf et al., 2021). In this study, however, CISM is not used to simulate HMA glaciers, since CISM grids for HMA and other mountain glacier regions are still under development. To enable future testing of CISM with CLM forcing for HMA, the EC-level output from this study has been saved.

**2.2 HMA-VR Grid and Performance.**

Variable-resolution CESM uses a high-connectivity VR spectral element grid (hereafter HMA_VR7) that is generated by the SQuadGen software package (Ullrich, 2014). SQuadGen offers two refinement types, LOWCONN and CUBIT (Guba et al., 2014). While LOWCONN grids have better numerical stability, the refinement is constructed from large 2x2 element building blocks, making it difficult to generate grids with irregular refinement boundaries. The HMA_VR7 grid (Figure 1a) has horizontally refined grid spacings of ~7 km (0.0625°) over the entire HMA domain, with its boundaries determined by the trace of the landscape surrounding the Tibetan Plateau and adjacent HMA mountain ranges, such as the Himalayas. Because of the irregular refinement boundaries, the HMA_VR7 grid is constructed using the CUBIT method.

Three transition grids with horizontal grid spacings of ~14 km (0.125°), ~28 km (0.25°), and ~55 km (0.5°) serve as a buffer between the HMA domain and the global 1° (~111 km) domain. The 0.5° grid extends from the Black Sea in the west to the Yellow Sea in the east, and from the Indian Peninsula in the south to Siberia in the north. To ensure a smooth

transition between varying grid resolutions as well as numerical stability, the buffer zone spans up to five spectral elements, with spring dynamics used to smooth the grid-transition zones and to create nearly orthogonal angles between elements (Guba et al., 2014). The HMA_VR7 grid contains 226,964 horizontal grid points, about 4.7 times more than on a globally uniform 1° SE grid (48,600 grid points).

To find a balance between computational cost and numerical stability, the CAM-SE time-stepping performance of the HMA_VR7 grid was tested in an Aquaplanet environment, a simplified environment that assumes the Earth's surface to be covered by oceans only (Zarzycki et al., 2014). We achieved stability with time steps of 225s and 18.75s for CAM physics and dynamics, respectively, which are 8 and 16 times smaller than the CAM physics (1800s) and dynamics (300s) time steps on a globally uniform 1° SE grid. Hyperviscosity coefficients that prevent numerical artifacts and instability are scaled according to the dimensions of a grid element (Guba et al., 2014). For each halving of the grid resolution, the hyperviscosity coefficients decrease by one order of magnitude (Zarzycki et al., 2014). The VR-CESM simulations were performed on the NCAR's supercomputing facility Cheyenne. With a computational cost of about 90,000 hours per simulated year, the HMA_VR7 grid is about 5% cheaper than a globally uniform 0.25° SE grid (777,600 grid points), which costs about 95,000 hours per simulated year, and about 35 times more expensive than a globally unform 1° SE grid, which costs about 2,500 hours per simulated year. Using 1584–2520 processors (44–70 nodes), the throughput of the HMA_VR7 grid is about 0.3–0.7 simulated years per wall-clock day.

The spectral-element dynamical core used by CESM is currently based on the hydrostatic approximation; non-hydrostatic vertical acceleration terms are neglected, which are important for the representation of deep convection, gravity waves, and flow over topography (Jeevanjee, 2017; Liu et al., 2022). Conventionally, the horizontal scales at which non-hydrostatic terms become important are assumed to be O(10 km), the vertical scale of the troposphere (e.g., Wedi and Smolarkiewicz, 2009). This could raise the question whether it is appropriate or not to use a 7 km regionally refined grid in combination with a hydrostatic model. In literature, there is, however, a spread of the grid spacing at which non-hydrostatic dynamics becomes important. There are, for instance, studies that indicate non-hydrostatic dynamics are only important for grid spacings finer than 10 km. Jeevanjee (2017) illustrated that in the FV3 model, the non-hydrostatic and hydrostatic vertical velocities only begin to diverge at dx = 2km or finer. Jang and Hong (2016) highlighted prior studies (e.g., Dudhia, 1993, 2014; Kato, 1996; Janjic et al., 2001) indicating that non-hydrostatic dynamics are 'weak' at grid spacings coarser than 5 km. Similarly, a study of the IFS model using hydrostatic and non-hydrostatic versions found very little sensitivity down to 5 km (Wedi et al., 2010). Other studies indicate that non-hydrostatic dynamics are also important at grid spacings coarser than 10 km, which is suggested by the studies of Yang et al. (2017) and Liu et al. (2022), who show that non-hydrostatic terms are important at grid spacings up to 25 km, particularly in its representation of tropical convective systems. Due to the inherent difficulty in testing the null hypothesis (Liu et al., 2022), and the spread in literature of the grid-spacing at which non-hydrostatic dynamics becomes important, we believe there is insufficient basis to deem our model configuration inappropriate. However, we are aware that using the hydrostatic version in combination with a 7 km regionally refined grid could propagate model-related uncertainties into model outputs, such as precipitation. Nonetheless, the ability of the

hydrostatic VR-CESM to simulate mountainous climate with a 7 km regionally refined grid has successfully been shown over the mountain ranges of western USA (Rhoades et al., 2018).

## 2.3. Topography & Land Surface

The topography of the HMA_VR7 grid was interpolated from the 30-arcsec (~1 km) Global Multi-resolution Terrain Elevation Data (GMTED2010; Danielson and Gesch, 2011) of the United States Geological Survey (USGS) using an updated version of the NCAR Topo software package (Lauritzen et al., 2015). The updated software package includes ridge finding and internal smoothing algorithms to improve the accuracy of high-resolution topography. Figure 2 shows a snapshot of the topography over western HMA (the Karakoram and upper Indus Basin, 32°–37°N; 72°–78°E) for the globally uniform 1° and 0.25° SE CESM grids (hereafter NE30 and NE120, respectively, where NE stands for the number of elements in each coordinate direction on a panel (Lauritzen et al., 2018)) and for the HMA_VR7 grid and GMTED2010. Compared to NE30 and NE120, the HMA_VR7 topography represents the detailed topography of this region more accurately. Over HMA, the maximum altitude increases from 5170 m and 5684 m for NE30 and NE120, respectively, to 6228 m for HMA_VR7. These maxima are observed in the northeastern, northwestern, and southeastern HMA subregions (Figure 1b), respectively (Table 1).

The land surface characteristics of the HMA_VR7 grid are partly transient and partly constant. The distribution of plant functional types (PFTs) is transient; time series of land use changes are derived and interpolated from the Land-Use Harmonization (LUH2) time series (Hurtt et al., 2020), which covers the period 1850–2015 and describes annual land-use changes based on the History Database of Global Environment (HYDE version 3.2; Goldewijk et al., 2017). Other land surface classes, such as glaciers and lakes, are assumed constant using the distributions of the year 2000 by default.

## 2.4  VR-CESM Simulations Setup

We performed two transient model simulations covering a 20-year period, 1979–1998, using the HMA_VR7 grid. Both simulations use several model modifications, including a new glacier region over HMA with a 36-EC scheme in CLM. The new glacier region makes it possible to simulate SMB in multiple (including virtual) ECs in HMA, while retaining the computationally cheaper default behavior of one EC per grid cell in other mountain glacier regions. The HMA glacier region covers a domain between 55°E and 110°E and between 20° N and 50°N.

For the first simulation (hereafter HMA_VR7a), the model is spun up for 1 year. The maximum allowed snow depth ($H_{max}$) is reduced to 1 m w.e. (the default value in CESM1) under the assumption that a 1-year spin-up is not sufficient to reach $H_{max}$=10 m w.e. Other CLM settings follow the CESM2 defaults, including a bare-ice albedo of 0.5 (0.3) for the visible (near-infrared) wavebands, the application of longwave downscaling, and rain-snow repartitioning using temperature thresholds of -2°C (0°C) for snow and 0°C (+2°C) for rain over glaciated (non-glaciated) land units. The rain-snow repartitioning is based on downscaled temperature. It uses a linear ramp that assumes precipitation to fall as snow (rain)

when the temperature is below (above) the lower (upper) temperature threshold, and as a mix of rain and snow when the temperature is between the two thresholds.

For the second simulation (hereafter HMA_VR7b), the model is initialized with a snow depth of 2.5 m w.e. over glaciated land units, followed by a CAM spin-up of 10 years and a CLM spin-up of 50 years. For the CAM spin-up, CESM is run according to the AMIP protocol, keeping the solar and external forcing, surface emissions, ozone and volcanic aerosols fixed to the 1979 level. The coupler output of this spin-up is used to initialize CLM which is run in a standalone mode and sub-cycles over a 10-year period (i.e., the length of coupler output) for 50 years. A period of 50 years is sufficient to equilibrate the terrestrial system components, such as snowpack. The outputs of the atmospheric and land spin-up are then used to initialize the transient simulation. For HMA_VR7b, we made several model changes to reduce biases compared to HMA_VR7a. These changes include

- An increase in $H_{max}$ from 1 m w.e. to 5 m w.e. to increase the refreezing capacity
- An increase in bare ice albedo to 0.6 (0.4) for visible (near-infrared) wave bands (i.e., the default values in CESM1) to reduce the ice melt.
- Modified rain-snow repartitioning temperature thresholds to 0°C for snow and +4°C for rain over all land units, consistent with observed rain-snow temperature thresholds in HMA (Jennings et al., 2018).
- No downscaling of downward longwave radiation
- Tunings of cloud cover and sea ice, including the MG3 cloud microphysics scheme (Gettelman et al., 2019b). This is an update of the MG2 scheme that includes the effects of graupel or hail. The tunings also include:
  - An increase in the strength of the pressure damping (*CLUBB C11b*) for the third moment of vertical velocity in the large skewness regime from 0.35 to 0.4, to increase low-level cloudiness.
  - An increase in the microphysical auto-conversion size threshold for ice to snow (*micro_mg_dcs*) from 500 x $10^{-6}$ to 1000 x $10^{-6}$ m.
  - An increase in snow grain radius (*r_snw*) from 1.25 to 1.5 standard deviations and a decrease in snow melt onset temperature (*dt_melt*) from 1.5 to 1.0°C to tune shortwave radiation and albedo over sea-ice.
- An updated CLM glacier-cover dataset, which is described in more detail in Section S1 of the Supplement.

The differences between HMA_VR7a and HMA_VR7b are also listed in Table 2.

## 2.5 Reference Data

To evaluate the HMA VR simulations, we compare the gridded outputs with outputs from an NE30 CESM run that covers the same 20-year period (1979–1998). We use the following (re)analysis, observation-based, and satellite-derived products, and WRF-based output:

1. The ERA5 reanalysis (Hersbach et al., 2020) is used to evaluate geopotential height and air temperature over the period 1979–1998. ERA5 is available at a spatial resolution of 0.25° x 0.25°, at monthly intervals, and at 37 pressure levels.

2. The WFDEI dataset (Weedon et al., 2014) is used to evaluate 2m temperature, rainfall, and snowfall for 1979–1998. WFDEI stands for Watch Forcing Data methodology applied to ERA-Interim. The dataset has a spatial resolution of 0.5° x 0.5° and uses ERA-Interim output that is corrected for elevation using environmental lapse rates (i.e., for temperature). The data are bias-corrected using gridded observations from datasets developed by the Climate Research Unit (CRU) or the Global Precipitation Climatology Centre (GPCC). In this study we used the WFDEI datasets that are bias-corrected based on GPCC and are available on monthly intervals.

3. Northern Hemisphere (NH) snow-cover extent from the National Snow and Ice Data Center (NSIDC) (Brodzik and Armstrong, 2013) is used to evaluate snow cover for 1979–1998. The NSIDC snow-cover data are available on a 25 x 25 km EASE 2.0 grid at weekly intervals. To enable evaluation of monthly model output, the weekly NSIDC snow-cover data are converted to monthly intervals.

4. The High Asia Refined analysis version 2 (HAR v2; Wang et al., 2021) is used to evaluate snow depth. HARv2 is based on Weather Research and Forecasting (WRF) simulations that are dynamically downscaled with ERA5 and is available on a 10 x 10 km grid at monthly intervals. The snow depth in HARv2 is corrected with snow depth from the Japanese 55-year reanalysis (Kobayashi et al., 2015), which has shown good performance compared to other global reanalysis datasets in simulating snow depth (Orsolini et al., 2019). Since HARv2 data is not available for 1979, we only evaluate the model output for the period 1980–1998.

5. The Japanese 55-year reanalysis (JRA55; Kobayashi et al., 2015) is used to evaluate surface-energy-balance components, including all-sky shortwave and longwave surface radiation fluxes as well as latent, sensible, and ground heat fluxes for 1979–1998. JRA55 is available on a T319 Gaussian grid at monthly intervals.

6. The CERES Energy Balanced and Filled (EBAF) dataset (Loeb et al., 2018; Kato et al., 2018) is used to evaluate shortwave cloud forcing and all-sky shortwave and longwave surface radiation fluxes for 1979–1998. The dataset is available on a 1° x 1° grid at monthly intervals. To evaluate the model output over a similar period (20 years), we use CERES EBAF data covering the period 2001–2020.

7. Glacier SMB is evaluated for 1979–1998 based on two data sources:
    - Area-weighted specific mass change rates derived from glaciological and geodetical observations (Zemp et al., 2019). The observation-based mass change rates are regional estimates based on the RGI regions Central Asia, South Asia West, and South Asia East.
    - Gridded 1° x 1° SMB outputs from a glacier mass balance gradient model (Kraaijenbrink et al., 2017) forced by temperature and precipitation fields from the Weather Research and Forecasting (WRF) model

(de Kok et al., 2020). The gridded SMB outputs are not available for 1979 and only cover the period 1980–1998.

To allow a uniform comparison among the different CESM grids and the (re)analysis and satellite-based datasets, the CESM output and reference data (e.g., WFDEI, NSIDC, HARv2, JRA55, CERES-EBAF, and ERA5) are regridded to a 1° finite-volume grid, unless noted otherwise.

**3 Results and Discussion**

**3.1 Northern Hemisphere Atmosphere**

Before evaluating the model's ability to simulate cryospheric-hydrological variables such as the glacier SMB, we evaluate its ability to simulate the Northern Hemisphere atmosphere. To this end, we compare geopotential height and air temperature variables simulated by the different CESM grids, using ERA5 output as a reference. Figure 3 shows the lower tropospheric 335 (500–1000 hPa) eddy geopotential thickness in winter and the upper tropospheric (500–200 hPa) mean temperature in summer (The surface of the Tibetan Plateau lies at about 500 hPa). The lower tropospheric eddy geopotential thickness in the left panels of Figure 3 shows the model's ability to simulate stationary wave patterns. The eddy thickness simulated by the NE30 and HMA VR configurations shows many similarities with the ERA5 output, along with some small differences. Both the CESM output and ERA5 output have a stationary wave pattern over the Northern Hemisphere with two negative eddy 340 thickness centers over Siberia and Canada, and two positive eddy thickness centers over the North Pacific and the North Atlantic. The negative eddy centers have a similar shape and magnitude, whereas the positive centers are similar in shape with small differences in magnitude. NE30 is more similar to ERA5 over the North Pacific, whereas the HMA VR configurations are more similar to ERA5 over the North Atlantic.

      The upper tropospheric summer temperature biases (relative to ERA5) can help to understand the effects of atmospheric 345 biases on temperature-sensitive surface variables such as ice melt and snow melt. However, atmospheric biases do not necessarily need to correspond with surface temperature biases as we will show in Section 3.3. Compared to ERA5, NE30 shows an elongated warm anomaly over the midlatitudes with the largest warm bias (about +2 K) over Mongolia and northern China. The HMA VR configurations show a similar pattern but with a larger warm bias (about +3K), which is likely related to its higher horizontal resolution. This is a known issue in GCMs and ESMs, particularly at midlatitudes 350 (Roeckner et al., 2006; Herrington and Reed, 2020). One explanation is that increasing horizontal resolution leads to larger resolved vertical velocities and increased condensational heating. This heating, computed in the macrophysics routine of CLUBB, especially raises upper tropospheric temperature at midlatitudes (Herrington and Reed, 2020).

## 3.2 Cloud Forcing

Figure 4 shows the model's annual mean shortwave cloud radiative forcing (SWCF) biases relative to the CERES-EBAF
product. SWCF quantifies the impact of clouds on incident shortwave and is computed as the difference between all-sky and
clear-sky shortwave radiative fluxes at the top of the atmosphere. NE30 shows biases typical of CESM2, with positive biases
near the marine stratocumulus decks off the coast of the southwestern U.S. The HMA_VR7a simulation has larger biases
than NE30, with more positive biases almost everywhere in the Northern Hemisphere, indicating thinner clouds. This
reduction in cloud thickness is primarily due to the smaller physics timestep in the HMA configurations. HMA_VR7b was
tuned to remove this sensitivity to physics timestep, resulting in a SWCF bias that looks more similar to the NE30 run
(Figure 4b). However, the positive SWCF biases over HMA remain in HMA_VR7b.

Figure 5 shows the summer mean SWCF, which contributes to summer melting. The clouds over the Himalayan front
range are too expansive in NE30 compared to CERES-EBAF, spanning 100s of km in the north-south direction. The HMA
VR configurations instead produce a narrow SWCF feature over the front range that is more similar to observations (Figure
5a). However, the clouds have mostly disappeared from the northern half of the Tibetan Plateau compared to the NE30
control and CERES-EBAF, as also shown in the annual mean plots in Figure 4. Our cloud tunings in HMA_VR7b are
therefore unable to restore clouds locally over the Tibetan Plateau. The VR model's inability to produce realistic mean cloud
states inside and outside the refined region is a major limitation of the variable-resolution method (Rauscher et al., 2013).

The bottom panel of Figure 5 shows the summer mean precipitation rates over HMA in the models and the WFDEI
validation product. NE30 does not capture the intensity and location of the summer precipitation patterns, in particular over
the west coast of India and the eastern side of the Himalayas, extending southward to the eastern coasts of Bangladesh and
Burma. In contrast, the HMA VR configurations improve the intensity and location of these features, indicating more
realistic monsoonal rainfall.

To explore the causes of the cloud and precipitation changes in the HMA VR runs, Figure 6 shows latitude-height
transects averaged over the longitude band 80˚-100˚. The monsoonal circulation in the NE30 run has two centers, a broad
region of ascent in the southern HMA region, primarily over the Indian Ocean, and a narrower region of ascent over the front
range of the Himalayas (Figure 6d). These ascent centers are collocated with condensational heating from CLUBB (Figure
6g), which sustains the monsoonal circulation. In the HMA VR runs, the ascending center associated with the front range is
displaced south, merging with the southern ascent center and manifesting as a single, broad region of ascent (Figures 6e-f).
This new circulation is more intense than the NE30 circulation, as shown by a broad region of anomalous condensational
heating upwind of the front range (Figures 6h-i) and coinciding with the larger mean precipitation rates in Figure 5.

The anomalous ascent on the south side of the mountains is balanced by anomalous descent on the north side of the
plateau (Figures 6e-f). This anomalous descent likely explains the increased temperatures (Figures 6k-l), reduced humidity
(Figures 6n-o), and reduced cloudiness (Figures 6q-r) on the north side of the plateau, as these are all consistent with a
subsiding environment. While the warming and drying patterns are largely the result of greater vertical velocities due to the

enhanced spatial resolution in the HMA VR runs, the shorter physics timestep also contributes to this warming and drying (not shown).

## 3.3 Surface Temperature & Precipitation

Whereas the summer tropospheric temperature over HMA rises with increasing horizontal resolution, the near-surface (2-m) temperature falls. Figure 7 shows the 2-m temperature differences and pattern correlations between the NE30 and HMA VR simulations and the observation/reanalysis-based WFDEI for each season. The differences are shown over four HMA subregions (Figure 1b), and the pattern correlations for the domain 20°-50°N, 65°-105°E. During summer (JJA), NE30 has a warm temperature bias (relative to WFDEI) up to ~2.5°C (here and hereafter, numbers in brackets referring to a bias represent the median bias) over all HMA subregions. This bias decreases with increasing resolution. For HMA_VR7, the JJA warm bias is present only in northern HMA (up to ~2.5°C), whereas southern HMA has a cold bias relative to WFDEI (up to ~2°C). During winter (DJF), spring (MAM) and autumn (SON), cold biases are present over SW-HMA (the region including the Karakoram and Hindu Kush Mountain ranges), growing larger (up to ~8°C) with increasing resolution. Over other HMA subregions, cold biases are also present, but are smaller (up to ~2°C). The cold temperature biases could partly be a result of uncertainties in WFDEI. Since WFDEI is bias-corrected using gridded observations of GPCC, the accuracy of the data relies on the availability of meteorological measurements that are scarce in High Mountain Asia, especially at higher altitude and in the more remote domains of HMA. The lack of measurements could result in temperature overestimates and precipitation underestimates (Lalande et al., 2021; Gu et al., 2012; Palazzi et al., 2015; Immerzeel et al., 2015), which can to some extent explain the cold temperature biases and, as we will show later, the wet precipitation biases that are visible in the NE30 and HMA VR simulation outputs.

The contrary temperature response between the atmosphere (increasing temperature with finer resolution) and land surface (decreasing temperature with finer resolution) can be explained by a combination of better resolved topography and multiple elevation classes for the fine grids. Whereas the NE30 runs have a single EC, the VR runs have 36 ECs for HMA glaciers. With multiple ECs, the atmospheric temperature received from CAM is downscaled with altitude over glaciated land units, which can lower the grid cell mean temperature in highly glaciated regions where glaciers lie above the grid cell mean elevation. The bias difference between NE30 and HMA_VR7 is especially prominent in SW-HMA, the most glaciated subregion. The pattern correlations between WFDEI and CESM as shown in Figure 7b are above 0.95, with the highest correlations in JJA and for HMA_VR7b. This suggests that CESM is able to simulate spatial temperature patterns that agree with the observation/reanalysis-based patterns shown by WFDEI.

Figure 8 shows the monthly mean precipitation (rainfall and snowfall) differences and pattern correlations for the NE30 and HMA VR simulations and WFDEI. Rainfall is overestimated relative to WFDEI during JJA in southern HMA (up to ~150 mm month$^{-1}$), particularly in SE-HMA, which is monsoon-dominated and the wettest HMA subregion with summer rainfall sums of 740 mm in WFDEI (Table S2). The interquartile range of the rainfall bias is similar among the CESM grids,

whereas the median of the rainfall bias is slightly larger for NE30 than for HMA VR in most subregions. The pattern correlation between WFDEI and CESM are worse during DJF, with correlations of 0.87–0.90, and best during SON, with correlations of 0.93–0.96. The best-performing grid relative to WFDEI varies per season, NE30 is better in DJF and HMA_VR7 in the other seasons. Snowfall is overestimated in the southern HMA subregions during DJF and MAM (up to ~50 mm month$^{-1}$), particularly in SW-HMA where winter snowfall sums up to 99 mm in WFDEI (Table S2). The snowfall bias is generally largest for HMA_VR7b, followed by HMA_VR7a and NE30. The snowfall bias is likely related to the cold bias in this region (Figure 7) and to some extent to the changed rain/snow repartitioning temperature thresholds in HMA_VR7b, which favor snowfall. The snowfall (and rainfall) biases could also be a result of the aforementioned uncertainties in WFDEI. Although the snowfall biases are large during DJF and MAM, the pattern correlations suggest that CESM, with correlations of 0.8–0.86, can reasonably simulate the snowfall patterns of WFDEI, with HMA_VR7 performing better than NE30. Only in JJA do the simulated snowfall patterns show a large deviation from the WFDEI snowfall patterns. The NE30 run has the worst performance, with a pattern correlation of 0.37. The HMA_VR7 grid gives pattern correlations of 0.68–0.7, consistent with the improved precipitation shown in Figure 5.

### 3.4 Snow Depth & Cover

Figure 9 shows the monthly mean snow-cover and snow depth differences and pattern correlations between the NE30 and HMA VR configurations, the observation-based NSIDC dataset, and the analysis-based HARv2 dataset. In most HMA subregions, snow cover is overestimated during winter (up to ~40%) and underestimated during summer (up to ~40%). Only in NE-HMA is the NE30 and HMA VR snow cover comparable to the NSIDC snow cover. The HMA VR simulations show some improvements compared to NE30 during spring and summer, but there are no significant differences between HMA_VR7a and HMA_VR7b. The pattern correlations between the NSIDC data and the NE30 and HMA VR runs are worst during JJA, with correlations of 0.69–0.72, and best during DJF, with correlations of 0.88–0.92. In general, HMA_VR7b has the best snow cover followed by HMA_VR7a and NE30. Compared to HARv2, snow depth is particularly overestimated over SW-HMA during winter and spring (up to ~100 mm w.e.), in part because of the cold bias. The snow depth from HMA_VR7a has a smaller bias than NE30, but the snow depth from HMA_VR7b shows a larger bias than HMA_VR7a. The increased bias is likely related to the snow model modifications in HMA_VR7b, including a higher maximum snow depth and changed rain/snow repartitioning thresholds. The pattern correlations show a better performance of NE30 and HMA_VR7 during winter and spring than during summer and autumn. In particular, HMA_VR7b underperforms relative to HARv2.

### 3.5 Surface Energy Balance

Figure 10 shows the annual cycle of surface-energy-balance (SEB) components over glaciated grid cells (encompassing all land units) in HMA. The downwelling, upwelling, and net shortwave radiation (SW$_d$, SW$_u$, and SW$_{net}$, respectively) for NE30 and HMA_VR7 are overestimated relative to CERES-EBAF but generally are closer to JRA55 (Figs. 10a-c, Table 3).

SW radiation fluxes are higher for HMA_VR7 than for NE30. The downwelling and upwelling longwave radiation ($LW_d$ and $LW_u$, respectively) for NE30 and HMA_VR7 are generally underestimated relative to CERES-EBAF, particularly in winter (Figs. 10d-e, Table 3). In summer, the $LW_u$ simulated by HMA_VR7 agrees more closely with CERES-EBAF than NE30 and JRA55, whereas the $LW_d$ is more accurately simulated by NE30 (relative to CERES-EBAF, Table 3). The net LW radiation ($LW_{net}$) is underestimated relative to CERES-EBAF while NE30 is in a closer agreement with CERES-EBAF than

HMA_VR7 (Fig. 10f, Table 3). The differences in SW and LW radiation fluxes among the different grids and reference output are likely related to differences in cloud cover. As described in Section 3.2, the cloud cover in CESM is too low, especially over the Tibetan Plateau and in the VR runs. With reduced cloud cover, more solar radiation reaches the surface and less thermal radiation is absorbed, increasing $SW_d$ and reducing $LW_d$. The reduced cloud cover can also contribute to the cold biases, especially during wintertime when the net radiation is already negative (Table 3). On daily basis, reduced cloud

cover could result in more shortwave insolation during the day, and enhanced radiative cooling during the night when the radiative balance is negative. The enhanced radiative cooling could then eventually lead to a (more) negative net daily radiative balance, especially during wintertime when the nights are longer, and the solar inclination angle is lower. Consequently, the surface temperature decreases, contributing to a cold bias.

    The seasonal cycle of the sensible and latent heat flux is compared only to JRA55. In summer, the upward sensible heat

flux (SHF) is higher for NE30 and HMA_VR7 than for JRA55, whereas the upward latent heat flux (LHF) is lower (Figs. 10g-h, Table 3). These biases could be caused by lower snow cover and subsequent lower sublimation and evapotranspiration. Summer surface albedo is lower for NE30 and HMA_VR7 than for JRA55 (Fig. 10i) but is in closer agreement with CERES-EBAF. In winter, the surface albedo for NE30 and HMA_VR7 is overestimated relative to JRA55 and CERES-EBAF, which could be linked to overestimated snow depth (Fig. 9).

The melt heat flux (MHF) (Fig. 10j, Table 3) is higher during summer for HMA_VR7 than for NE30 or JRA55. Here, HMA_VR7a has a higher MHF than HMA_VR7b, with more ice melt in summer. The greater cloud cover in HMA_VR7b (relative to HMA_VR7a) reduces $SW_d$ at the surface and increases $LW_d$, which overall results in lower net radiation (i.e., the sum of $SW_{net}$ and $LW_{net}$) and reduced melting and heating. NE30 and JRA55 have a lower summer MHF than HMA_VR7 because they are unable to simulate ice melt over High Mountain Asia.

The conductive or ground heat flux (GHF, Figure 10k, Table 3) simulated by JRA55 and NE30 is positive in autumn/winter and negative in spring/summer, whereas the GHF simulated by HMA_VR7 is negative only during spring and strongly positive during the other seasons. Here, GHF is positive for heat conduction from the bedrock toward the surface, which usually occurs when the surface is colder than the layers below (van Kampenhout et al., 2020). GHF is typically positive during winter, at night, or after refreezing of snowmelt, and negative during spring/summer and daytime.

The combination of a positive winter GHF partly countering the negative radiative balance, and latent and sensible heat fluxes close to zero (Table 3), could trigger a so-called stability-induced cooling feedback (Slater et al., 2001). Via the stability-induced cooling feedback, radiative cooling is enhanced, causing surface temperature to be lower than the overlying air temperature (i.e., a decoupling of the surface from the atmosphere), which could eventually also explain the cold biases.

The positive summer GHF simulated by HMA_VR7 is likely not related to refreezing of snowmelt (which is highest in spring and limited during summer, see Fig. 10l, Table 3), but rather to extensive ice melt, which requires heat extraction from the environment, and, in turn, also can contribute to cold biases. Greater MHF for HMA_VR7a than for HMA_VR7b requires a more positive summer GHF. Positive GHF is also observed over the ablation zones of the Greenland ice sheet (GrIS), although this is likely related to high rates of snow refreezing (van Kampenhout et al., 2020).

### 3.6 Surface Mass Balance

Figure 11 shows the area-averaged glacier surface mass balance derived from geodetical/glaciological observations, WRF-based output, and the HMA VR simulations for three Randolph Glacier Inventory (RGI) regions: Central Asia (NE-HMA + NW-HMA), South Asia West (SW-HMA), and South Asia East (SE-HMA). The observation- and WRF-based SMB are in the same range (between $+0.5 - -0.5$ m w.e. $yr^{-1}$), whereas the SMB of HMA_VR7 is too negative. The SMB bias is larger for HMA_VR7a, with SMB values of -2.5 to -4.0 m w.e. $yr^{-1}$. HMA_VR7b shows significant improvement with SMB values of -0.5 to -2.5 m w.e. $yr^{-1}$.

SMB improvements over southern and southeastern HMA are also visible in Figure 12, which shows the spatial distribution of mean annual SMB (m w.e. $yr^{-1}$) for 1979–1998 as simulated by HMA_VR7a and HMA_VR7b. The SMB for HMA_VR7a is mostly negative except for a few small regions in southernmost HMA where a positive SMB is simulated (Figs. 12a, b). The positive SMB in these regions can likely be attributed to high monsoon precipitation. The SMB is mostly negative in southeastern HMA, in part because of inaccuracies in the original glacier-cover dataset. The SMB for HMA_VR7b is less negative for the majority of glaciated grid cells, with the greatest improvements in southeastern HMA (Figs 12c-f), where the updated glacier-cover dataset is based on more accurate glacier outlines and includes glaciers located at altitudes between than 6000 m and 7000 m. Glaciers at these altitudes generally have a more positive SMB than lower-altitude glaciers (Figure 13). Also, HMA_VR7b has more grid cells with a positive SMB, especially in SE-HMA. Compared to HMA_VR7a, the number of grid cells with a positive SMB at the end of a model run nearly triples, from 105 to 299.

Table 4 shows the SMB components, which are integrated over the glacier area of the glacier-cover datasets used in the respective HMA VR simulations. To enable comparison between the two simulations, the SMB components of HMA_VR7a have also been integrated over the glacier area of the updated dataset (marked as HMA_VR7a_GC2). The largest positive SMB term in both simulations is precipitation. CESM simulates total annual precipitation of $111 \pm 6$ Gt $yr^{-1}$ ($90 \pm 5$ Gt $yr^{-1}$) and $93 \pm 4$ Gt $yr^{-1}$ in HMA_VR7a (HMA_VR7a_GC2) and HMA_VR7b, respectively. Integrated over the updated dataset, total annual precipitation is higher in HMA_VR7b, likely because of increased snowfall and rainfall during spring in southern HMA (Figure 8). For instance, in SE-HMA, rainfall and snowfall increase from 131 mm $month^{-1}$ in HMA_VR7a to 148 mm $month^{-1}$ in HMA_VR7b, possibly because of the tunings on cloud cover and the application of a newer microphysics scheme. The second largest positive SMB term is refreezing, which increases from $29 \pm 2$ Gt $yr^{-1}$ in

HMA_V7a_GC2 to 32 ± 2 Gt yr⁻¹ in HMA_VR7b. The increase in refreezing can mainly be attributed to the greater maximum snow depth.

The largest SMB loss term in the HMA VR simulations is total melt, which is dominated by ice melt. The total (ice) melt volumes decrease from 432 ± 23 Gt yr⁻¹ (356 ± 26 Gt yr⁻¹) for HMA_VR7a_GC2 to 324 ± 18 Gt yr⁻¹ (228 ± 20 Gt yr⁻¹) for HMA_VR7b. The decreased melt can be attributed to several factors, including 1) a decrease in net radiation with increased cloud cover, 2) increased snowfall and higher maximum allowed snow depth, which increase refreezing capacity and snow persistence, 3) improved accuracy of glacier-elevation distribution with the updated glacier-cover dataset, and 4) a higher albedo for bare ice. The latter increases the upwelling shortwave radiation. From Figure 10 and Table 3, it is, however, not apparent that upwelling shortwave radiation increases between the two simulations, because SEB components in Figure 10 and Table 3 are representative for the mean of all land units in glaciated grid cells. However, when selecting grid cells with a higher ice fraction (i.e., higher than 25%), the upwelling shortwave radiation increases from 104 W m⁻² for HMA_VR7a to 113 W m⁻² for HMA_VR7b, as a result of a higher bare-ice albedo. The second largest negative SMB term, sublimation and/or evaporation, is 12 Gt yr⁻¹ for both simulations. Overall, mass loss exceeds accumulation, giving a negative SMB. The integrated SMB is -352 ± 27 Gt yr⁻¹ for HMA_VR7a_GC2 and -224 ± 21 Gt yr⁻¹ for HMA_VR7b. The latter value is about 1.2 times higher than the Greenland ice sheet SMB (-182 ± 45 Gt yr⁻¹) and 7 times higher than the Antarctic ice sheet SMB (-32 ± 4 Gt yr⁻¹) for 2003–2016 (Lenaerts et al., 2019) (The HMA ranges here represent one standard deviation, whereas Lenaerts et al. (2019) stated ranges with two standard deviations). Both values simulated on the HMA_VR7 grid are much more negative than the observed ranges of -13 ± 17 Gt yr⁻¹ and -19 ± 2.5 Gt yr⁻¹ for 1979–1998 and 2000–2015/2016, respectively (Brun et al., 2017; Shean et al., 2020; Zemp et al., 2019).

## 3.7 Future Directions

Although many cryospheric-hydrological variables improve with increasing resolution, and VR-CESM can simulate positive SMB in parts of HMA, the negative SMB bias is cause for concern and suggests that model improvements are needed. There are several possible explanations for this bias. One explanation is the warm temperature bias over Central Asia, particularly over the Tibetan Plateau as suggested in Sections 3.1, 3.2, and 3.3. This warm bias could be related to underestimated cloud cover, which results in a higher net surface radiation due to increased downwelling solar radiation (partly offset by decreased downwelling longwave radiation, as suggested in Section 3.5). Improved convection and cloud parameterization schemes are needed. To make progress on the parameterization problem, we plan to carry out additional experiments nudging the upper vertical levels of CAM for variables such as horizontal wind, temperature, and humidity from ERA-Interim or ERA5, as previously done with WRF in HMA (de Kok et al., 2020). CAM contains flexible nudging capabilities (Wu et al., 2022; Kruse et al., 2022), with options for nudging over a particular region or level. The nudging tendencies by definition are collocated with the model bias, and an analysis of those tendencies constrains how the physics should interact with the dynamics at a particular grid resolution in order to provide a more realistic large-scale state, e.g., avoiding the anomalous

large-scale subsidence in the free-running VR runs responsible for cloud biases on the north side of the Tibetan Plateau. The nudging experiments also provide us with an opportunity to analyze the simulated surface mass balance given a realistic atmospheric state, thereby helping to further isolate the cause of the deficiencies to the atmosphere or surface components.

Another reason for the negative SMB bias could be a lack of representation of surface processes in CLM that are important in complex mountain environments. For example, CLM neglects the role of aspect and orientation in the SEB (via solar illumination), and the glacier SMB. Wang et al. (2022) found out that 27.19% of the HMA glaciers are subject to topographic shading, particularly for the north-facing glaciers of the Tibetan Plateau and the surrounding mountain ranges to the northwest and northeast (i.e., Kunlun Shan, Qilian Shan). Hillslope parameterization schemes (e.g., Swenson et al.,
2019), which include the effects of aspect and orientation of a hillslope on the incoming solar radiation, could improve the simulation of SEB and SMB. Further, CLM does not represent debris cover on mountain glaciers. About 12%–13% of the glaciers in HMA are at least partly covered by supraglacial debris which, depending on its thickness, can accelerate or reduce glacial melt (by reducing surface albedo or insulating the surface, respectively) (e.g., Herreid and Pellicciotti, 2020). Including debris cover in CLM surface datasets and adding debris-related glacial melt processes could be beneficial. Another
weak point is the simulation of orographic precipitation in CLM. Currently, precipitation can be repartitioned as snow or rain based on downscaled air temperature, but the effect of topography on total precipitation is ignored and the repartitioning applies fixed rain-snow temperature thresholds. Topography-based subgrid parameterizations such as the Elevation Ratio Weighted Method (ERWM; Tesfa et al., 2020) and more advanced rain-snow repartitioning schemes applying temperature-humidity-surface pressure-based rain-snow thresholds (Jennings et al., 2018) could improve the model's skill in simulating
high-altitude precipitation.

The negative SMB bias could also be related to the elevation-class (EC) downscaling scheme in CLM and the way CLM and CAM are coupled. Figure 14 shows the relation between grid-cell-mean SMB and glacier fraction (GCF) in 15 elevation zones at intervals of 250 m (based on the grid-cell-mean elevation). At lower elevations where glacier tongues reside, the glacier fraction is relatively low, whereas the most highly glaciated grid cells (with GCF higher than 90%) have a mean grid
cell elevation above 5000 m. In most elevation zones, SMB declines with decreasing glacier fraction. In some zones, such as 4000 m – 4250 m, the SMB is more than 2 m w.e. yr$^{-1}$ lower for sparsely glaciated cells (GCF of 0–10%) than for highly glaciated cells (GCF of 90–100%). In sparsely glaciated cells, elevation downscaling is only applied over glacier land units, whereas vegetated land units are excluded from downscaling. Moreover, the coupling between CLM and CAM is based on the homogenized state of a CLM grid cell, i.e., the average over all land units within the grid cell. This means that in grid
cells with small ice fractions, the impact of glaciers is limited, and the mean depends mainly on fluxes from vegetated land units. CAM receives this homogenized state and subsequently couples back to CLM, with downscaling of a few atmospheric variables, such as near-surface temperature. If the glacier patch lies higher than the grid-cell-mean elevation (and thus the elevation of the vegetated land unit), the glacier land unit sees a cooler air temperature than the vegetated land unit. However, the atmospheric temperature coupled to CLM could be dominated by the lower-lying and warmer vegetated land
unit. This could result in warmer glacier temperatures that make the SMB more negative. One potential solution is to apply

elevation downscaling over all land units using the hillslope model of Swenson et al. (2019). Another is to improve the subgrid coupling between CLM and CAM by introducing CLM patch information into CAM; this is the focus of the CLASP (Coupling of Land and Atmosphere Subgrid Parameterizations) project (Waterman et al., 2022). The necessity to improve land-atmosphere coupling (via elevation downscaling) is also suggested by offline 3-year (1979–1981) CLM simulations performed on the HMA_VR7 grid and driven by observation/reanalysis-based meteorological forcings of the Global Soil Wetness Project version 3 (GSWP3; Dirmeyer et al., 2006), a default offline mode in CLM. The offline CLM simulations (referred as HMALOa and HMALOb i.e., following the HMA_VR7a and HMA_VR7b settings, respectively) show likewise a declining SMB with decreasing glacier fraction (Figure S3), which suggests that improvement of land-atmosphere coupling is crucial. The offline CLM simulations show nonetheless that the colder and wetter surface climate in the HMA VR simulations help to reduce the SMB bias (Figures S4 and S5, Table S3).

In this study, the spectral-element dynamical core used by CESM is based on the hydrostatic approximation, which means non-hydrostatic vertical acceleration terms are neglected. Although we believe there is an insufficient basis to deem our model configuration inappropriate, we are aware that the hydrostatic version in combination with a 7 km regionally refined grid could propagate model-related uncertainties into model outputs, such as precipitation, wind, and cloud cover, which eventually also could affect the SEB and SMB. To address these model-related uncertainties, one potential solution could be to apply a similar regionally refined grid in combination with the newly developed MPAS (Model for Prediction Across Scales) non-hydrostatic dynamical core, which has recently successfully been applied over the western US mountain ranges (Huang et al., 2022).

Finally, the version of CAM used in this study has 32 levels in the vertical, which was developed for the standard 1° configuration in CESM2, and is a typical vertical resolution used for low top (~40 km) CMIP6 class models. This coarser vertical grid may not be appropriate when increasing the horizontal resolution beyond 1° (Skamarock et al., 2019; Lindzen and Fox-Rabinovitz, 1989), such as our HMA_VR7 configuration, but has nonetheless shown to improved model fidelity regardless (Gettelman et al., 2018; Herrington and Reed, 2020; Herrington et al., 2022; van Kampenhout et al., 2019; Rhoades et al., 2018), as also shown in our study here. A new 58 level low-top vertical grid slated for the CESM3 model may be preferable to reduce noise and spurious flow features.

## 4. Conclusions

We have investigated the ability of the variable-resolution Community Earth System Model to simulate cryospheric-hydrological variables such as glacier surface mass balance and snow conditions over High Mountain Asia. To this end, we developed a new VR grid with regional refinement up to 7 km over HMA. With this grid, we ran two 20-year (1979–1998) model simulations. The second model simulation includes an updated glacier-cover dataset and several atmosphere and land model modifications. We evaluated the results by comparison to gridded outputs derived from a globally uniform 1° CESM grid, reanalysis-, satellite-, and observation-based datasets, and an WRF-based glacier model.

The evaluations show that the large-scale circulation on the HMA_VR7 grid generally compares well to ERA5, but with an upper-troposphere warm temperature bias at mid-latitudes during summer. This warm bias grows with increasing horizontal resolution (+2 K for NE30 and +3 K for HMA_VR7). Further, the HMA VR runs have less cloud cover than NE30 runs, which is likely related to enhanced subsidence driven by more intense monsoonal circulation, and to the shorter physics time step in the HMA runs. The HMA VR runs also have a summertime warm temperature bias (up to about +2.5 °C) at the surface, but this bias is smaller than for NE30, likely because of elevation downscaling. Most HMA subregions have cold temperature biases during winter and to lesser extent during other seasons; these biases grow with increasing resolution. Overall, the HMA VR runs simulate rainfall, snowfall, snow cover, and snow depth better than NE30, but with some biases. Rainfall biases occur mainly in monsoon-dominated regions, whereas snowfall, snow cover, and snow depth biases are largest during winter and spring in southwestern HMA, in part because of the cold temperature bias and the HMA_VR7b model modifications. Snow cover is underestimated during summer but agrees better than NE30 with NSIDC snow cover.

The HMA VR runs have more downwelling shortwave radiation and less downwelling longwave radiation compared to NE30, JRA55 and CERES-EBAF. This can be attributed to negative cloud-cover biases that reduce longwave absorption and lower the albedo. The HMA VR runs have greater net radiation at the surface, resulting in higher melting and a positive (upward) conductive heat flux. The high volumes of snow and ice melting translate into an SMB that is more negative than observation-based glaciological/geodetical mass balances and WRF-based results. The HMA_VR7b simulation has a smaller SMB bias than HMA_VR7a, with an integrated SMB over glaciers of -224 ± 21 Gt yr$^{-1}$, compared to -352 ± 28 Gt yr$^{-1}$ for HMA_VR7a.

This study shows that VR-CESM generally reduces climatological biases relative to a coarse-resolution CESM grid and could be a useful tool for simulating cryospheric-hydrological variables, such as glacier SMB, in HMA. However, improvements are still needed to reduce cloud-cover and temperature biases, improve the model physics and land-atmosphere coupling, and thereby simulate SMB more accurately. This study could be a starting point for simulating cryospheric-hydrological variables in HMA and other mountain glacier regions with VR-CESM and other GCMs and ESMs.

*Code and Data Availability.*

Publicly available data are stored in two separate data archives on Zenodo. The first archive (https://doi.org/10.5281/zenodo.7864689) contains the model scripts and files that were used to create the updated glacier cover dataset. The second archive (https://doi.org/10.5281/zenodo.7864633) contains the NE30 and HMA_VR7 grid variables that were used to generate most of the figures in this manuscript. The remainder of the data are available on request.

*Author Contributions.*

R.R. Wijngaard, A. R. Herrington, W.H. Lipscomb, and G. R. Leguy designed the study. R.R. Wijngaard and A.R. Herrington modified the model code as needed. R.R. Wijngaard ran the model. R.R. Wijngaard and A.R. Herrington analyzed model results and prepared the text and figures. All authors contributed to the final text.

*Competing Interests.*

The authors declare that they have no conflict of interest.

*Acknowledgements.*

We would like to thank the editor and reviewers for their constructive remarks and suggestions that helped us to improve the
paper significantly. We greatly acknowledge Samar Minallah, Leo van Kampenhout, Bill Sacks, Julio Bacmeister, and Peter Hjort Lauritzen for the helpful discussions and support. This work is supported by the National Research Foundation of Korea (NRF) grant funded by the Korea government (MSIT) (NRF-2018R1A5A1024958). A. R. Herrington, W. H. Lipscomb, and G. R. Leguy are supported by the National Center for Atmospheric Research, which is a major facility sponsored by the National Science Foundation under cooperative agreement no. 1852977. Computing and data storage
resources, including the Cheyenne supercomputer (https://doi.org/10.5065/D6RX99HX, Computational and Information Systems Laboratory, 2019), were provided by the Computational and Information Systems Laboratory (CISL) at NCAR.

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

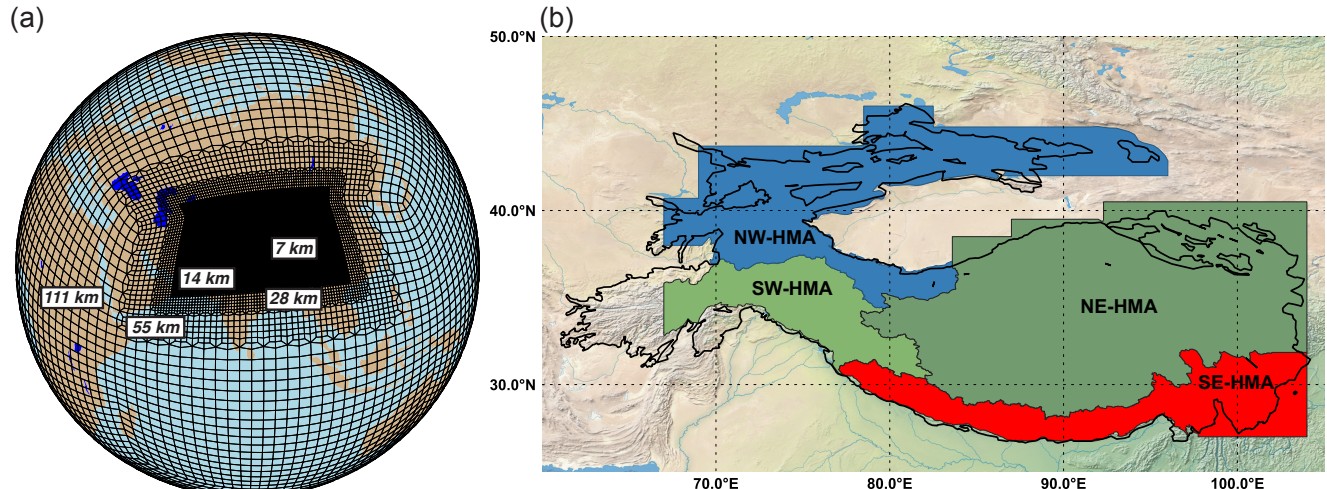

**Figure 1.** (**a**) Variable-resolution (VR) spectral element (SE) grid developed for this study, and (**b**) map showing High Mountain Asia (HMA) subregions used to evaluate cryospheric-hydrological variables, along with the outline of HMA. The source of the background imagery in Figure 1b is http://naturalearthdata.com/ (last access: 12 April 2022). The HMA subregions are derived from Randolph Glacier Inventory (RGI) version 6 regions (RGI Consortium, 2017), where SW-HMA and SE-HMA represent RGI regions South Asia West and South Asia East, respectively, and NW-HMA and NE-HMA represent RGI region Central Asia. Central Asia has been split into two subregions to better represent the regional climate characteristics. The HMA outline is derived from the Global Mountain Biodiversity Assessment (GMBA) Mountain Inventory version 1.2 (Körner et al., 2017).

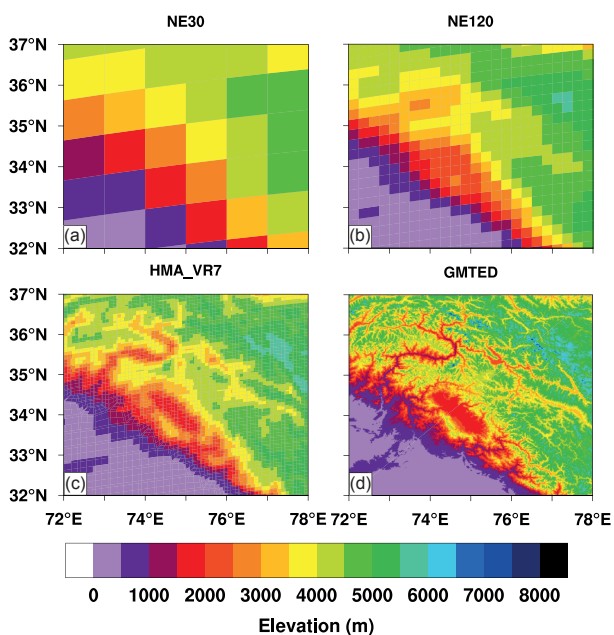

**Figure 2.** Topographical representation of western High Mountain Asia (i.e., Upper Indus River and Karakoram Mountains, 32°–37° N; 72°–78° E) by globally uniform 1° and 0.25° CESM grids (NE30 and NE120, respectively) (**a-b**), the HMA_VR7 grid (**c**), and a reference dataset, GMTED2010 (Danielson and Gesch, 2011) (**d**).


**Table 1.** Maximum altitude per HMA subregion (Figure 1b) as observed for NE30, NE120, HMA_VR7 and GMTED2010.

|  | **NW-HMA** | **NE-HMA** | **SW-HMA** | **SE-HMA** |
|---|---|---|---|---|
| **NE30** | 5162 m | 5170 m | 4978 m | 4452 m |
| **NE120** | 5684 m | 5526 m | 5513 m | 5369 m |
| **HMA_VR7** | 6167 m | 5778 m | 5870 m | 6228 m |
| **GMTED2010** | 7325 m | 7099 m | 8190 m | 8625 m |



**Table 2.** Overview of the modelling setup differences between the first and second HMA VR simulations (HMA-VR7a and HMA-VR7b, respectively).

| | Simulation 1 (HMA_VR7a) | Simulation 2 (HMA_VR7b) |
|---|---|---|
| Spin-up length | 1 year | 10 years for atmospheric component |
| | | 50 years for land surface component |
| Snow depth (m w.e.) | 1 m w.e. | 5 m w.e. |
| Bare ice albedo visible (near-infrared) | 0.5 (0.3) | 0.6 (0.4) |
| Longwave downscaling | Yes | No |
| Rain-snow temperature thresholds snow (rain) | -2 °C (0 °C) for glacier land units<br>0 °C (+2 °C) for non-glacier land units | 0 °C (+4 °C) |
| Glacier-cover dataset | Default | Updated (see Supplement Section S1) |
| Other tunings | | Tunings on cloud cover and sea ice +<br>MG3 cloud microphysics scheme |


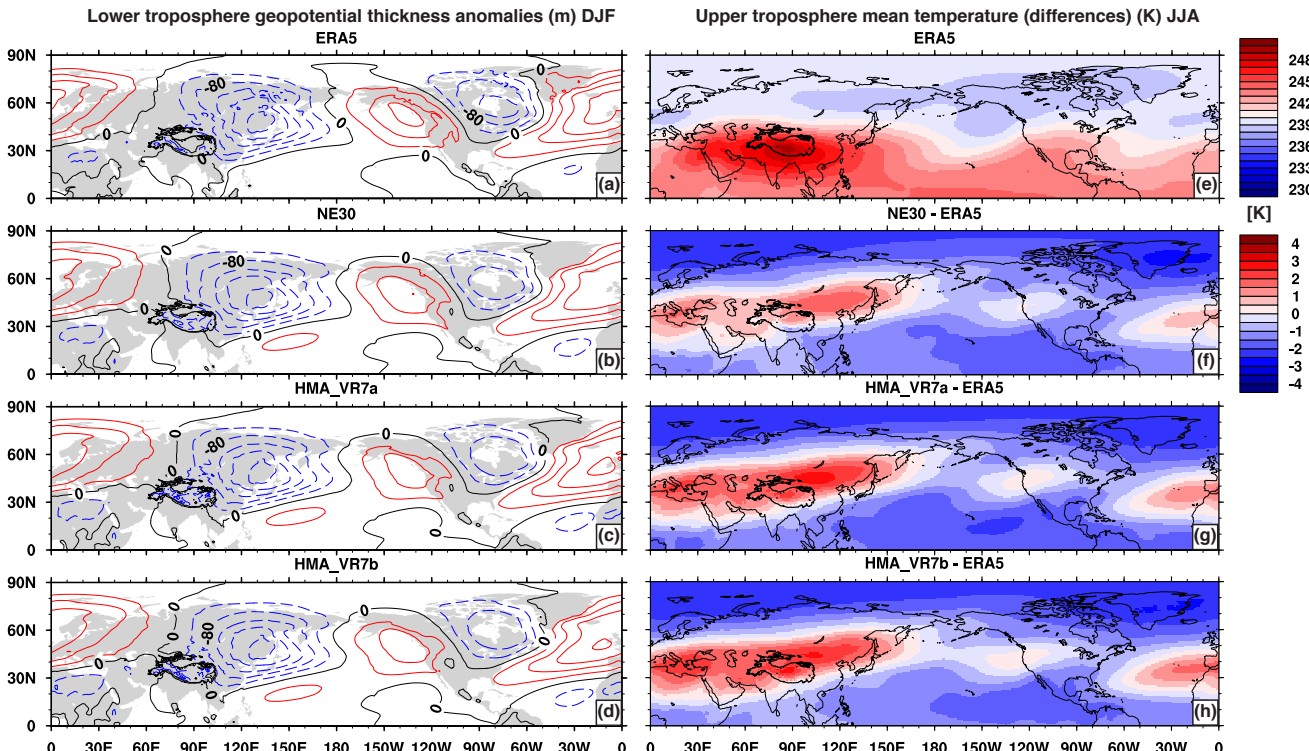

**Figure 3.** (**a-d**) Northern Hemisphere, lower troposphere eddy of the 500–1000 hPa geopotential thickness (m) in winter (DJF) for ERA5 (**a**), NE30 (**b**), HMA_VR7a (**c**), and HMA_VR7b (**d**). (**e-h**) Northern Hemisphere, upper troposphere (200–500 hPa) summer (JJA) mean temperature (K) for ERA5 (**e**), and the temperature differences relative to ERA5 for NE30 (**f**), HMA_VR7a (**g**), and HMA_VR7b (**h**).

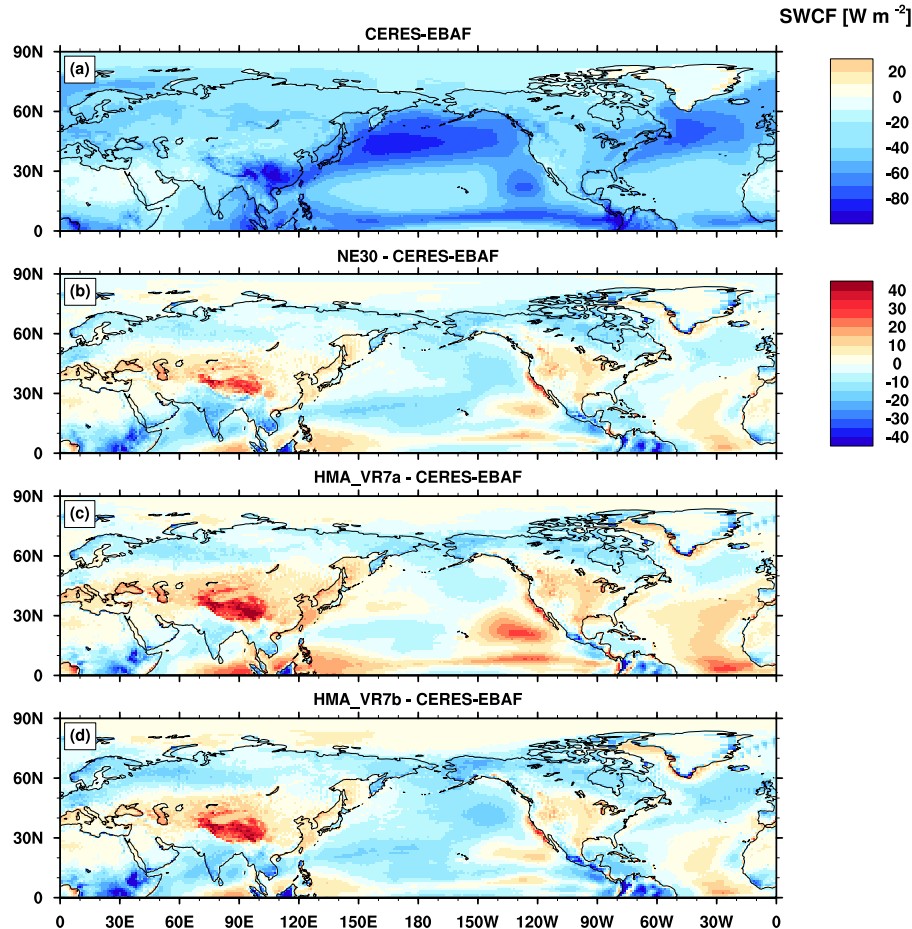

**Figure 4.** Northern Hemisphere annual mean shortwave cloud forcing (W m$^{-2}$) from CERES-EBAF (**a**) and model biases relative to CERES-EBAF (**b-d**) for NE30 (**b**), HMA_VR7a (**c**), and HMA_VR7b (**d**). All differences are computed after mapping model fields to the CERES-EBAF 1° grid.

025

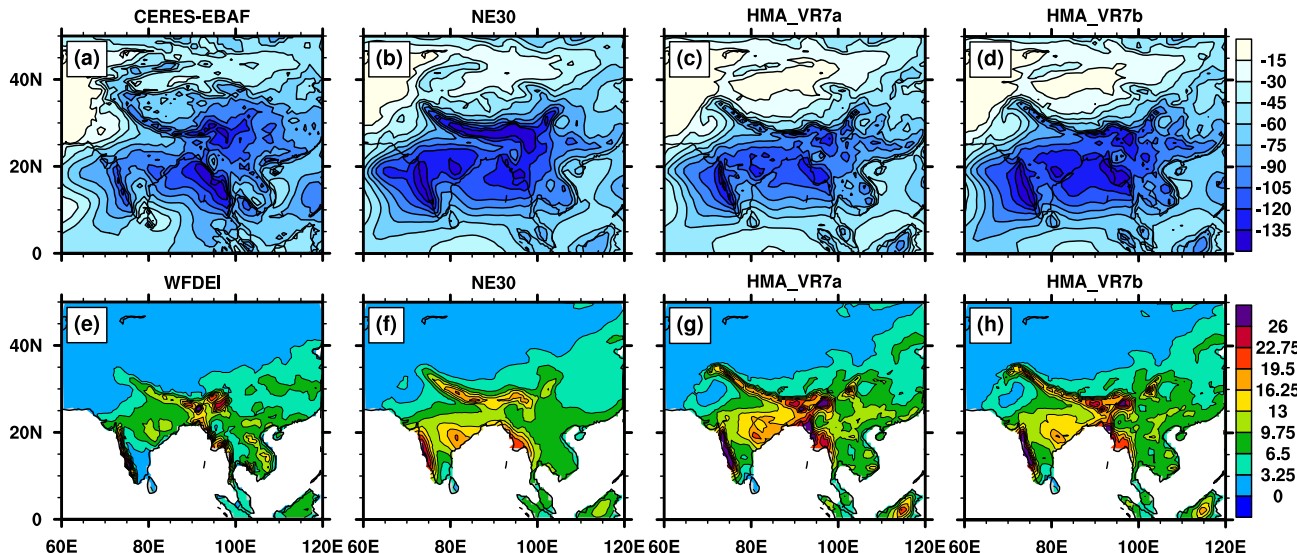

**Figure 5.** Northern Hemisphere summer climatological (**a-d**) shortwave cloud forcing (W m$^{-2}$) and (**e-h**) precipitation rate (mm/day) in observations and models. (**a**) CERES-EBAF, (**b,f**) NE30, (**c,g**) HMA_VR7a, (**d,h**) HMA_VR7b, and (**e**) WFDEI precipitation rates. Top panels show fields mapped to the CERES-EBAF 1˚ grid, and the bottom panels show fields mapped to the NE30 grid.

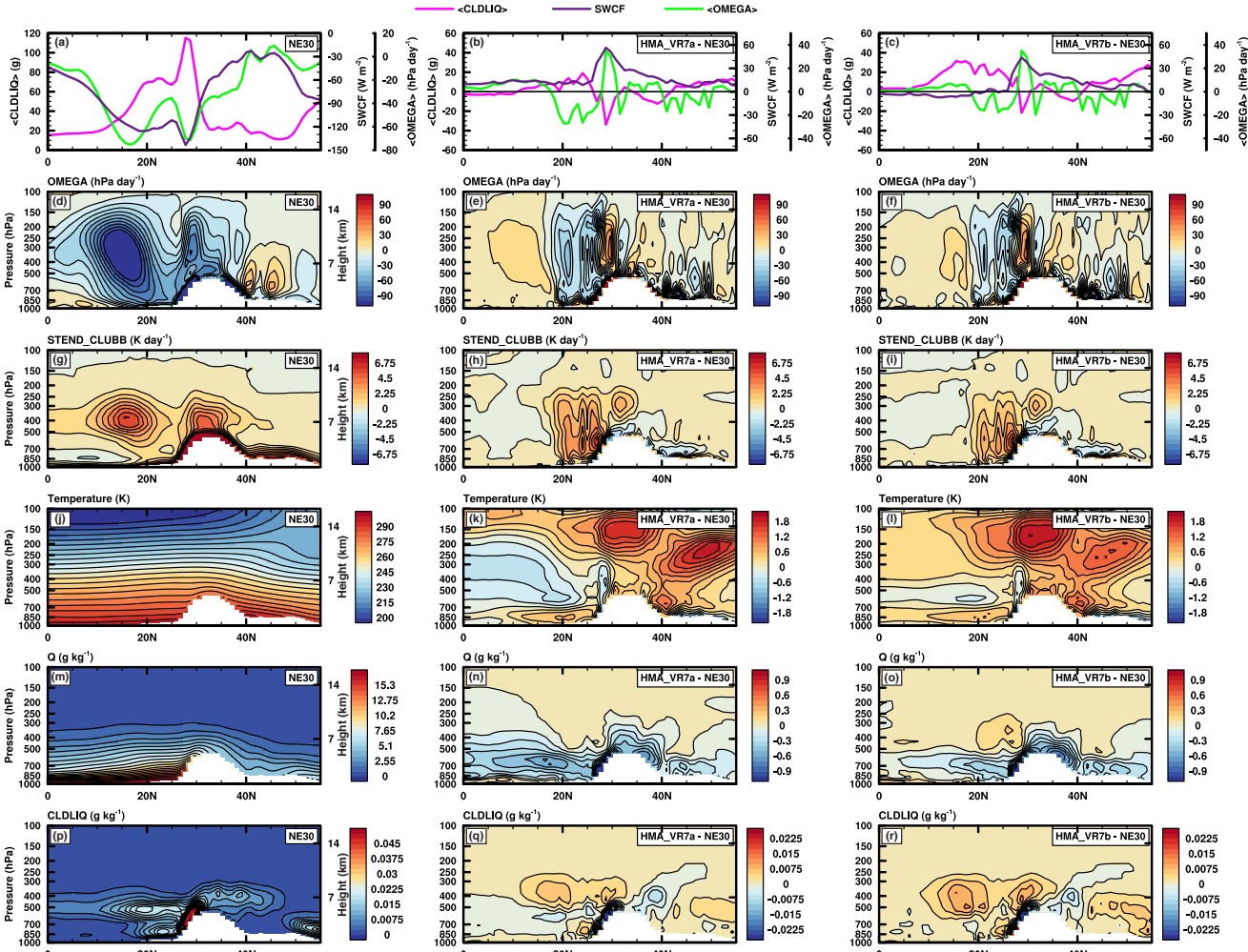

**Figure 6.** Northern Hemisphere summer climatological mass fraction of cloud liquid amount (CLDLIQ, g kg$^{-1}$), shortwave cloud forcing (SWCF, W m$^{-2}$), vertical velocity (OMEGA, hPa day$^{-1}$), CLUBB condensational heating (STEND_CLUBB, K day$^{-1}$), air temperature (K), and specific humidity (Q, g kg$^{-1}$) in a latitude-height transect averaged over 80°–100° longitude. (Left column) NE30, (middle column) HMA_VR7a differences relative to NE30, (right column) HMA_VR7b differences relative to NE30.

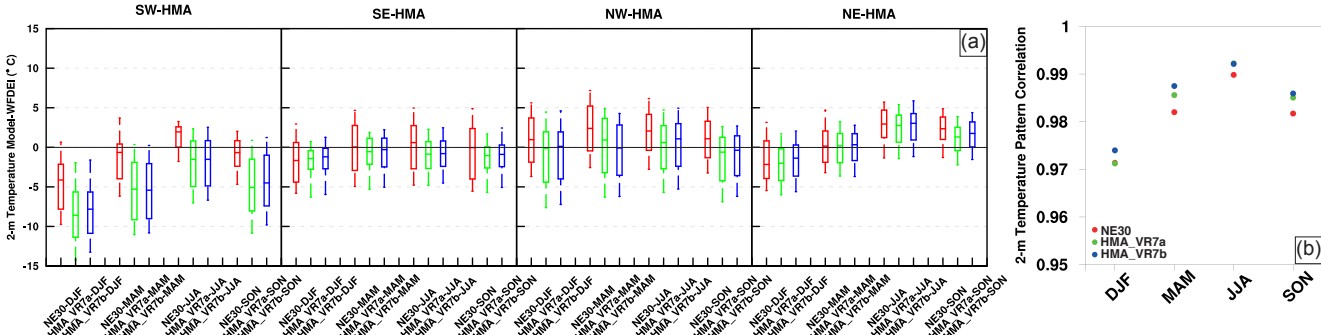

**Figure 7.** (**a**) Boxplots of 2-m temperature differences (°C) between the simulation outputs of NE30 (red), HMA_VR7a (green), HMA_VR7b (blue), and the observation/reanalysis-based WFDEI for each season and HMA subregion (shown in Figure 1b). The box represents the biases between the 25th and 75th percentile, the line in the box denotes the median, and the whiskers represent the 10th and 90th percentile of temperature differences. (**b**) Pattern correlations between the simulation outputs of NE30 (red), HMA_VR7a (green), HMA_VR7b (blue), and the observation/reanalysis-based WFDEI. The pattern correlations are calculated for each season and for the entire HMA region.

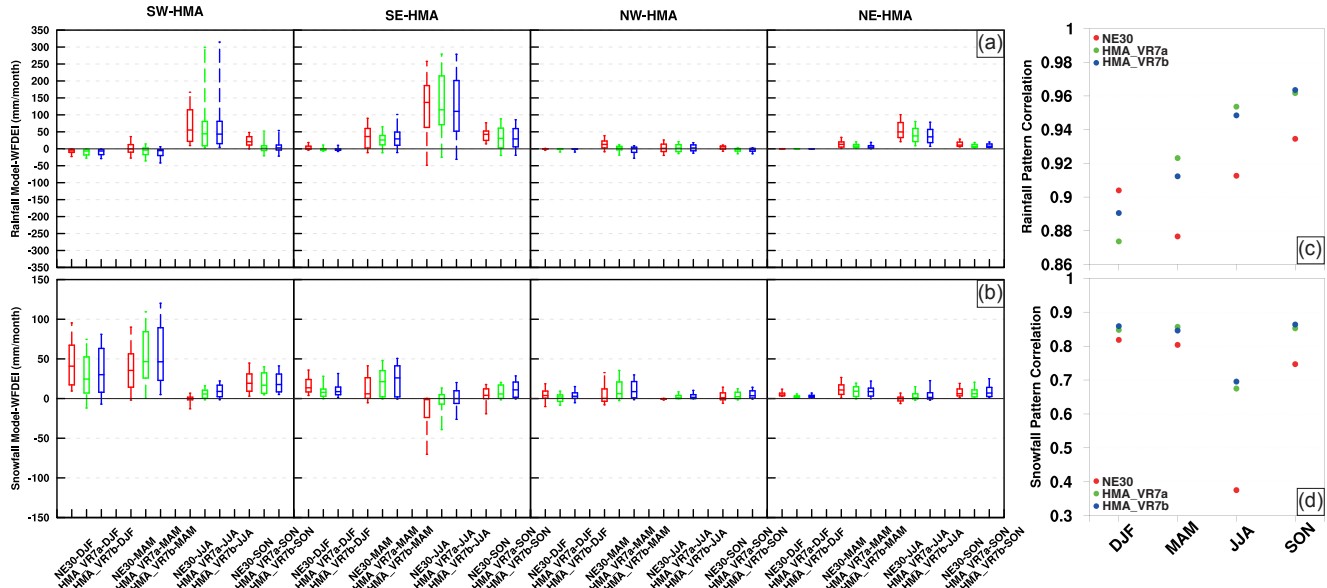

**Figure 8.** Same as Figure 7, but for rainfall (mm month$^{-1}$) (**a-c**) and snowfall (mm month$^{-1}$) (**b-d**)

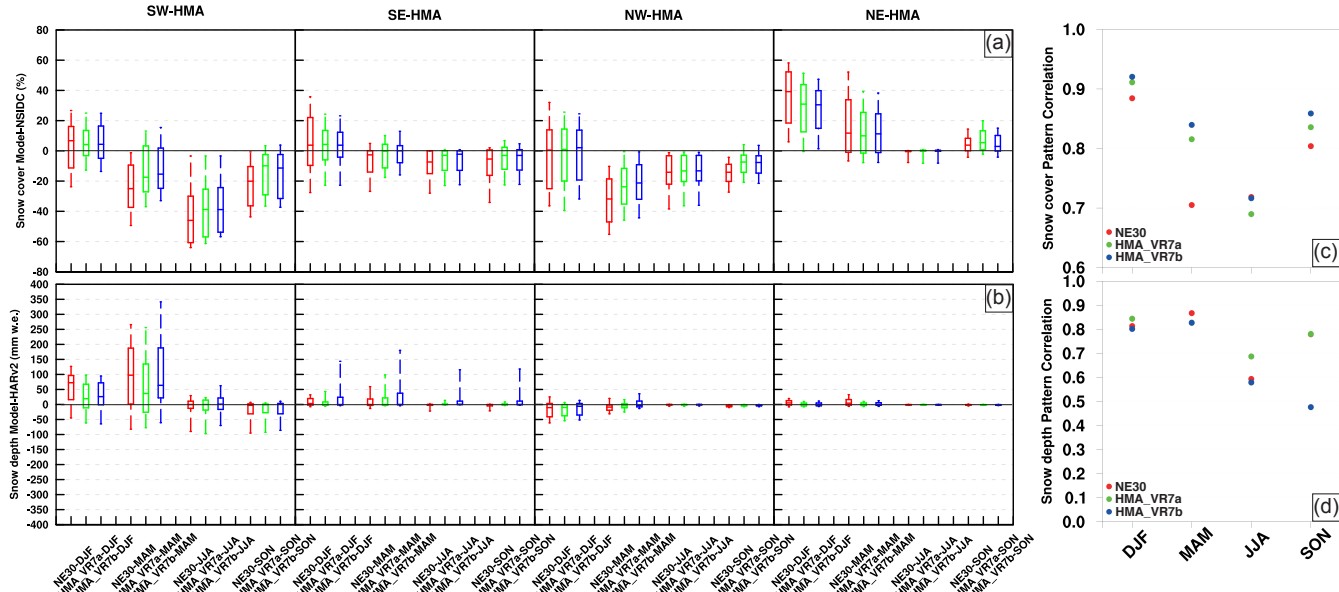

**Figure 9.** Same as Figure 7, but for snow cover (%) (**a-b**) and snow depth (mm w.e.) (**c-d**). The snow cover and snow depth differences are calculated between CESM simulation output (i.e., NE30, HMA_V7a and HMA_VR7b) and snow cover derived from NSIDC, and snow depth derived from HARv2, respectively.

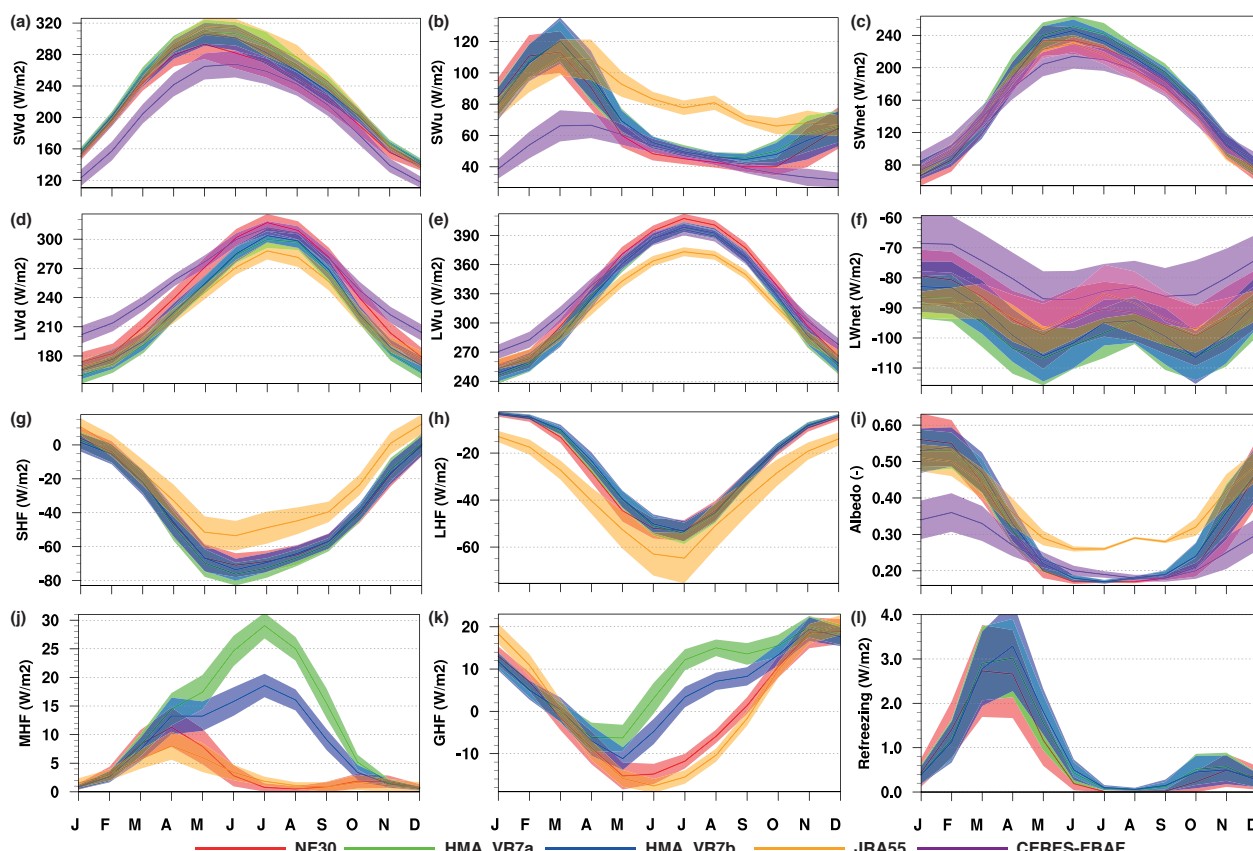

**Figure 10.** Annual cycle of (**a**) downwelling shortwave radiation (SW$_d$), (**b**) upwelling shortwave radiation (SW$_u$), (**c**) net shortwave radiation (SW$_{net}$), (**d**) downwelling longwave radiation (LW$_d$), (**e**) upwelling longwave radiation (LW$_u$), (**f**) net longwave radiation (LW$_{net}$), (**g**) sensible heat flux (SHF), (**h**) latent heat flux (LHF), (**i**) albedo, (**j**) melt heat flux (MHF), (**k**) conductive/ground heat flux (GHF), and (**l**) refreezing heat flux. The annual cycle of the various SEB fluxes (W m$^{-2}$) is representative for glaciated grid cells (encompassing all land units) in HMA over the periods 1979–1998 for NE30 (red), HMA_VR7a (green), HMA_VR7b (blue) and JRA55 (orange), and 2001–2020 for CERES-EBAF (purple). The shading denotes the standard deviation in time and SEB fluxes are defined positive towards the surface.

**Table 3.** Mean annual (ANN), wintertime (DJF|, in *italic*), and summertime (JJA, in *italic*) surface-energy-balance (SEB) fluxes (W m$^{-2}$) for NE30, HMA_VR7a (HMA1), HMA_VR7b (HMA2), JRA55 and CERES-EBAF (C-EBF). SEB fluxes are defined positive towards the surface and represent the mean fluxes in glaciated grid cells (encompassing all land units), averaged over the entire HMA region and the periods 1979–1998 (for NE30, HMA_VR7a, HMA_VR7b, and JRA55) and 2001–2020 (for CERES-EBAF).

| SEB fluxes (W m$^{-2}$) | ANN | | | | | DJF\|JJA | | | | |
|---|---|---|---|---|---|---|---|---|---|---|
| | NE30 | HMA1 | HMA2 | JRA55 | C-EBF | *NE30* | *HMA1* | *HMA2* | *JRA55* | *C-EBF* |
| SW$_d$ | 223 | 236 | 231 | 236 | 201 | *162\|267* | *168\|287* | *164\|281* | *162\|295* | *133\|256* |
| SW$_u$ | 66 | 71 | 70 | 83 | 48 | *87\|45* | *85\|51* | *83\|50* | *80\|81* | *42\|49* |
| SW$_{net}$ | 157 | 165 | 161 | 153 | 153 | *75\|222* | *83\|236* | *80\|230* | *81\|214* | *92\|207* |
| LW$_d$ | 242 | 226 | 230 | 225 | 255 | *179\|309* | *165\|293* | *171\|296* | *175\|280* | *207\|306* |
| LW$_u$ | 333 | 324 | 325 | 316 | 335 | *261\|401* | *253\|392* | *256\|393* | *263\|369* | *277\|391* |
| LW$_{net}$ | -91 | -98 | -95 | -91 | -80 | *-82\|-92* | *-88\|-98* | *-85\|-97* | *-89\|-89* | *-71\|-85* |
| R$_{net}$ | 66 | 67 | 66 | 62 | 73 | *-7\|130* | *-5\|138* | *-5\|133* | *-8\|125* | *21\|122* |
| SHF | -38 | -39 | -38 | -24 | - | *~0\|-68* | *-0.5\|-72* | *-2\|-69* | *7\|-49* | - |
| LHF | -26 | -25 | -24 | -36 | - | *-5\|-50* | *-4\|-50* | *-4\|-50* | *-14\|-60* | - |
| GHF | 1 | 9 | 5 | 1 | - | *13\|-11* | *12\|10* | *12\|2* | *16\|-15* | - |
| MHF | 3 | 12 | 9 | 3 | - | *2\|1* | *2\|26* | *2\|17* | *2\|2* | - |
| Refreezing | 0.81 | 0.92 | 0.95 | - | - | *0.7\|0.07* | *0.6\|0.19* | *0.6\|0.22* | - | - |

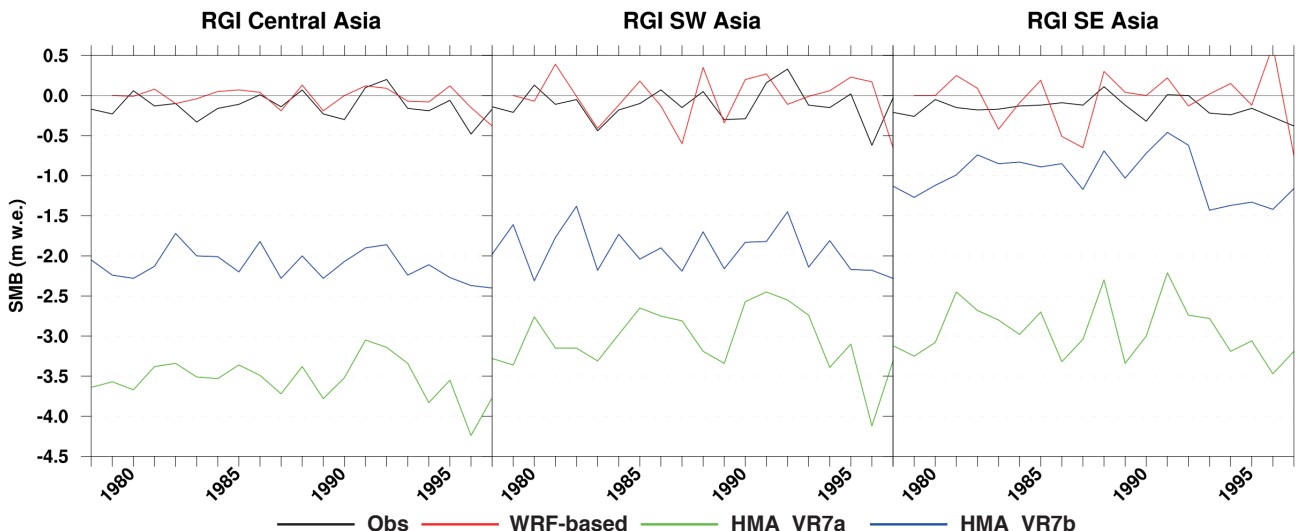

**Figure 11.** Area-averaged annual SMB (m w.e. yr$^{-1}$) over the period 1979–1998 derived from geodetical and glaciological observations (Zemp et al., 2019) (Obs, black), gridded outputs from a glacier model forced by WRF output (de Kok et al., 2020) (WRF-based, red), and simulation outputs of HMA_VR7a (green) and HMA_VR7b (blue). The area-averaged annual SMB is calculated for three different regions: RGI region Central Asia, which includes the HMA subregions NW-HMA and NE-HMA (Figure 1), and RGI regions SW Asia and SE Asia, which represent HMA subregions SW-HMA and SE-HMA, respectively (Figure 1).

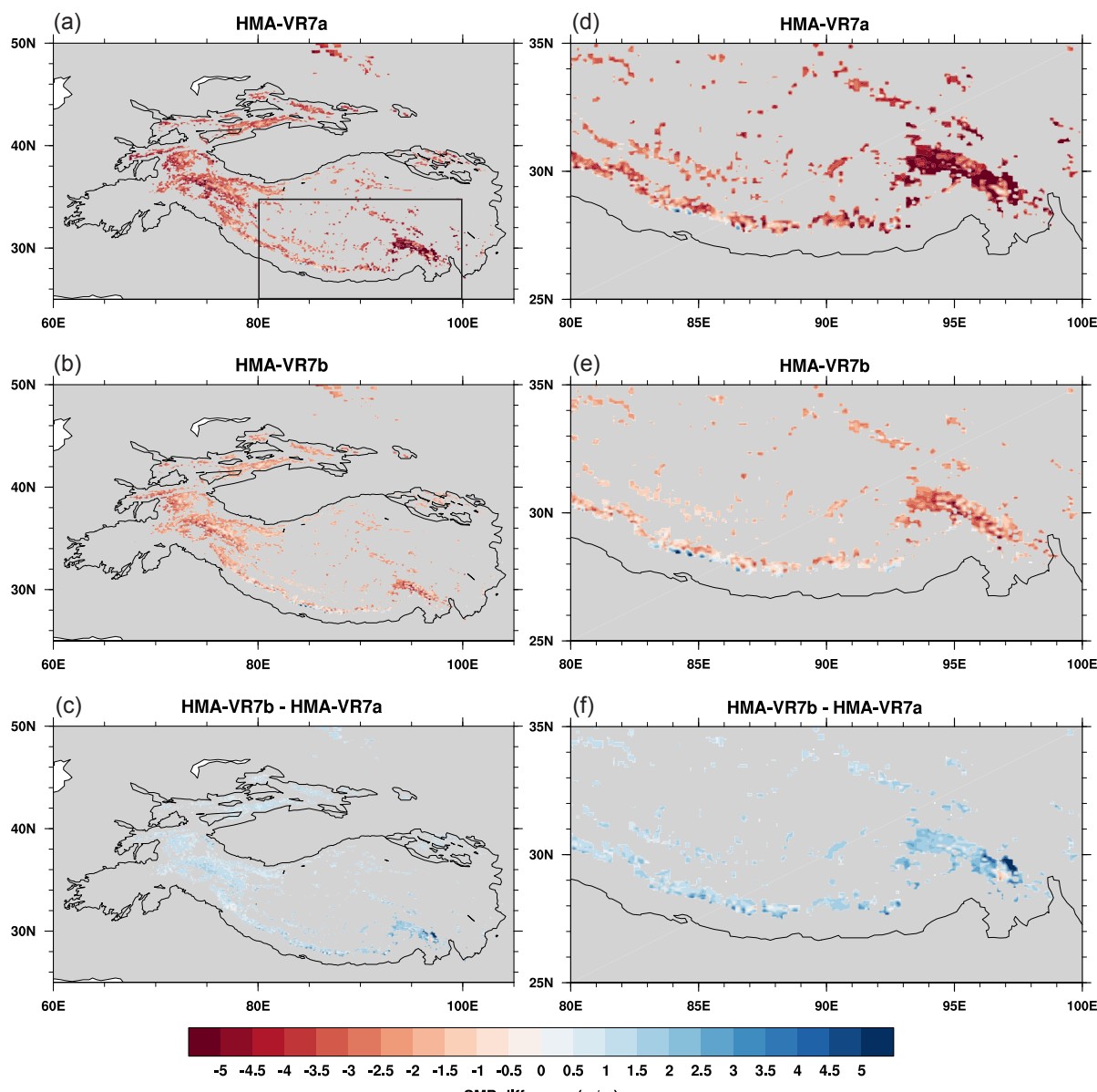

**Figure 12.** Spatial distribution of mean annual SMB (m w.e. yr⁻¹) over the period 1979–1998 as simulated by HMA_VR7a (**a-b**) and HMA-VR7b (**c-d**), and the SMB differences between the two simulations (**e-f**). The black box in Figure 12a denote the area (25°–35°N; 80°–100°E) of the insets over southeastern HMA (**d, e, f**). The HMA outline is derived from the Global Mountain Biodiversity Assessment (GMBA) Mountain Inventory version 1.2 (Körner et al., 2017).

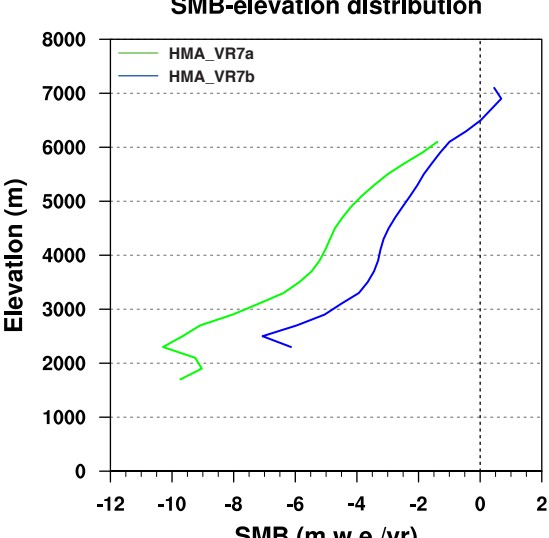

**Figure 13.** Mean elevation profile of glacier SMB (m w.e. yr[-1]) over HMA for the period 1979–1998. The green and blue lines denote the SMB-elevation profiles for HMA_VR7a and HMA_VR7b, respectively.

**Table 4.** Mean integrated SMB mass fluxes (Gt yr[-1]) for the period 1979–1998 in gigatons per year. The numbers in brackets denote the standard deviation in time. The integrated SMB mass fluxes have been calculated over two different areas of integration for HMA_VR7a, derived from the original and updated glacier-cover datasets, respectively. The SMB mass fluxes for HMA_VR7a that are integrated over the glacier areas of the updated glacier-cover dataset are denoted as HMA_VR7a_GC2. The mass fluxes of HMA_VR7b are only integrated over the glacier areas of the updated dataset.

| Simulation | Glacier Area (km$^2$) | Precipitation (Gt yr[-1]) | Ice Melt (Gt yr[-1]) | Total Melt (Gt yr[-1]) | Refreezing (Gt yr[-1]) | Runoff (Gt yr[-1]) | Sublimation/ Evaporation (Gt yr[-1]) | SMB (Gt yr[-1]) |
|---|---|---|---|---|---|---|---|---|
| HMA_VR7a | 120,087 | 111 (6) | 463 (33) | 553 (29) | 34 (2) | 554 (33) | 15 (1) | -459 (33) |
| HMA_VR7a_GC2 | 96,493 | 90 (5) | 356 (26) | 432 (23) | 29 (2) | 430 (26) | 12 (0.6) | -352 (27) |
| HMA_VR7b | 96,493 | 93 (4) | 228 (20) | 324 (18) | 32 (2) | 306 (21) | 12 (0.5) | -224 (21) |

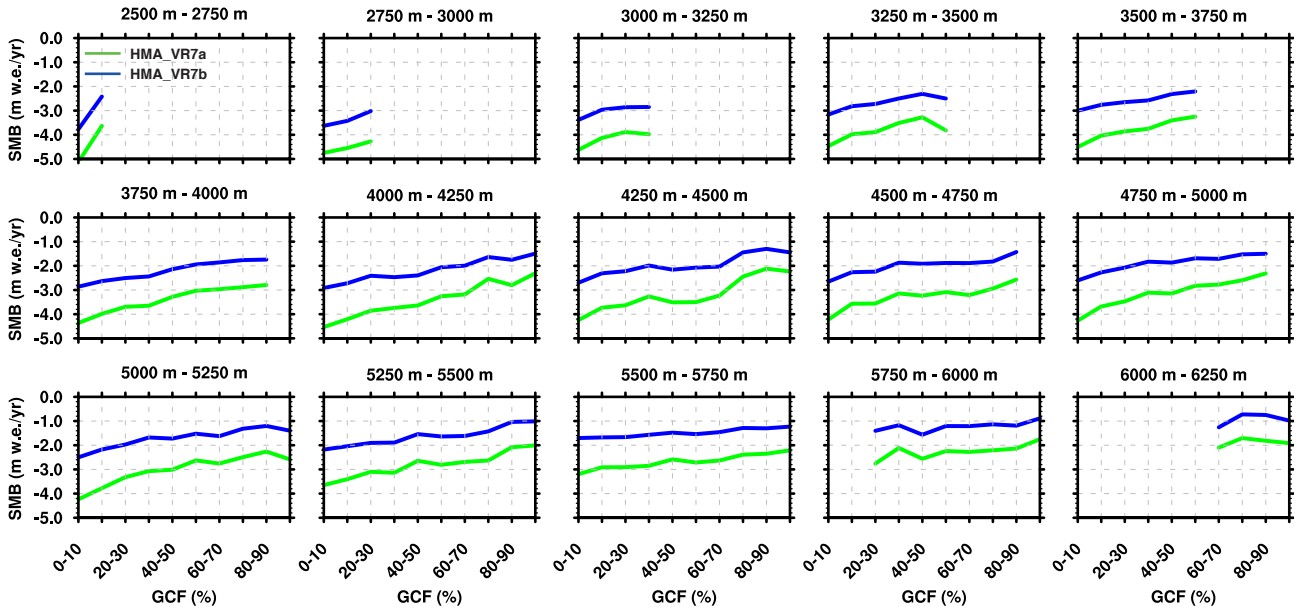

**Figure 14.** CLM grid-cell-mean SMB-glacier fraction distributions for HMA_VR7a (green) and HMA_VR7b (blue). The SMB–glacier elevation distributions are calculated for 15 different 250m elevation zones between 2500 m and 6250 m altitude, where elevation zones are based on CLM grid-cell-mean elevation distributions.