# Peer review of "Exploring the ability of the variable-resolution CESM to simulate cryospheric-hydrological variables in High Mountain Asia"

_The Cryosphere, 2022_

## Author Comment (AC1)

We gratefully acknowledge the reviewer for his/her remarks and suggestions, which improved the quality of the manuscript significantly. We have carefully considered the suggestions of the reviewer and we provide a point-by-point response to the reviewer's comments. For clarity, the reviewer's comments are given in bold italics and the responses are given in plain text. References that do not refer to those in the main manuscript are listed below. The manuscript will be modified according to the responses that are given to the comments.

*The authors evaluate the performance of a variable-resolution (VR) configuration of the Community Earth System version 2 over the High Mountain Asia (HMA) region, focusing on the cryospheric-hydrological variables. A new VR mesh is produced for this study with its grid spacing refined to ~ 7 km over the HMA region from ~ 1° in the coarse-resolution domain. A new glacier-cover input data is also produced for the VR grid.*

*The performance of VR configuration is compared to a globally uniform ~1° grid (NE30) through 20-year long simulations (1979-1998) and also evaluated against a variety of observational dataset. While the regionally refined mesh improves some aspects of the simulation quality, such as the circulation patterns forced by the topography, other aspects are degraded from the NE30 simulation. One reason for the degradation is model sensitivity to spatial resolution and time step length, i.e., less optimal tuning. Another VR simulation with several tuning to alleviate such sensitivities showed improved performance, but model biases still remain. They also suggest several future directions to reduce the VR model bias and improve physical representations in both the atmosphere and land models, as well as their coupling method.*

*The study addresses questions relevant to the scope of TC. The target region (HMA) is an important natural resource for the population in the broad Asia region. The model they evaluate (CESM) is a widely used community model. The grid resolution within the regional refinement is higher than a previous study focusing on the same region using CESM, and the new glacier input data is a original product from this study. The text is well written, and figures are overall high quality, altough I have several minor suggestions. All together, I think this study can be an important contribution to the community. However, I have one major concern about their atmospheric model configuration and one suggestion of additional experiment that could strengthen the scientific quality and impact of the manuscript. Please consider the following major comments before publication.*

We thank the reviewer for his/her evaluation of the manuscript. We have tried to address all concerns.

*Major comments*

*I believe the spectral element dynamical core used in this study is the hydrostatic version, which is not expected to be appropriate for the 7-km grid spacing. Several previous studies found that hydrostatic and non-hydrostatic schemes produce significantly different solutions in sub-10km grid spacings, or even larger gridcell sizes over steep topography like HMA (Wedi and Smolarkiewicz 2009, Prein et al., 2015, Yang et al., 2017). Errors would appear in vertical acceleration or unphysical propagations of gravity waves, which certainly affect moist physics behavior over HMA. No assessment of those aspects were provided in the manuscript.*

*Another related concern is the vertical resolution. The small horizontal grid spacing is probably not balanced by the rather coarse 32 vertical levels (Lindzen and Fox-Rabinovitz 1989, Skamarock et al., 2019). Any testing was done with different vertical levels?*

*Because I believe that the model is used with the resolution outside of its intended use, I strongly suggest the authors providing justifications and caveats to the readers about their results. The atmospheric model configuration used here should not be recommended as the standard for future modeling studies on the 7km VR grid.*

Regarding the appropriateness of using a 7 km regional refined grid in combination with the hydrostatic VR-CESM, we understand the concerns of the reviewer. We'd like to orient this discussion by beginning with the scale analysis for non-hydrostatic dynamics invoked by Wedi and Smolarkiewicz (2009) that the reviewer has cited. Referring to their Figure 4, they state that "the results are consistent with estimates typically obtained from a heuristic scale analysis of non-hydrostatic motions in NWP, i.e. horizontal scales L = O(10 km) resolved with grid intervals dx = O(2 km)." We think it's fair to equate this scale analysis as the null hypothesis; that non-hydrostatic terms are only important at length scales below O(10km). In our 7km simulations, the smallest resolved scales are between 4 to 6 times the grid spacing (Skamarock et al., 2014; Lauritzen et al., 2018), or between 28 km and 42 km, which is greater than 10 km.

Testing this hypothesis requires experiments in which the dynamical core has a hydrostatic and non-hydrostatic "switch," and with pairs of tests performed across a range of grid spacings spanning the dx=2km transition. As discussed in Liu et al. (2022), models that support both hydrostatic and non-hydrostatic dynamics may contain design differences in addition to just the non-hydrostatic terms. For example, non-hydrostatic dynamics have one more prognostic variable, and therefore one more degree

of freedom for applying the explicit damping operators, and how those operators behave in regions of steep topography matters a great deal (as does the vertical coordinate). This is not to discount the results cited in, e.g., Yang et al. (2017), but rather rejecting the null hypothesis in a single study may not be sufficient for the atmospheric modelling community as a whole to reject the null hypothesis, due to the challenges in testing this hypothesis.

Liu et al. (2022) provides a nice synthesis of studies that have cast doubt on the null hypothesis. It seems that prior to the Yang et al. (2017) study, studies showed conflicting results regarding the importance of non-hydrostatic dynamics at grid spacings between 2km and 30km. The recent studies of Yang et al. (2017) and Liu et al. (2022) do, however, provide convincing evidence to reject the null hypothesis, that non-hydrostatic dynamics are important at grid spacings up to 25 km, particularly in its representation of tropical convective systems.

But we believe that the reviewer's statement that "the atmospheric model configuration used here should not be recommended … for future modelling studies …" is too definitive given the challenges of testing this hypothesis and the relatively few studies that definitively reject the null hypothesis. However, we agree with the reviewer that this crucial discussion on the controversy of using a hydrostatic model at high resolution is missing. We will add the following discussion in the revised manuscript (under section *2.2 HMA-VR Grid and Performance*):

"The spectral-element dynamical core used by CESM is currently based on the hydrostatic approximation; non-hydrostatic vertical acceleration terms are neglected, which are important for the representation of deep convection, gravity waves, and flow over topography (Jeevanjee, 2017; Liu et al., 2022). Conventionally, the horizontal scales at which non-hydrostatic terms become important are assumed to be O(10 km), the vertical scale of the troposphere (e.g., Wedi and Smolarkiewicz, 2009). In our 7 km simulations, the smallest resolved scales are between 4 to 6 times the grid spacing (Skamarock et al., 2014; Lauritzen et al., 2018), or between 28 km and 42 km, which suggests that the hydrostatic approximation is appropriate. However, the more recently published studies of Yang et al. (2017) and Liu et al. (2022) have shown that non-hydrostatic terms are important at grid spacings up to 25 km, particularly in its representation of tropical convective systems. Due to the inherent difficulty in testing the null hypothesis (Liu et al., 2022), and that only two studies the authors are aware of have categorically rejected the null hypothesis, we do not believe this is grounds to conclude whether it is appropriate or not to use a 7 km regionally refined grid in combination with a hydrostatic model. Nonetheless, simulations with the hydrostatic VR-CESM in combination with a 7 km regionally refined grid have been performed before. Rhoades et al. (2018) have successfully shown the ability of VR-CESM to simulate mountainous climate with a 7 km regionally refined grid over the mountain ranges of western USA. Due to its location in the mid latitudes, the hydrostatic simulations with a 7 km grid spacing are still considered to be appropriate since the differences between hydrostatic and non-hydrostatic dynamics, as suggested by (Yang et al., 2017) mainly occur in the Tropics. Considering that High Mountain Asia is at about the same latitudes as the western US mountain ranges, we assume it is appropriate to apply a 7 km grid spacing over HMA in combination with a hydrostatic model."

Also, we would like to note that simulations with the hydrostatic VR-CESM in combination with a 7km regionally refined grid have been performed before. Rhoades et al. (2018) have successfully shown the ability of VR-CESM to simulate mountainous climate with a 7km regionally refined grid over the mountain ranges of western USA. Due to its location in the mid latitudes, the hydrostatic simulations with a 7km grid spacing are still considered to be appropriate since the differences between non-hydrostatic and hydrostatic dynamics, as suggested by Yang et al. (2017), mainly occur in the tropics. Considering that High Mountain Asia is at about the same latitudes as the western US mountain ranges, we think it is appropriate to apply a 7km grid spacing over HMA in combination with a hydrostatic model.

Regarding the vertical resolution, the 32 vertical level scheme was the default scheme of the Community Atmosphere Model (CAM) when the simulations were performed. Recent developmental versions of CESM also have the option to run with 58 vertical levels by default. The main reason of not performing the simulations with a higher number of vertical levels is the computational cost that increases significantly with the number of vertical levels. However, we understand the concerns of the reviewer. We expect that future simulations with the HMA_VR7 grid will also be performed with 58 vertical levels.

***2) First, I appreciate the authors for not only identifying model biases but also suggesting certain model components/characteristics that contribute to the biases, e.g., tropospheric warming due to stronger vertical motion and additional heating from CLUBB at higher resolution and with a shorter time step. But some of bias attribution remain speculative/qualitative or too general (section 3.7), which I feel is limiting the impact of the manuscript.***

***Noting that enough materials are presented in the current draft, I'd still like to suggest conducting off-line CLM5 simulations on the same two grids, NE30 and HMA_VR, forced by observed meteorology. Off-line land simulations are much cheaper than the coupled atmosphere-land simulation and would help partition model biases in the surface/near surface variables, especially SEB and SMB, into those from the land model alone or those from erroneous forcing from the atmosphere (or from problems arising from coupling or feedback). It is also worth asking if finer grid resolution and/or the new glacier data improve the off-line CLM5 performance. Having more accurate knowledge of CLM5 performance will help strengthen the logical basis for the discussion in section 3.7.***

References

Lindzen, R. S., & Fox-Rabinovitz, M. (1989). *Consistent vertical and horizontal resolution. Monthly Weather Review, 117,* 2575–2583.

Lu, J., Chen, G., Leung, L. R., Burrows, D. A., Yang, Q., Sakaguchi, K., & Hagos, S. (2015). *Toward the Dynamical Convergence on the Jet Stream in Aquaplanet AGCMs. Journal of Climate, 28(17),* 6763–6782. https://doi.org/10.1175/JCLI-D-14-00761.1

Pope, V. D., & Stratton, R. A. (2002). *The processes governing horizontal resolution sensitivity in a climate model. Climate Dynamics, 19(3–4),* 211–236. https://doi.org/10.1007/s00382-001-0222-8

Prein, A. F., Langhans, W., Fosser, G., Ferrone, A., Ban, N., Goergen, K., et al. (2015). *A review on regional convection-permitting climate modeling: Demonstrations, prospects, and challenges. Reviews of Geophysics, 53,* 1–39. https://doi.org/10.1002/2014RG000475.Received

Rauscher, S. a., & Ringler, T. D. (2014). *Impact of variable-resolution meshes on midlatitude baroclinic eddies using CAM-MPAS-A. Monthly Weather Review, 142(11),* 4256–4268. https://doi.org/10.1175/MWR-D-13-00366.1

Skamarock, W. C., Snyder, C., Klemp, J. B., & Park, S.-H. (2019). *Vertical Resolution Requirements in Atmospheric Simulation. Monthly Weather Review, 147(7),* 2641–2656. https://doi.org/10.1175/mwr-d-19-0043.1

Wedi, N. P., & Smolarkiewicz, P. K. (2009). *A framework for testing global non-hydrostatic models. Quarterly Journal of the Royal Meteorological Society, 135,* 469–484. https://doi.org/10.1002/qj.377

Yang, Q., Leung, L. R., Lu, J., Lin, Y. L., Hagos, S., Sakaguchi, K., & Gao, Y. (2017). *Exploring the effects of a nonhydrostatic dynamical core in high-resolution aquaplanet simulations. Journal of Geophysical Research: Atmospheres, 122(6),* 3245–3265. https://doi.org/10.1002/2016JD025287

Thank you for your valuable suggestions. Based on your suggestions, we performed CLM-offline simulations with the HMA_VR grid (following the HMA_VR7a and HMA_VR7b settings, referred as HMALOa and HMALOb, respectively), although the period is relatively short (3 years; 1979–1981). The relatively high computational cost of an HMALO simulation (~12,000 core hours per simulated year) limited the amount of years we could run. The HMALO simulations are driven by observation-based meteorological forcings from the Global Soil Wetness Project (GSWP), one of the default offline modes in CLM. As for the NE30 grid, we choose not to run offline CLM simulations. As we will explain later, the NE30 simulations we used for evaluation do not compute SMB over the HMA region, but only over the Greenland and Antarctic glacier regions. Therefore, we think (from the SMB perspective) that an offline CLM simulation with the NE30 grid will not have an additional value until NE30 simulations have been performed that also compute SMB over HMA.

The HMALO simulations show an increased SMB bias relative to the equivalent HMA VR simulations (Table RC1.1). The larger SMB bias (relative to HMA_VR7) can be attributed to the warmer surface climate in the HMALO simulations (Figure RC1.1), which results in less refreezing and more melt than in the HMA VR simulations. Also, precipitation volumes in HMALO simulations (46 Gt yr$^{-1}$) are smaller than the precipitation volumes passed by CAM in the HMA VR simulations (90-93 Gt yr$^{-1}$), which translates into less snowfall and less accumulation (Figure RC1.2). Compared to the observation-based WFDEI, the temperature and precipitation biases are small. However, it should be mentioned that observation-based datasets, such as WFDEI and GSWP, rely on the availability of meteorological observations, which are often poorly distributed over HMA, especially at higher altitudes. Therefore, it could result in temperature overestimates and precipitation estimates (Palazzi et al., 2015; Immerzeel et al., 2015; Gu et al., 2012; Lalande et al., 2021), which can (partly) explain the cold and wet biases that are visible in the HMA VR simulations. From a few perspectives, the HMALO simulations can still help to understand the performance of CLM/CESM over HMA better. First, the colder surface temperatures and higher precipitation volumes in the HMA VR simulations help to reduce to the SMB bias. Second, as Figure RC1.3 shows, the SMB in HMALO simulations also shows a declining trend with decreasing glacier fraction, which confirms the necessity to improve land-atmosphere coupling. We will include the presented tables and figures in the manuscript or in the supplementary information, and highlight the key points mentioned above in Section 3.7.

**Table RC1.1.** Mean integrated SMB mass fluxes (Gt yr⁻¹) for the period 1979–1998 (1979–1981 for HMALO simulations) in gigatons per year. The numbers in brackets denote the standard deviation in time. The integrated SMB mass fluxes have been calculated over two different areas of integration for HMA_VR7a, derived from the original and updated glacier-cover datasets, respectively. The SMB mass fluxes for HMA_VR7a (HMALOa) that are integrated over the glacier areas of the updated glacier-cover dataset are denoted as HMA_VR7a_GC2 (HMALOa_GC2). The mass fluxes of HMA_VR7b (HMALOb) are only integrated over the glacier areas of the updated dataset.

| Simulation | Glacier Area (km²) | Precipitation (Gt yr⁻¹) | Ice Melt (Gt yr⁻¹) | Total Melt (Gt yr⁻¹) | Refreezing (Gt yr⁻¹) | Runoff (Gt yr⁻¹) | Sublimation/ Evaporation (Gt yr⁻¹) | SMB (Gt yr⁻¹) |
|---|---|---|---|---|---|---|---|---|
| HMA_VR7a | 120,087 | 111 (6) | 463 (33) | 553 (29) | 34 (2) | 554 (33) | 15 (1) | -459 (33) |
| HMA_VR7a_GC2 | 96,493 | 90 (5) | 356 (26) | 432 (23) | 29 (2) | 430 (26) | 12 (0.6) | -352 (27) |
| HMA_VR7b | 96,493 | 93 (4) | 228 (20) | 324 (18) | 32 (2) | 306 (21) | 12 (0.5) | -224 (21) |
| HMALOa_GC2 | 96,493 | 47 (2) | 463 (14) | 501 (9) | 13 (2) | 492 (12) | 19 (1.0) | -464 (11) |
| HMALOb | 96,493 | 46 (2) | 333 (19) | 385 (8) | 16 (5) | 367 (12) | 19 (0.9) | -339 (11) |

[Figure]

**Figure RC1.1.** Boxplots of 2-m temperature differences (°C) between the simulation outputs of HMALOa (orange), HMALOb (yellow), HMA_VR7a (green), HMA_VR7b (blue), and the observation/reanalysis-based WFDEI for each season and HMA subregion (shown in Figure 1b). The box represents the biases between the 25th and 75th percentile, the line in the box denotes the median, and the whiskers represent the 10th and 90th percentile of temperature differences.

[Figure]

**Figure RC1.2.** Same as Figure RC1.1, but for rainfall (mm month⁻¹) (**a**) and snowfall (mm month⁻¹) (**b**)

[Figure]

**Figure RC1.3.** CLM grid-cell-mean SMB-glacier fraction distributions for HMALOa (orange), HMALOb (yellow) HMA_VR7a (green), and HMA_VR7b (blue). The SMB–glacier elevation distributions are calculated for 15 different 250m elevation zones between 2500 m and 6250 m altitude, where elevation zones are based on CLM grid-cell-mean elevation distributions.

*Minor comments*

*L98*
*"In the western USA and the Chilean Andes, it has been used with regional refinements up to 7 km to simulate regional climate and snowpack (Huang et al., 2016; Rhoades et al., 2016; Bambach et al., 2021; Xu et al., 2021).Also, over the Tibetan Plateau, South Asia, and East Asia, it has been applied to study the regional climate and snow characteristics (Rahimi et al., 2019; Xu et al., 2021). The application of VR-CESM to simulate glacier SMB has been limited, this far, to the Greenland Ice Sheet (van Kampenhout et al., 2019;Herrington et al., 2022)."*

*I do not find any of the cited studies carrying out 7-km resolution refinement with the CESM-SE or a comparable model; the finest grid spacing seems to be ~ 12km.*

We will add a citation to the manuscript that refers to a study carrying out 7-km resolution refinement with the CESM-SE, which is from Rhoades et al. (2018). This study investigated the sensitivity of mountain hydroclimate simulations in VR-CESM to horizontal resolution and microphysics by applying 7-km horizontal refinement, amongst others, over the Sierra Nevada Mountain range in western US.

*L127~*
*Deep convection parameterization is not mentioned. Was it turned off for the simulations in this study?*

Deep convection in CESM2 is parameterized by the Zhang-McFarlane (ZM) convection scheme (Zhang and McFarlane, 1995) and is by default turned on. We will add an extra sentence to the Data and Methods section to cover this information.

*L227*
*"including a new glacier region over HMA with a 36-EC scheme in CLM. The new glacier region makes it possible to simulate SMB in multiple (including virtual) ECs in HMA, while retaining the computationally cheaper default behavior of one EC per grid cell in other mountain glacier regions. "*
*To clarify, in the default CESM/CLM, can a user set different number of ECs for each grid column? Or is this region-dependent EC numbers is a special configuration prepared for this study?*

In the default CLM, the world's glaciers and ice sheets are broken down in several glacier regions covering Greenland, the peripheral regions of Greenland, Antarctica, and all others. In this study, we added another glacier region covering High Mountain Asia. For each specific region it is possible to set the behavior of melt and runoff, and the use of ECs, which are needed to specify the use of elevation downscaling and computation of SMB. By default, 10 ECs are used over the Greenland (and peripheral) and Antarctic glacier regions, and a single EC is applied over the other mountain glacier regions. It is, however, possible to use another EC scheme within CLM, varying from 3 ECs to 36 ECs. Nonetheless the number of ECs that is chosen is fixed for all glacier regions where the use of ECs is set to multiple/virtual, which means it is not possible to set different number of ECs for each grid column. In this light, the 36 ECs we chose for this study are not only applied over HMA, but also over the Greenland and Antarctic glacier regions. We will add an extra sentence to the manuscript clarifying that the 36 ECs in this study are also applied over the Greenland and Antarctic glacier regions.

*Why the difference in spin-up procedure for each VR run? I suppose spin-up procedure affect the model bias against observations, especially those of cryospheric-hydrological variables. So signals from the spin-up difference are likely to be mixed with those from the configuration/resolution differences. The authors can look at model biases of the HMA_VR7b spin-up run and compare them to those after spin-up to get some ideas of the spin-up impact on the analyzed variables.*

The differences in spin-up procedures for each VR run can be explained as follows. For the first simulation (HMA_VR7a), CLM was forced with a land climatology originating from a NE120 run, which happens by default. Due to computational constraints at that time, we chose for a spin-up period of 1 year, which was assumed to be sufficiently long to equilibrate the atmosphere and land models. Following a spin-up period of 1 year, we applied a maximum allowed snow depth of 1 m w.e. (default CESM1), since a 1-year period was not assumed to be sufficiently long to reach a maximum allowed snow depth of 10 m w.e. (default CESM2).

For the second simulation, we choose to initialize the snow depth over glacierized land units to 2.5 w.e. To equilibrate the atmosphere and land model, and in particular the snowpack, a longer spin-up period was required. Since the HMA grid is expensive in computational cost, we could not afford long spin-up runs with VR-CESM. Instead, we decided to split-up the spin-up procedure into two parts: 1) an AMIP-style spin-up run of 10 years with CAM, and 2) an offline CLM spin-up run of 50 years. Offline CLM runs are computationally cheaper and therefore allow to run the model for a longer period. To initialize CLM, the coupler output of the CAM spin-up run was used, which was sub cycled over a 10-year period for 50 years. We found that a period of 50 years is sufficiently long to equilibrate the snowpack in the model (Figure RC1.4).

[Figure]

**Figure RC1.4.** Area-averaged annual snow thickness (mm w.e.) over the period 1981-2030 derived from the simulation outputs of CLM spin-up of HMA_VR7b. The area-averaged snow thickness is calculated for the entire HMA region.

To give an idea of the impact of the spin-up on analyzed variables, such as snow depth, Figure RC1.5 shows the snow depth differences between the CAM spin-up, the last sub cycle of the CLM spin-up (2021-2030), the transient HMA_VR7b run, and the JRA55 dataset. In most regions and throughout most seasons, the snow depth bias relative to JRA55 reduces between the CAM spin-up and the transient HMA VR run, where the largest changes occur during spring over SW-HMA (i.e., ~45 mm w.e. difference between the median snow depth biases of the CAM spin-up and the transient HMA VR run). This means that the spin-up procedure has a positive impact on the snow depth in the model. When we look at the changes in snow depth biases resulting from the configuration differences (HMA_VR7a vs HMA_VR7b; Figure 9), we see an opposite trend with increasing snow depth biases between HMA_VR7a and HMA_VR7b. Therefore, we think it is unlikely that the signals of the different spin-up procedures and configurations are mixed.

[Figure]

**Figure RC1.5.** Boxplots of snow depth differences (mm w.e.) between the simulation outputs of the CAM spinup (purple), the last sub cycle of the CLM spinup (2021–2030; cyan), HMA_VR7b (blue), and JRA-55 for each season and HMA subregion (shown in Figure 1b). The box represents the biases between the 25th and 75th percentile, the line in the box denotes the median, and the whiskers represent the 10th and 90th percentile of snow depth differences.

*L344~*

*"The monsoonal circulation in the NE30 run has two centers, a broad region of ascent in the southern HMA region, primarily over the Indian Ocean, and a narrower region of ascent over the front range of the Himalayas."*
*Please specify which sub-figures or rows are being discussed (e.g., "Figure 6, second row") to help readers follow the text. Maybe it's useful to add indices (a,b,c,...) to each row.*

The text cited by the reviewer refers to second row and first column of Figure 6 or Figure 6d. We will add indices to each panel of Figure 6 to specify which sub-figures are discussed.

*L353-356*
*"While the warming and drying patterns are largely the result of greater vertical velocities due to the enhanced spatial resolution in the HMA VR runs, the shorter physics timestep also contributes to this warming and drying (not shown), which is a common response to reducing the physics timestep (Williamson, 2008; Herrington et al., 2022)."*

*Williamson (2008) does not specifically discuss warming and drying as seen in this work. He illustrated model sensitivities to both the timestep and spatial resolution.*

*According to Herrington et al. (2022), shorter time step contributes only to the warming of the lower troposphere. Pope and Straten (2002) found a similar warming of the mid-latitude troposphere because of eddy flux and its convergence are enhanced with higher resolution. Although largescale feature, mid-latitude waves simulated at different resolutions converge only at 50 km or finer grid spacing according to Lu et al.(2015). An enhanced mid-latitude eddies inside regional refinement is reported by Rauscher and Ringler (2014) in their VR simulations as well, so the same processes may be occuring in the simulations here.*

Figure RC1.6 shows latitude-height transects averaged over the longitude band 80˚-100˚ (same as Figure 6) for NE30 and NE30* differences relative to NE30, where NE30* represents NE30 simulations with a (shorter) physics time step of 450s. The broad warming originating on the north side of the domain (Figures RC1.6h), and the drying on the south-side of the domain (Figures RC1.6j), is evident from the figures. We agree with the reviewer that the cited studies do not specifically show a drying with reduced physics timestep. Therefore, we will remove the latter part of the sentence (L353-356) "which is a common response to reducing the physics timestep (Williamson, 2008; Herrington et al., 2022)" from the revised manuscript.

[Figure]

**Figure RC1.6.** Same as Figure 6 but comparing NE30 (control) against NE30 with a physics timestep of 450 s (NE30*). (Left column) NE30, (right column) NE30* differences relative to NE30.

*Section 3.3 & 3.4*
*Please clarify what "absolute monthly mean xxxx (e.g., precipitation) differences" means. Precipitation, snow cover, and snow depth are positive quantities, so no need to use their absolute values.*

Our aim was to make a distinction between "absolute" and "relative" changes, where "absolute" change is expressed in terms of a quantitative change and a "relative" change is expressed in terms of a percental change. To avoid further confusion, we will remove "absolute" from the manuscript text and figures' captions.

*Section 3.6*
*Why is the NE30 result not included in this section?*

The globally uniform NE30 simulations we used in this study do not compute SMB over the HMA region, but only over the Greenland and Antarctic glacier regions. The SMB is not computed over the HMA region since the HMA region was previously part of all other mountain glacier regions, where, by default, the glacier melt water remains in place until it refreezes, which means ice melt does not result in runoff and SMB is not computed. To enable the computation of SMB over HMA, we added a new glacier region and applied a different glacier behavior setting that allows glacier melt water to runoff.

*L517, typo*
*"Figure 15 shows the relation between grid-cell-mean SMB and glacier fraction (GCF)"*
*->"Figure 14 shows..."*

We have changed "Figure 15 shows…" to "Figure 14 shows…".

*Figure 8*
*The minimum and maximum biases are not good to be used as whiskers for this figure because they push the y-axis limits so wide that we cannot see the differences in the quantile boxes. It's probably better to use 95th percentile, 3 standard deviations, or any other quantities that only moderately widens the y-axis range compared to the quantile range.*

We agree with the reviewer that the minimum and maximum biases are not good to be used as whiskers. Therefore, to improve Figure 8 and other figures showing boxplots (Figures 7 and 9), we will replace the minimum and maximum values by the $10^{th}$ and $90^{th}$ percentiles, respectively. Using these percentiles narrows down the y-axis, which improves the visibility of the quantile boxes.

*typo in the caption*
*"for rainfall (mm month-1) (a-b) and snowfall (mm month-1) (c-d)"*
*should be*
*"for rainfall (mm month-1) (a,c) and snowfall (mm month-1) (b,d)"*

We have changed "for rainfall (mm month-1) (a-b) and snowfall (mm month-1) (c-d)" to "for rainfall (mm month-1) (a-c) and snowfall (mm month-1) (b-d)"

*Figure 11, typo in the caption*
*Obs, blue*
*-> Obs, black*

We have changed "Obs, blue" to "Obs, black"

*Figure 12, typo in the caption*
*"The black box in Figure 14a denote"*
*-> "The black box in Figure 12a denotes"*

We have changed "The black box in Figure 14a denote…" to "The black box in Figure 12a denote…"

*Code and Data Availability.*
*"Data will be available before publication in Zenodo." What data are you referring to? Will the codes that produced the input data be publicly available?*

Publicly available data will be stored in two separate data archives on Zenodo. The first archive (https://doi.org/10.5281/zenodo.7864689) will contain the model scripts and files that were used to create the updated glacier cover dataset. The second archive (https://doi.org/10.5281/zenodo.7864633) will contain the NE30 and HMA_VR7 grid variables that were used to generate most of the figures in this manuscript. The remainder of the data will be available on request. We will update the data availability statement accordingly.

*Supplement material*

***Section S1 describes the workflow to produce the new glacier data in great detail. A critical information missing is the codes and/or applications used. Please consider sharing those information as well. Without, it is difficult for other researchers to reproduce the data or apply the same procedure to other regions.***

We would like to refer to the previous response.

*Figure S1*
***Not sure how the redbox in the inset of Figure S1b represents the outline of HMA. Do the two insets cover the same area?***

The red box in both insets of Figure S1b denote the location of the glacier outlines that are shown in Figure S1b, whereas the black outline in the leftmost inset denotes the outline of High Mountain Asia. The leftmost inset shows the location of the red box within the outline of High Mountain Asia, where the rightmost inset zooms in to the region where the red box and corresponding glacier outlines are located. We realize that the visibility of the red box is in the leftmost inset is not very clear. Therefore, we will improve its visibility. Also, we will change the text in the Figure's caption as follows:

"The red box in the insets denote the location of the glacier outlines and the black outline represents the outline of High Mountain Asia, where the HMA outlines are retrieved from the Global Mountain Biodiversity Assessment (GMBA) Mountain Inventory version 1.2 (Körner et al., 2017)."

**References**

Gu, H., Wang, G., Yu, Z., and Mei, R.: Assessing future climate changes and extreme indicators in east and south Asia using the RegCM4 regional climate model, Clim Change, 114, https://doi.org/10.1007/s10584-012-0411-y, 2012.

Immerzeel, W. W., Wanders, N., Lutz, A. F., Shea, J. M., and Bierkens, M. F. P.: Reconciling high altitude precipitation with glacier mass balances and runoff, Hydrol Earth Syst Sci, 12, 4755–4784, https://doi.org/10.5194/hessd-12-4755-2015, 2015.

Jeevanjee, N.: Vertical Velocity in the Gray Zone, J Adv Model Earth Syst, 9, https://doi.org/10.1002/2017MS001059, 2017.

Körner, C., Jetz, W., Paulsen, J., Payne, D., Rudmann-Maurer, K., and M. Spehn, E.: A global inventory of mountains for bio-geographical applications, Alp Bot, 127, https://doi.org/10.1007/s00035-016-0182-6, 2017.

Lalande, M., Ménégoz, M., Krinner, G., Naegeli, K., and Wunderle, S.: Climate change in the High Mountain Asia in CMIP6, Earth System Dynamics, 12, https://doi.org/10.5194/esd-12-1061-2021, 2021.

Lauritzen, P. H., Nair, R. D., Herrington, A. R., Callaghan, P., Goldhaber, S., Dennis, J. M., Bacmeister, J. T., Eaton, B. E., Zarzycki, C. M., Taylor, M. A., Ullrich, P. A., Dubos, T., Gettelman, A., Neale, R. B., Dobbins, B., Reed, K. A., Hannay, C., Medeiros, B., Benedict, J. J., and Tribbia, J. J.: NCAR Release of CAM-SE in CESM2.0: A Reformulation of the Spectral Element Dynamical Core in Dry-Mass Vertical Coordinates With Comprehensive Treatment of Condensates and Energy, J Adv Model Earth Syst, 10, https://doi.org/10.1029/2017MS001257, 2018.

Liu, W., Ullrich, P. A., Guba, O., Caldwell, P. M., and Keen, N. D.: An Assessment of Nonhydrostatic and Hydrostatic Dynamical Cores at Seasonal Time Scales in the Energy Exascale Earth System Model (E3SM), J Adv Model Earth Syst, 14, https://doi.org/10.1029/2021MS002805, 2022.

Palazzi, E., Von Hardenberg, J., Terzago, S., and Provenzale, A.: Precipitation in the Karakoram-Himalaya: a CMIP5 view, Clim Dyn, 45, 21–45, https://doi.org/10.1007/s00382-014-2341-z, 2015.

Rhoades, A. M., Ullrich, P. A., Zarzycki, C. M., Johansen, H., Margulis, S. A., Morrison, H., Xu, Z., and Collins, W. D.: Sensitivity of Mountain Hydroclimate Simulations in Variable-Resolution CESM to Microphysics and Horizontal Resolution, J Adv Model Earth Syst, 10, 1357–1380, https://doi.org/10.1029/2018MS001326, 2018.

Skamarock, W. C., Park, S. H., Klemp, J. B., and Snyder, C.: Atmospheric kinetic energy spectra from global high-resolution nonhydrostatic simulations, J Atmos Sci, 71, https://doi.org/10.1175/JAS-D-14-0114.1, 2014.
Wedi, N. P. and Smolarkiewicz, P. K.: A framework for testing global non-hydrostatic models, Quarterly Journal of the Royal Meteorological Society, 135, https://doi.org/10.1002/qj.377, 2009.

Yang, Q., Leung, L. R., Lu, J., Lin, Y. L., Hagos, S., Sakaguchi, K., and Gao, Y.: Exploring the effects of a nonhydrostatic dynamical core in high-resolution aquaplanet simulations, J Geophys Res, 122, https://doi.org/10.1002/2016JD025287, 2017.

Zhang, G. J. and McFarlane, N. A.: Sensitivity of climate simulations to the parameterization of cumulus convection in the canadian climate centre general circulation model, Atmosphere - Ocean, 33, https://doi.org/10.1080/07055900.1995.9649539, 1995.

---

## Author Comment (AC2)

We gratefully acknowledge the reviewer for his/her remarks and suggestions, which improved the quality of the manuscript significantly. We have carefully considered the suggestions of the reviewer and we provide a point-by-point response to the reviewer's comments. For clarity, the reviewer's comments are given in bold italics and the responses are given in plain text. References that do not refer to those in the main manuscript are listed below. The manuscript will be modified according to the responses that are given to the comments.

*Summary*

*Wijngaard et al. in "Exploring the ability of the variable-resolution CESM to simulate cryospheric-hydrological variables in High Mountain Asia" evaluate a first-of-its-kind study on High Mountain Asia glaciers using the variable-resolution capabilities in the Community Earth System Model at 9 km horizontal refinement. The authors do an admirable job of comprehensibly evaluating all aspects of snow energy and mass balance (SEB and SMB) drivers of HMA mountain glaciers. Unfortunately, it is identified that over the 1979-1998 period HMA mountain glacier SMB loss in VR-CESM is 10x higher than expected (expected SMB is -20 GT/yr, VR-CESM SMB is, at best, -200 GT/yr), although not due to any fault of the authors. The authors provide several plausible bias sources and hypotheses to test in future VR-CESM studies.*

*Overall, I think the paper fits perfectly within the scope of The Cryosphere and could be, given more work, an extremely valuable contribution to the scientific community. The findings have both scientific and societal impact as mountain glacier modeling, particularly in Earth system models, is a recent model development across the community of models (requiring comprehensive historical evaluation studies such as this one) and glacial melt supplements water supplies and poses significant flood hazards (e.g., glacier lake outburst flood events) to billions of people that reside downstream of them.*

*I think there are still several major(ish) revisions that need to happen prior to this paper being accepted. Most of my suggested revisions are minor, however a few (if feasible) may require some time to address, and I wouldn't want the authors to be time pressured by a quick turnaround with a suggestion of minor revisions.*

We thank the reviewer for his/her evaluation of the manuscript. We have tried to address all concerns.

*Suggested Revisions*
*Line 14 – change "could help…" to "can help through targeted grid refinement"*

We have changed "could help…" to "can help through targeted grid refinement…".

*Line 18 – delete "to evaluate the outcomes"*

We have deleted "to evaluate the outcomes".

*Line 22 – add "but is still underestimated"*

We have added "but is still underestimated".

*Line 37 – change "differs per region" to "is regionally dependent"*

We have changed "differs per region" to "is regionally dependent".

*Line 41 – change "as a response to climate change" to "in response to climate change"*

We have changed "as a response to climate change" to "in response to climate change".

*Line 52-53 – consider citing - Li, D., Lu, X., Walling, D.E. et al. High Mountain Asia hydropower systems threatened by climate-driven landscape instability. Nat. Geosci. 15, 520–530 (2022). https://doi.org/10.1038/s41561-022-00953-y*

Thanks for the suggestion. We will include the citation of Li et al. (2022) in the manuscript.

*Line 60 – change "with or without" to "with"*

We have changed "with or without" to "with".

*Line 63 – change "high horizontal…" to "fine horizontal…across many glaciers"*

We have changed "high horizontal…for many glaciers" to "fine horizontal…across many glaciers".

*Line 71 – change "giving" to "providing"*

We have changed "giving to "providing".

*Line 79 and Line 99 – consider citing - Rhoades, A. M., Ullrich, P. A., Zarzycki, C. M., Johansen, H., Margulis, S. A., Morrison, H., et al. (2018). Sensitivity of mountain hydroclimate simulations in variable-resolution CESM to microphysics and horizontal resolution. Journal of Advances in Modeling Earth Systems, 10, 1357– 1380. https://doi.org/10.1029/2018MS001326*

Thanks for the valuable suggestion. We will include the citation of (Rhoades et al., 2018) in the manuscript.

*Line 94 – FWIW Rhoades et al. 2018 provides model throughput timing for a wider range of refinement regions (including a 7km refinement patch)*

Thank you for the information. It is interesting to see that for about the same number of processors, the HMA_VR7 grid can produce about 35% (~0.7 SYPD) of the number of simulated years that the CAL_VR7 grid can produce within one wall-clock day (1.96 SYPD). This is mainly related to the larger domain that is covered by the 7 km refined patch in the HMA_VR7 simulations.

*Line 102 – change "this far" to "thus far"*

We have changed "this far" to "thus far".

*Line 106 – why was 1979-1998 chosen rather than a period that encapsulates more of the satellite record?*

Most AMIP-style runs in CESM start by default in 1979 and cover the period 1979–1998, which is the reason we also followed this approach. However, we agree with the reviewer that a later period that encapsulates more of the satellite record is more beneficial and helpful for evaluating the model outcomes. We expect that future model simulations will also cover later period, enabling comparisons with satellite products, such as those from the Sentinel project.

*Line 152 – is a constant lapse rate and relative humidity with elevation an appropriate assumption? At the very least, is there a study to cite here?*

The constant relative humidity with elevation in CLM is needed to control the amount of water vapor with elevation, i.e., to prevent conservation errors in CESM. Both relative and specific humidity are computed in CAM and then passed to CLM. Here specific humidity is downscaled based on the assumption of constant relative humidity, downscaled air temperature and downscaled saturation water vapor pressure, where downscaled saturation water vapor pressure is scaled by a polynomial fit between air temperature and saturation vapor pressure (Flatau et al., 1992). Since the temperature generally decreases with elevation, the saturation vapor pressure and specific humidity also decrease with altitude. By keeping the relative humidity constant, the actual water vapor pressure is scaled down in a similar manner, which prevents the model from simulating a surplus or deficit in actual water vapor. Based on the idea that a constant relative humidity prevents conservation errors, we think it is an appropriate assumption. We will add a citation to the study of Lipscomb et al. (2013).

*Line 156 – cite the topography dataset developer manuscript/data archive.*

We will include a citation to a data archive on Zenodo (https://doi.org/10.5281/zenodo.7864689) to the manuscript.

*Equation 1 and 2 – can black/brown carbon deposition influence SEB and SMB in VRCESM? If so, is black/brown carbon deposition too high and helping to drive the SMB bias? Also, can surface melt pools and flow channels develop (influences albedo and, potentially, energy transport through glacier)?*

Thank you for raising these points. Our CESM2 simulations are forced with both prescribed CMIP6-based tropospheric and stratospheric aerosols that include mineral dust, sea salt, and black carbon (BC), amongst others. Aerosols can potentially influence SMB and SEB in various ways in CESM2. These include a warming of the atmosphere due to aerosol-radiation interactions and reductions in snow albedo due to darkening of the snow surface. The aerosol-radiation interactions in CESM are simulated by the Modal Aerosol Model with 4 modes (MAM4; Liu et al., 2016) combined with the Rapid Radiative Transfer Model for GCMs (RRTMG; (Iacono et al., 2008)). Aerosol-induced changes in snow albedo and radiative heating transfers in the snowpack are simulated by SNICAR (Flanner et al., 2007), the radiative transfer model coupled to CLM.

Aerosol-radiation interactions and effects of aerosol deposition, such as black carbon (BC) and dust deposition, on snow darkening in CESM have previously been investigated by Rahimi et al. (2019), amongst others. The authors investigated the effects of BC and dust deposition on the South Asian summer monsoon and snow darkening with a 14 km regionally refined grid in combination with CESM1.2. According to the authors, BC deposition mainly result in an intensification of summer monsoon rains via aerosol-radiation interactions. Also, BC deposition has a snow-darkening effect that leads to reductions in SWE, particularly over the western Tibetan Plateau (corresponding to SW-HMA in our study). The authors show, however, that the effect of BC deposition on snow darkening is weaker in the VR simulations than in the 1-degree simulations due to the lower amount of BC deposition that is simulated by the VR grid.

To have a general understanding of the BC concentrations simulated by the NE30 and HMA VR grids, we looked at the BC concentrations in the top snow layer, defined as the ratio between the mass of BC and the mass of snow in the top snow layer (the thickness of the top snow layer is 1–3 cm, depending on the snow depth and related amount of snow layers). Another commonly used method is to define the BC concentrations as the ratio between the total BC deposition from the atmosphere and the total precipitation (e.g., He et al., 2014). However, this method introduces uncertainties in the calculation of in-snow BC concentrations, which is mainly related to the use of total precipitation that can overestimate the amount of snowfall and BC removed by snow and is prone to biases relative to observed precipitation (He et al., 2014). Therefore, we did not choose for this method and decided to focus on the BC concentrations in the top snow layer.

Tables RC2.1 and RC2.2 show the spatial mean and standard deviation of seasonal BC concentrations in the top snow layer for NE30 and HMA_V7b simulation outputs, respectively. The highest BC concentrations are simulated during spring and summer, which is most likely caused by the aging of snow and reduced influx of fresh snow (due to higher temperatures). Spatially, the higher BC concentrations are simulated in SE-HMA, which is more prone to BC deposition originating from the Indo-Gangetic Plains (not shown). Looking at the differences between the simulated NE30 and HMA_VR7b BC concentrations, we observe that the NE30 grid simulates higher BC concentrations than the VR grid, which is also in line with the findings of Rahimi et al. (2019). Compared to observations, such as those listed in He et al. (2014) or Kang et al. (2020), the simulated BC concentrations in the top snow layer fall in range of what is observed over HMA. However, the range of observed values is large (i.e., between 16 ng/g and 3000 ng/g; Kang et al., 2020). Also, most observations are based on measurements conducted over a short timeframe (e.g., one season). Nonetheless, based on the available observations we think that the BC deposition is not too high, and that the impact of the BC deposition on the SMB bias is relatively small.

**Table RC2.1.** Mean seasonal BC concentrations (ng/g) in top snow layer as simulated by the NE30 grid in the four HMA subregions (Figure 1b). The numbers between brackets denote the standard deviation of the BC concentrations. The BC concentrations are derived for grid cells where glaciers are present, and the mass of snow is higher or equal to 0.3 kg/m$^2$ (i.e., equivalent to 1 mm of snow, assuming a snow density of 300 kg/m$^3$). The latter was necessary to filter out unrealistically high BC concentrations due to the limited amount of snow in the NE30 simulations during summer and autumn.

|  | NW-HMA | NE-HMA | SW-HMA | SE-HMA |
|---|---|---|---|---|
| **DJF** | 84 (50) | 51 (31) | 76 (45) | 109 (39) |
| **MAM** | 136 (89) | 75 (44) | 125 (96) | 188 (119)[1] |
| **JJA** | 162 (110) | 75 (117) | 386 (267) | 485 (434)[1] |
| **SON** | 54 (20) | 38 (24) | 128 (84)[1] | 90 (68) |

[1] To filter out unrealistically high BC concentrations, grid cells where the mass of BC is higher than the global mean (~0.03 kg/m$^2$) are filtered out.

**Table RC2.2.** Same as Table RC2.1 but for HMA_VR7b.

|  | NW-HMA | NE-HMA | SW-HMA | SE-HMA |
|---|---|---|---|---|
| **DJF** | 42 (26) | 56 (47) | 73 (64) | 144 (72) |
| **MAM** | 85 (48) | 74 (51) | 108 (76) | 208 (90) |
| **JJA** | 93 (101) | 32 (65) | 238 (241) | 184 (224) |
| **SON** | 62 (29) | 41 (31) | 97 (70) | 92 (46) |

As for the surface melt pools and flow channels, these cannot develop in CLM, since CLM does not take the complex topography of mountain glaciers into account. To sufficiently simulate surface melt pools, flow channels, and other important glacier features (e.g., crevasses or ice cliffs), models are required with a high horizontal resolution up to 1 m (e.g., Bonekamp et al., 2020).

*Line 195 – delete "more"*

We have deleted "more".

*Line 218-219 – could the authors add the observed max elevation for each HMA region so that readers can contrast with model representation max elevations?*

We will include the following table, which shows the observed maximum altitude for each HMA subregion:

**Table 1.** Maximum altitude per HMA subregion (Figure 1b) as observed for NE30, NE120, HMA_VR7 and GMTED2010.

|            | NW-HMA  | NE-HMA  | SW-HMA  | SE-HMA  |
|------------|---------|---------|---------|---------|
| **NE30**      | 5162 m  | 5170 m  | 4978 m  | 4452 m  |
| **NE120**     | 5684 m  | 5526 m  | 5513 m  | 5369 m  |
| **HMA_VR7**   | 6167 m  | 5778 m  | 5870 m  | 6228 m  |
| **GMTED2010** | 7325 m  | 7099 m  | 8190 m  | 8625 m  |

Also, we have changed the following sentence (L218-219):
"Over HMA, the maximum altitude increases from 5170 m and 5684 m for NE30 and NE120, respectively, to 6227 m for HMA_VR7."

into

"Over HMA, the maximum altitude increases from 5170 m and 5684 m for NE30 and NE120, respectively, to 6228 m for HMA_VR7. These maxima are observed in the northeastern, northwestern, and southeastern HMA subregions (Figure 1b), respectively (Table 1)."

*Line 224 – glaciers are assumed constant? Is this true for these simulations or just the default setting? If so, how would keeping glaciers constant (I'm guessing areal extent of glacier cover in grid cell?) shape the results?*

Yes, the areal extent of glaciers is assumed to be constant throughout the simulation periods. To simulate dynamical changes in glacial extent, CLM needs to be coupled with the Community Ice Sheet Model (CISM), the dynamic ice sheet component of CESM. In the CESM model version we used, active CLM-CISM coupling is only possible over the Greenland Ice sheet. Ongoing developments, however, will enable active CLM-CISM coupling over mountain regions such as HMA. Therefore, we expect that in future CESM model versions, it will be possible to simulate dynamical changes in HMA glaciers.

The effects of constant glacier extent on simulation outcomes depend on the amount of time a glacier needs to respond to climatic changes. Small glaciers have generally a short response time and are more vulnerable to climate change than large glaciers with a long response time. For instance, (Shea et al., 2015) found that small glaciers in the Everest region (i.e., located in the SE-HMA subregion) have a response time of 20–50 years, whereas large debris-covered glaciers have a response time of 200–500 years. Since our simulations cover a 20-yr period (1979–1998), we expect that the effects of constant glacial extent on simulation outcomes are rather small. However, to quantify these effects, an active coupling between CLM and CISM would be required.

*Line 235 and 250 – could these fixed, solely temperature-based thresholds influence SMB bias? Would implementing Jennings et al. 2018 temperature-relative humidity-based rainsnow thresholds help (see Equations 3-4 in Jennings et al. 2018) to increase the probability of more snowfall and a more positive SMB (see discussion on hydrometeor energy balance in Jennings et al. 2018 for physical intuition on why accounting for humidity matters for snowfall)? This might be difficult given computational limitations (although several authors are at NCAR and may have access to additional computational time on Cheyenne), but could the authors run an experiment (5-10 years) with the HMA 7km grid and simply swap in the Jennings et al., 2018 temperature-humidity based rainsnow partition scheme to compare how SMB biases are altered (maybe add to Table 3)?*

*Jennings, K.S., Winchell, T.S., Livneh, B. et al. Spatial variation of the rain–snow temperature threshold across the Northern Hemisphere. Nat Commun 9, 1148 (2018). https://doi.org/10.1038/s41467-018-03629-7*

Thank you for raising this point. The Jennings et al.-based temperature-based thresholds that are used in the HMA_VR7b simulation have shown an increase in the amount of snowfall relative to the temperature-based thresholds used in the HMA_VR7a simulation. Figure 2 of Jennings et al. (2018) plots the relative humidity (RH) and surface pressure (PS) against surface air temperature (TS). The figure shows that, for example, for Ts = 2 °C, lower RH and PS values result in a higher snowfall frequency. In this light, using a TS-RH-based threshold (Jennings Equation 3) or a TS-RH-PS-based threshold (Jennings Equation 4) could potentially help to increase the snowfall frequency and lower the SMB bias. Jennings Figure 6 shows, however, that on average the Eq. 3 and Eq. 4 based methods result in a lower snowfall frequency than the 2°C 50% rain-snow TS threshold used for the HMA_VR7b simulation. Nonetheless, the lower snowfall frequency probably does not occur over HMA as shown in Jennings Supplementary Figure 2, which shows a positive difference over HMA between the Eq. 4 and Eq. 3 based methods. This could mean that compared to the fixed TS-based threshold used in this study, the Eq. 4 based method also results in more snowfall over HMA. Therefore, it could be worth to exploring the effects of the Eq. 4 based method on the SMB bias.

Unfortunately, the computational resources are currently limited and mainly reserved for nudging experiments over HMA. For this reason, implementing Jennings Eq. 4 is not possible in the short term. However, we would like to consider implementing Jennings Eq. 4 and exploring its effects in future simulations when allocation will be used to improve land-atmosphere coupling. Also, we will add some text to the discussion in the manuscript to point to the limitations of using a fixed TS-based threshold.

*Line 138, 232 and 247 – the assumption of capping snow depths at 1 m and 5 m w.e. in HMA seems like another culprit for SMB bias. How often does snow depth hit the cap over the 20-year simulations? Could the authors provide a cumulative annual "snow loss" estimate due to snow capping and compare contrast with SMB bias (I'm guessing this might influence refreezing portion of SMB in Table 3 values)?*

The snow depth generally hits the cap only in those grid cells where a positive SMB is simulated. These grid cells are located in the southernmost part of HMA. The cumulative annual "snow loss" due to snow capping, which can be expressed as the excess amount of snow due to snow capping, amounts to about 2 (+/- 1) Gt yr$^{-1}$ for HMA_VR7a and about 1 (+/- 0.1) Gt yr$^{-1}$ for HMA_VR7b. This means the increased maximum allowed snow depth in HMA_VR7b has a positive impact on the cumulative annual "snow loss" and has resulted in increased refreezing (from 29 Gt yr$^{-1}$ for HMA_VR7a to 32 Gt yr$^{-1}$ for HMA_VR7b), which itself has a positive impact on the SMB bias. Also, the cumulative annual "snow loss" cannot be compared with the SMB bias, which is more than a factor 100 larger than the "snow loss". In other regions, the snow depth does not reach the cap and "snow loss" due to capping cannot occur. From the perspective of CLM, the SMB is zero in these grid cells where snow is present or negative when snow is absent. In these regions, we simulate a negative SMB, which can be attributed mainly to large amounts of ice melt (and thus the absence of snow). Therefore, we think that the assumption of capping snow depths cannot be a culprit for the SMB bias.

*Line 265-290 – the authors might consider evaluating SWE model estimates from VRCESM compared with Liu et al. 2021 (note the period of record does not overlap with the VR-CESM simulations, but a climatological comparison might still be useful). In addition, ERA5-Land might also be useful too. This would be especially insightful for the snow capping assumption/issue. "…It can be accessed through*
*https://nsidc.org/data/HMA_SR_D/ (last access: 22 April 2021) or*
*Https://doi.org/10.5067/HNAUGJQXSCVU (Liu et al., 2021). The dataset is provided as NetCDF files for each 1∘×1∘ tile shown in Fig. 1, available at 16 arcsec (~500 m) and daily resolution from WYs 2000 to 2017…"*

*Liu, Y., Fang, Y., and Margulis, S. A.: Spatiotemporal distribution of seasonal snow water equivalent in High Mountain Asia from an 18-year Landsat–MODIS era snow reanalysis dataset, The Cryosphere, 15, 5261–5280, https://doi.org/10.5194/tc-15-5261-2021,2021.*

*https://cds.climate.copernicus.eu/cdsapp#!/dataset/reanalysis-era5-land?tab=overview*

Thank you for your suggestions. We decided not to use ERA5-Land and the datasets of Liu et al. (2021) for several reasons. The datasets of Liu et al. (2021) are very valuable, but require a lot of storage and are computationally heavy. For example, 5 years of output for one 1-degree tile (e.g., 28-29N; 84-85E) require around 3 GB of data. ERA5-Land and ERA5 are both known to overestimate snow depth and snowfall over HMA (Orsolini et al., 2018). For this reason, we initially selected JRA-55 as the evaluation dataset since it has the best performance in the representation of snow depth. We have, however, found an alternative dataset that also can be used for snow depth evaluation, which is the High Asia Refined analysis version 2 (HARv2; Wang et al., 2021). This dataset is based on WRF simulations that are dynamically downscaled with ERA5 and corrected with snow depth from JRA-55. Also, the spatial resolution of HARv2 (~10 km) and period (1980–1998) match good with the HMA VR simulations. Figure RC2.1 shows the snow depth differences between the NE30 and HMA VR configurations, and the reanalysis-based HARv2 dataset. Snow depth is mainly overestimated over SW-HMA and SE-HMA during winter and spring, in part because of the cold bias. During autumn (over SW-HMA), and winter (over NW-HMA) the snow depth is slightly underestimated, which interestingly matches better with the negative snow cover biases during autumn and winter in the respective regions (Figure 9a). In the manuscript we will replace the JRA-55 based snow depth evaluation by the HARv2 snow depth evaluation.

[Figure]

**Figure RC2.1** Boxplots of snow depth differences (mm w.e.) between the simulation outputs of NE30 (red), HMA_VR7a (green), HMA_VR7b (blue), and HARv2 for each season and HMA subregion (shown in Figure 1b). The box represents the biases between the 25th and 75th percentile, the line in the box denotes the median, and the whiskers represent the 10th and 90th percentile of snow depth differences.

*Line 305 – eddy or geopotential height?*

It is eddy geopotential thickness, which is defined as the deviation of the geopotential thickness from the zonal mean of the geopotential thickness, where the geopotential thickness is calculated as the difference between geopotential heights at 500 hPa and 1000 hPa.

*Line 314-321 – very informative! It might be worth mentioning here that 2m surface temperatures don't always correspond/agree with free/upper atmosphere warm biases, particularly over HMA (the authors later show that 2m surface temperatures are colder than expected in most HMA regions) to hedge reader expectation/confusion.*

We have changed the following sentence (Line 314-315) from:

"The upper tropospheric summer temperature biases (relative to ERA5) can help to understand the effects of atmospheric biases on temperature-sensitive surface variables such as ice melt and snow melt."

to

"The upper tropospheric summer temperature biases (relative to ERA5) can help to understand the effects of atmospheric biases on temperature-sensitive surface variables such as ice melt and snow melt. However, atmospheric biases do not necessarily need to correspond with surface temperature biases as we will show in Section 3.3."

*Figure 3 – for the temperature anomaly plots, the authors might consider decreasing the number of color bins to 0.5 deg C bins (current precision is hard to interpret).*

We will decrease the number of color bins to 0.5 deg C as suggested by the reviewer.

*Figure 6 – could the authors add subpanel labels (a, b, …) and point to them while describing physical interpretation in Lines 343-356 (which is well explained, but pointing to specific plots might help those that aren't super familiar with monsoonal/precipitation dynamics in this region). Define 2nd y-axes on first column of subpanels (or remove).*

We will add subpanel labels and point to them accordingly in the manuscript. Also, we will add a label to define the 2nd y-axis.

*Line 358-376 – should median biases be stated? I think the authors should at least highlight that most of the distribution of the box-and-whiskers fall below the 0 deg C bias line (cold bias), save for JJA in the NW- and NE-HMA, and particularly VR-based distributions.*

We think it is good to quantify the biases to give an impression on the magnitude of the biases for the reader. In our opinion, the median of the biases could be a good reference point. For this reason, we prefer to keep the median biases in the text.

*Figure 7 – "absolute temperature" shouldn't all values then be positive?*

Our aim was to distinguish between "absolute" and "relative" changes, where "absolute" change is expressed in terms of a quantitative change and a "relative" change is expressed in terms of a percent change. To avoid further confusion, we will remove "absolute" from the manuscript text and figures' captions.

*Figure 8 – the use of a-b and c-d are incorrect, should be a-c (rainfall) and b-d (snowfall)*

We have changed "for rainfall (mm month-1) (a-b) and snowfall (mm month-1) (c-d)" to "for rainfall (mm month-1) (a-c) and snowfall (mm month-1) (b-d)"

*Line 368-376, 323-330 and 358-376 – Bambach et al. 2022 and Rhoades et al. 2018 (and others) have shown that a cold bias with elevation is also seen in non-glaciated regions in mid-latitude mountain regions (and without the application of the downscaling/ECs). Do the authors think that the thinner clouds could be a culprit for the colder surface temperatures (Figure 7)? I'm thinking during the day there could be more shortwave insolation, but in the night (lower cloud fraction/thinner clouds) there could be enhanced radiative cooling with the net daily balance being negative? Or is this surface temperature cold bias driven more by a lack of longwave feedbacks and/or minimal boundary layer turbulence over snow? Slater et al. (2001) had a hypothesis for a positive feedback loop that creates stable boundary layers, particularly over winter/snow conditions (see Figure 8 in Slater et al., 2001). Also, if this cold bias is fixed, it would likely worsen the SMB bias (unless the cold bias is partly due to the melt energy extracted from the atmosphere to the glacier, as the authors hypothesize).*

*Bambach, N. E., Rhoades, A. M., Hatchett, B. J., Jones, A. D., Ullrich, P. A., & Zarzycki, C. M. (2022). Projecting climate change in South America using variable-resolution Community Earth System Model: An application to Chile. International Journal of Climatology, 42( 4), 2514– 2542. https://doi.org/10.1002/joc.7379*

*Rhoades, A. M., Ullrich, P. A., Zarzycki, C. M., Johansen, H., Margulis, S. A., Morrison, H., et al. (2018). Sensitivity of mountain hydroclimate simulations in variable-resolution CESM to microphysics and horizontal resolution. Journal of Advances in Modeling Earth Systems, 10, 1357– 1380. https://doi.org/10.1029/2018MS001326*

*Slater, A. G., Schlosser, C. A., Desborough, C. E., Pitman, A. J., Henderson-Sellers, A., Robock, A., Vinnikov, K. Y., Entin, J., Mitchell, K., Chen, F., Boone, A., Etchevers, P., Habets, F., Noilhan, J., Braden, H., Cox, P. M., de Rosnay, P., Dickinson, R. E., Yang, Z., Dai, Y., Zeng, Q., Duan, Q., Koren, V., Schaake, S., Gedney, N., Gusev, Y. M., Nasonova, O. N., Kim, J., Kowalczyk, E. A., Shmakin, A. B., Smirnova, T. G., Verseghy, D., Wetzel, P., & Xue, Y. (2001). The Representation of Snow in Land Surface Schemes: Results from PILPS 2(d), Journal of Hydrometeorology, 2(1), 7-25. Retrieved Feb 13, 2023, from https: //journals.ametsoc.org/view/journals/hydr/2/1/1525-7541_2001_002_0007_trosil_2_0_co_2.xml*

There are several potential explanations for the cold biases in the NE30 and HMA VR simulation outputs besides the combination of better resolved topography and EC downscaling we mentioned in the manuscript. First, we consider it likely that the cold bias is partly a result of uncertainties in the observation/reanalysis-based WFDEI we used to evaluate surface temperatures. Since WFDEI is bias-corrected using gridded observations of GPCC, the accuracy of the data relies on the availability of meteorological measurements that are scarce in High Mountain Asia, especially at higher altitudes and in the more remote domains of HMA. The lack of measurements could result in temperature overestimates, but also in precipitation underestimates (Palazzi et al., 2015; Immerzeel et al., 2015; Gu et al., 2012; Lalande et al., 2021), which can to some extent explain the cold and wet biases that are visible in the NE30 and HMA VR simulation outputs.

Second, the thinner clouds can contribute to the colder surface temperatures as suggested by the reviewer, especially during wintertime when the net radiative balance is already negative (Table RC2.3). During the day, the thinner clouds could result in more shortwave insolation, which in summer can contribute to more melting. During the night, when the radiative balance is negative, the thinner clouds could result in enhanced radiative cooling. The enhanced radiative cooling could then eventually lead to a negative net daily radiative balance, especially during wintertime. Therefore, the thinner clouds could also explain to some extent the cold biases that are present in the simulation outputs.

Third, the cold biases could partly be a result of the stability-induced cooling feedback (Figure 8 in Slater et al., 2001). During winter, the net radiative balance is negative (Table RC2.3) and sensible/latent heat fluxes are close to zero. Further, the ground heat flux is positive towards the surface, partly countering the negative net radiative balance. Radiative cooling occurs, which causes the surface temperature to be lower than the overlying air temperature (i.e., a decoupling of the surface from the atmosphere). Via the stability-induced cooling feedback, the radiative cooling can be enhanced, which eventually results in cold biases, particularly during wintertime. One way to prevent a stability-induced cooling feedback could be to improve the coupling between land and atmosphere as already outlined in the manuscript. We will add text discussing the possible links between cold biases and the thinner clouds, stability-induced cooling feedback, and the uncertainties in the observation/reanalysis-based WFDEI.

**Table RC2.3.** Mean wintertime (DJF) surface-energy-balance (SEB) fluxes (W m$^{-2}$) for NE30, HMA_VR7a (HMA1), HMA_VR7b (HMA2), JRA55 and CERES-EBAF (C-EBF). SEB fluxes are defined positive towards the surface and represent the mean fluxes in glaciated grid cells (encompassing all land units), averaged over the entire HMA region and the periods 1979–1998 (for NE30, HMA_VR7a, HMA_VR7b, and JRA55) and 2001–2020 (for CERES-EBAF).

| SEB fluxes (W m$^{-2}$) | NE30 | HMA1 | HMA2 | JRA55 | C-EBF |
|---|---|---|---|---|---|
| SW$_d$ | 162 | 168 | 164 | 162 | 133 |
| SW$_u$ | 87 | 85 | 83 | 80 | 42 |
| SW$_{net}$ | 75 | 83 | 80 | 81 | 92 |
| LW$_d$ | 179 | 165 | 171 | 175 | 207 |
| LW$_u$ | 261 | 253 | 256 | 263 | 277 |
| LW$_{net}$ | -82 | -88 | -85 | 89 | -71 |
| R$_{net}$ | -7 | -5 | -5 | -8 | 21 |
| SHF | ~0 | -0.5 | -2 | 7 | - |
| LHF | -5 | -4 | -4 | -14 | - |
| GHF | 13 | 12 | 12 | 16 | - |
| MHF | 2 | 2 | 2 | 2 | - |
| Refreezing | 0.7 | 0.6 | 0.6 | - | - |

*Line 377-391 – could the authors provide cumulative annual/monthly precipitation totals to better contextualize monthly average biases? I'm not familiar with the HMA region's cumulative precipitation totals and can't contextualize the +/- monthly biases stated.*

We will add a table (Table S2) to the Supplementary Information that contains the mean winter (DJF) and summer (JJA) 2m temperature, rainfall, snowfall, snow cover, and snow depth for NE30, HMA_VR7a, HMA_VR7b, and the observation/satellite/reanalysis-based datasets. The seasonal sums/means are calculated over the HMA subregions for the period 1979–1998.

**Table S2.** Mean winter (DJF) and summer (JJA) 2m temperature, rainfall, snowfall, snow cover, and snow depth for NE30, HMA_VR7a, HMA_VR7b, WFDEI (temperature + precipitation), NSIDC (snow cover), and HARv2 (snow depth). The seasonal means/sums are calculated over the HMA subregions (Figure 1b) for the period 1979–1998.

| | SW-HMA | | SE-HMA | | NW-HMA | | NE-HMA | |
|---|---|---|---|---|---|---|---|---|
| *2m Temperature (°C)* | **DJF** | **JJA** | **DJF** | **JJA** | **DJF** | **JJA** | **DJF** | **JJA** |
| NE30 | -12 | 13 | -1 | 15 | -9 | 18 | -14 | 12 |
| HMA_VR7a | -16 | 10 | -2 | 14 | -11 | 16 | -14 | 12 |
| HMA_VR7b | -15 | 10 | -1 | 14 | -11 | 17 | -14 | 12 |
| WFDEI | -8 | 12 | 0 | 15 | -10 | 16 | -12 | 10 |
| | | | | | | | | |
| *Rainfall (mm)* | **DJF** | **JJA** | **DJF** | **JJA** | **DJF** | **JJA** | **DJF** | **JJA** |
| NE30 | 25 | 383 | 52 | 1109 | 14 | 84 | 1 | 342 |
| HMA_VR7a | 21 | 420 | 37 | 1171 | 11 | 82 | 0.4 | 290 |
| HMA_VR7b | 19 | 435 | 37 | 1130 | 11 | 84 | 0.3 | 281 |
| WFDEI | 48 | 145 | 38 | 740 | 16 | 75 | 1 | 163 |
| | | | | | | | | |
| *Snowfall (mm)* | **DJF** | **JJA** | **DJF** | **JJA** | **DJF** | **JJA** | **DJF** | **JJA** |
| NE30 | 237 | 9 | 76 | 4 | 62 | 2 | 30 | 20 |
| HMA_VR7a | 193 | 33 | 58 | 41 | 50 | 12 | 17 | 34 |
| HMA_VR7b | 208 | 43 | 62 | 53 | 61 | 14 | 19 | 39 |
| WFDEI | 99 | 12 | 23 | 50 | 45 | 3 | 9 | 18 |
| | | | | | | | | |
| *Snow Cover (%)* | **DJF** | **JJA** | **DJF** | **JJA** | **DJF** | **JJA** | **DJF** | **JJA** |
| NE30 | 81 | 2 | 30 | 0.1 | 66 | 0.2 | 61 | 0.2 |
| HMA_VR7a | 83 | 5 | 28 | 2 | 65 | 1 | 53 | 1 |
| HMA_VR7b | 83 | 7 | 28 | 2 | 67 | 1 | 52 | 1 |
| NSIDC | 75 | 42 | 24 | 9 | 68 | 17 | 25 | 2 |
| | | | | | | | | |
| *Snow Depth (mm w.e.)* | **DJF** | **JJA** | **DJF** | **JJA** | **DJF** | **JJA** | **DJF** | **JJA** |
| NE30 | 157 | 27 | 26 | 0.6 | 42 | 2 | 22 | 0.2 |
| HMA_VR7a | 121 | 24 | 27 | 8 | 38 | 1 | 15 | 0.4 |
| HMA_VR7b | 130 | 39 | 50 | 28 | 43 | 2 | 16 | 1 |
| HARv2 | 96 | 34 | 17 | 5 | 57 | 4 | 14 | 1 |

*Line 381 – change "worst" to "worse"*

We have changed "worst" to "worse".

*Line 386 – I'm curious how accounting for both temperature and humidity within the Jennings et al. 2018 rain-snow partitioning scheme(s) would alter this snowfall bias*

We also wonder how the Jennings rain-snow partitioning scheme would alter the snowfall bias. We hope to find out the performance of Jennings rain-snow partitioning scheme in future simulations.

*Line 400-406 and Figure 9 – I'd recommend comparison with Liu et al. (2021) to better verify snow depth/SWE. The authors might also consider using ERA5-Land. ERA5-Land could be useful for other surface variable comparisons (esp. since it's at a much more comparable resolution, ~9km, to VR simulations and the authors wouldn't need to coarsen for comparison).*

*Liu, Y., Fang, Y., and Margulis, S. A.: Spatiotemporal distribution of seasonal snow water equivalent in High Mountain Asia from an 18-year Landsat–MODIS era snow reanalysis dataset, The Cryosphere, 15, 5261–5280, https://doi.org/10.5194/tc-15-5261-2021, 2021.*

*https://cds.climate.copernicus.eu/cdsapp#!/dataset/reanalysis-era5-land?tab=overview*

To verify snow depth, we have chosen to compare snow depth with the High Asia Refined analysis version 2 instead of JRA-55.

*Line 411-413 – this finding matches one of my earlier comments/questions about LW feedbacks potentially driving surface temperature cold biases. Although fixing this bias might further impact SMB bias since snow/ice are nearly blackbodies in LW spectrum…*

We think that improving the subgrid coupling between land and atmosphere by introducing CLM patch information into CAM could reduce the cold bias. Also, nudging the vertical levels of CAM with data from ERA5 could reduce the cold and SMB bias by increasing the cloud cover over HMA. We are planning to conduct nudging experiments and experiments with improved land-atmosphere coupling schemes (with CLASP, Waterman et al., 2022) that will hopefully improve the SMB and temperature biases.

*Line 427 – I may have misinterpreted this, but I thought that there was reduced/thinner cloud cover (see Line 327-330) not greater, as stated here.*

Yes, relative to NE30 the HMA VR simulations have reduced/thinner cloud cover, but relative to HMA_VR7a, HMA_VR7b has greater cloud cover due to the cloud tunings that were implemented in HMA_VR7b. We will clarify the text by making clear that the greater cloud cover mentioned here is an observation relative to the cloud cover simulated in HMA_VR7a.

*Line 436 – the extensive ice melt (and heat extraction from environment) could also be driving the surface temperature bias?*

Although the surface temperature cold bias is smaller during summer, we think it likely that the extensive ice melt can also contribute to the surface temperature bias. As mentioned in the manuscript, the extensive melt requires heat extraction from the environment, causing the ground heat flux to be positive towards the surface and the surface temperature to be lower. Over the ablation zones of the Greenland Ice Sheet, cold temperature biases are also present during summer that, to some extent, overlap with regions where ground heat fluxes are positive and melt heat fluxes are large (Figures 4 and 7 of Kampenhout et al., 2020). We will mention in the manuscript that the extensive ice melt via heat extraction from the environment likely also contributes to the cold temperature biases.

*Line 441-457 and Figure 13 – given that SMB doesn't appear to be water limited (i.e., VRCESM produces too much precipitation/snowfall in most HMA regions in Figure 8), how much does the atmospheric variable SMB downscaling method influence the SMB bias (particularly temperature/surface energy fluxes)? I'm wondering how the use of a fixed lapse rate (6 K/km and 32 W/m^2\*km) and constant relative humidity with elevation shapes the negative SMB, particularly in mountains where lapse rates can vary quite a lot, even ones much smaller and with less heterogeneity than HMA (see Lute and Abatzoglou, 2020)*

*Lute, AC, Abatzoglou, JT. Best practices for estimating near-surface air temperature lapse rates. Int J Climatol. 2021; 41 (Suppl. 1): E110– E125. https://doi.org/10.1002/joc.6668*

The effects of EC downscaling on SMB and other SEB variables such as melt heat energy have been investigated by Sellevold et al. (2019). Comparing four CESM1 simulations applying four different temperature lapse rates (1 K km$^{-1}$, 4 K km$^{-1}$, 6 K km$^{-1}$, and 9.8 K km$^{-1}$), the authors found that the lapse rates of 6 K km$^{-1}$ and 9.8 K km$^{-1}$ result in the most realistic SMB and melt gradients, respectively. The simulations using lapse rates of 1 K km$^{-1}$ and 4 K km$^{-1}$ performed worse. Although the study of Sellevold et al. (2019) demonstrate that a lapse rate of 6 K km$^{-1}$ results in a realistic SMB, the study focusses on Greenland and applies a fixed lapse rate in the same way we do. As the reviewer already mentions, lapse rates can vary both spatially and temporally over mountain ranges, such as HMA. For example, in Langtang Valley, a glaciated valley located in Nepal (SE-HMA), temperature lapse rates (based on observations) can vary between 5.2 K km$^{-1}$ in summer and 7.6 K km$^{-1}$ in spring (Wijngaard et al., 2019). Further it is found that temporally variable lapse rates can improve the simulation of melt and runoff in glaciated valleys (Immerzeel et al., 2014). We are therefore convinced that spatially and temporally variables lapse rates can give better results. However, the limitation is that the estimation of temporally and variable lapse rates often relies on meteorological observations, which are scarce in HMA and other mountain ranges, particularly at higher altitude.

*Figure 11 – do the initial condition spin ups of the VR-CESM experiments have anything to do with why SMB starts, even in the first year, with such large losses compared to both observations/WRF? This is likely a naïve comment (not a glacier expert), but would there be any way to initialize the VR-CESM simulations to start with glacier/snow thickness comparable to observations/WRF instead of relying on the CESM spin up procedure? Or could the authors use ERA5 to spin up a standalone CLM simulation that could then provide initial conditions to the AMIP VR-CESM simulations?*

For the HMA_VR7a simulation, the initial conditions spin-up could have played a role in starting with large losses compared to both observations/WRF-based outputs. However, for the HMA_VR7b simulation, we initialized the model with a snow depth of 2.5 m w.e. in every glacierized land unit. This could be similar to forcing CESM with snow thickness that is comparable with observations, except that we only applied the initialization over glacierized land units. Our experience was that in the first 5 years of the atmospheric spin-up, most of the snow over the glacierized land units already disappeared. Based on these findings, we do not think that the initial condition spin-ups have a negative impact on the SMB bias.

As for spinning up a CLM standalone simulation with ERA5, we have conducted CLM standalone simulations driven by GSWP meteorological forcings (for details we refer to our response to the second comment of Reviewer 1). As it turned out, the CLM standalone simulations forced by GSWP resulted in a more negative SMB than the HMA VR simulations, primarily due to the warmer surface climate and smaller precipitation volumes in GSWP.

*Line 455 – delete "than"*

"Than" is embedded in a sentence where a comparison is made between the SMB simulated over higher-altitude glaciers and the SMB simulated over "lower-altitude" glaciers. Grammatically it will therefore be incorrect to delete "than".

*Line 486-501 – agreed… major bummer on the 10x SMB issue in HMA (expected SMB is -20 GT/yr, VR-CESM is, at best, -200 GT/yr). I hope the nudging exercise can help alleviate some of the large-scale biases in temperature. I hope some of my random ideas above help too.*

We are very grateful for all the helpful suggestions. We also hope that the nudging experiments and experiments with improved land-atmosphere coupling schemes will improve the SMB bias and can alleviate some of the biases in temperature.

*Line 546 – I'd argue most regions simulated by VR-CESM have cold biases (and across most seasons too)*

We agree. We will rephrase the text to put more emphasis on the cold temperature biases.

**References**

Flatau, P. J., Walko, R. L., and Cotton, W. R.: Polynomial Fits to Saturation Vapor Pressure, Journal of Applied Meteorology, 31, https://doi.org/10.1175/1520-0450(1992)031<1507:pftsvp>2.0.co;2, 1992.

Gu, H., Wang, G., Yu, Z., and Mei, R.: Assessing future climate changes and extreme indicators in east and south Asia using the RegCM4 regional climate model, Clim Change, 114, https://doi.org/10.1007/s10584-012-0411-y, 2012.

He, C., Li, Q. B., Liou, K. N., Zhang, J., Qi, L., Mao, Y., Gao, M., Lu, Z., Streets, D. G., Zhang, Q., Sarin, M. M., and Ram, K.: A global 3-D CTM evaluation of black carbon in the Tibetan Plateau, Atmos Chem Phys, 14, https://doi.org/10.5194/acp-14-7091-2014, 2014.

Iacono, M. J., Delamere, J. S., Mlawer, E. J., Shephard, M. W., Clough, S. A., and Collins, W. D.: Radiative forcing by long-lived greenhouse gases: Calculations with the AER radiative transfer models, Journal of Geophysical Research Atmospheres, 113, https://doi.org/10.1029/2008JD009944, 2008.

Immerzeel, W. W., Petersen, L., Ragettli, S., and Pellicciotti, F.: The importance of observed gradients of air temperature and precipitation for modeling runoff from a glacierized watershed in the Nepalese Himalayas, Water Resour Res, 50, 2212–2226, https://doi.org/10.1002/2013WR014506, 2014.

Immerzeel, W. W., Wanders, N., Lutz, A. F., Shea, J. M., and Bierkens, M. F. P.: Reconciling high altitude precipitation with glacier mass balances and runoff, Hydrol Earth Syst Sci, 12, 4755–4784, https://doi.org/10.5194/hessd-12-4755-2015, 2015.

Kang, S., Zhang, Y., Qian, Y., and Wang, H.: A review of black carbon in snow and ice and its impact on the cryosphere, https://doi.org/10.1016/j.earscirev.2020.103346, 2020.

Lalande, M., Ménégoz, M., Krinner, G., Naegeli, K., and Wunderle, S.: Climate change in the High Mountain Asia in CMIP6, Earth System Dynamics, 12, https://doi.org/10.5194/esd-12-1061-2021, 2021.

Li, D., Lu, X., Walling, D. E., Zhang, T., Steiner, J. F., Wasson, R. J., Harrison, S., Nepal, S., Nie, Y., Immerzeel, W. W., Shugar, D. H., Koppes, M., Lane, S., Zeng, Z., Sun, X., Yegorov, A., and Bolch, T.: High Mountain Asia hydropower systems threatened by climate-driven landscape instability, Nat Geosci, 15, 520–530, https://doi.org/10.1038/s41561-022-00953-y, 2022.

Liu, Y., Fang, Y., and Margulis, S. A.: Spatiotemporal distribution of seasonal snow water equivalent in High Mountain Asia from an 18-year Landsat-MODIS era snow reanalysis dataset, Cryosphere, 15, https://doi.org/10.5194/tc-15-5261-2021, 2021.

Palazzi, E., Von Hardenberg, J., Terzago, S., and Provenzale, A.: Precipitation in the Karakoram-Himalaya: a CMIP5 view, Clim Dyn, 45, 21–45, https://doi.org/10.1007/s00382-014-2341-z, 2015.

Rahimi, S., Liu, X., Wu, C., Lau, W. K., Brown, H., Wu, M., and Qian, Y.: Quantifying snow darkening and atmospheric radiative effects of black carbon and dust on the South Asian monsoon and hydrological cycle: experiments using variable-resolution CESM, Atmos Chem Phys, 19, 12025–12049, https://doi.org/10.5194/acp-19-12025-2019, 2019.

Rhoades, A. M., Ullrich, P. A., Zarzycki, C. M., Johansen, H., Margulis, S. A., Morrison, H., Xu, Z., and Collins, W. D.: Sensitivity of Mountain Hydroclimate Simulations in Variable-Resolution CESM to Microphysics and Horizontal Resolution, J Adv Model Earth Syst, 10, 1357–1380, https://doi.org/10.1029/2018MS001326, 2018.

Shea, J. M., Immerzeel, W. W., Wagnon, P., Vincent, C., and Bajracharya, S. R.: Modelling glacier change in the Everest region, Nepal Himalaya, Cryosphere, 9, 1105–1128, https://doi.org/10.5194/tc-9-1105-2015, 2015.

Slater, A. G., Schlosser, C. A., Desborough, C. E., Pitman, A. J., Henderson-Sellers, A., Robock, A., Vinnikov, K. Y., Mitchell, K., Boone, A., Braden, H., Chen, F., Cox, P. M., De Rosnay, P., Dickinson, R. E., Dai, Y. J., Duan, Q., Entin, J., Etchevers, P., Gedney, N., Gusev, Y. M., Habets, F., Kim, J., Koren, V., Kowalczyk, E. A., Nasonova, O. N., Noilhan, J., Schaake, S., Shmakin, A. B., Smirnova, T. G., Verseghy, D., Wetzel, P., Xue, Y., Yang, Z. L., and Zeng, Q.: The representation of snow in land surface schemes: Results from PILPS 2(d), J Hydrometeorol, 2, https://doi.org/10.1175/1525-7541(2001)002<0007:TROSIL>2.0.CO;2, 2001.

Wang, X., Tolksdorf, V., Otto, M., and Scherer, D.: WRF-based dynamical downscaling of ERA5 reanalysis data for High Mountain Asia: Towards a new version of the High Asia Refined analysis, International Journal of Climatology, 41, 743–762, https://doi.org/10.1002/joc.6686, 2021.

Wijngaard, R. R., Steiner, J. F., Kraaijenbrink, P. D. A., Klug, C., Adhikari, S., Banerjee, A., Pellicciotti, F., van Beek, L. P. H., Bierkens, M. F. P., Lutz, A. F., and Immerzeel, W. W.: Modeling the response of the langtang glacier and the hintereisferner to a changing climate since the little ice age, Front Earth Sci (Lausanne), 7, https://doi.org/10.3389/feart.2019.00143, 2019.

---

## Author Response (AR2)

We gratefully acknowledge the reviewer for his/her remarks and suggestions, which improved the quality of the manuscript significantly. We have carefully considered the suggestions of the reviewer and we provide a point-by-point response to the reviewer's comments. For clarity, the reviewer's comments are given in bold italics and the responses are given in plain text. References that do not refer to those in the main manuscript are listed below. The manuscript will be modified according to the responses that are given to the comments.

*I'd like to thank authors' addressing many of my questions and comments (in RC1) by their revision and replies. I am able to better understand the logics from the revised figures and texts. Especially I gained more insights from the new off-line CLM simulations.*

*While the revised manuscript reads great to me, I still have a major concern about how the authors justify their using hydrostatic dynamical core (along with other configurations of CAM).*

*First, I appreciate the authors introducing a new study that I was not aware of, Liu et al., 2022, and also clarifies that Rhodes et al. (2018) indeed conducted CESM-VR simulation with grid spacing reined to 7 km. Going over these two studies and the authors' replies, I still feel it is misleading how the authors (and Rhodes et al., 2018) to a lesser extent) explain the applicability of hydrostatic dynamical dynamical cores to the 7-km regionally-refined grid over Tibet, with the full (moist) physics.*

*I also believe that making the readers aware of these limitations would not affect the main conclusions of this study nor degrades its contribution to the community. I think the way the authors should justify the current CAM configuration is not by insisting that the hydrostatic dynamical core is appropriate for the grid used in this study, but instead explaining the following:*

*#1 This study is experimental; it describes new capabilities and/or high-resolution dataset for CESM's cryospheric-hydrological cycles over HMA, and evaluates the impact of these capabilities.*

*#2 Atmospheric model configurations fully appropriate for the spatial resolution required by this study are not available in the versions of CESM used (high vertical resolutions, non-hydrostatic dynamical cores, and the authors probably want to mention scale-aware deep convection schemes as well)*

*#3 The focus is the land surface (glaciers) and near-surface land-atmosphere coupling, not the atmospheric dynamics*

*#4 most importantly, the consistency between the results from atmosphere-land coupled simulations and off-line land simulations suggest that the main findings of the study is rather insensitive to atmospheric forcing; therefore, if all simulations are switched to non-hydrostatic atmospheric model, the main conclusions are expected to hold.*

*For #2, the authors can further mention the following for the readers:*
*- non-hydrostatic version of the SE dynamical core already exists (as in Liu et al. 2022), and another non-hydrostatic dynamical core MPAS is being tested in the CESM framework (Huang et al., 2023),*

*- (as the authors stated in their reply) "Recent developmental versions of CESM also have the option to run with 58 vertical levels by default."*

*- scale-aware deep convection schemes are also being ported to CESM/CAM (e.g., Jang et al., 2022)*

*therefore, it is expected that future studies can utilize more appropriate CAM configurations for the regionally-refined mesh developed in this study.*

*With that, let me further clarify why I do not think the hydrostatic dynamical core is appropriate for the model grid used in this study. There are three reasons.*

*1) The general agreement within the modeling communities, especially those who focus on moist convection with meso-scale and convection permitting resolution, is that grid spacing smaller than ~ 10km should use non-hydrostatic model.*

*To give more context, the topic of hydrostatic vs non-hydrostatic models is not new in the weather forecasting and mesoscale modeling fields (e.g., Tag and Rosmond 1980; Martin and Pielke 1983; Dudhia 1992; Kato and Saito 1995; Jang and Hong, 2016; Qi et al., 2018)*

*Difference between hydrostatic and non-hydrostatic models are somewhat model-dependent, but several studies with full moist physics suggest that even > 10 km grid spacing show difference between hydrostatic and non-hydrostatic models (Kato and Saito 1995; Jang and Hong, 2016; Qi et al., 2018). Furthermore, a review article and a textbook state that simulations with grid spacing < 10 km should use non-hydrostatic dynamical cores:*

*Review                                                                                                                            article:*
*Steppeler, J., Hess, R., Schättler, U., & Bonaventura, L. (2003). Review of numerical methods for nonhydrostatic weather prediction models. Meteorology and Atmospheric Physics, 82(1–4), 287–301. https://doi.org/10.1007/s00703-001-0593-8*

*"Operational numerical weather prediction (NWP) models are currently close to the 10km horizontal resolution threshold, beyond which the hydrostatic approximation becomes inaccurate." (p1, Introduction)*

*Textbook*
*Satoh, M. (2014). Atmospheric Circulation Dynamics and General Circulation Models. , 2nd ed., Berlin, Heidelberg: Springer Berlin Heidelberg. https://doi.org/10.1007/978-3-642-13574-3*

*"... As introduced in Chapter 20, however, the recent progress in computer resources has enabled us to drastically increase the horizontal resolution of global models, say, less than 10 km. For these high-resolution simulations, we need to switch the governing equations from hydrostatic equations to nonhydrostatic equations." (p608, ch.24 "Non-hydrostatic modeling")*

*therefore, I believe that the general consensus is that the grid spacing used in this study requires non-hydrostatic dynamical core.*

*2) Unrealistic, truncation-scale (below the effective resolution) vertical motions are more likely to be simulated in hydrostatic dynamical cores, especially when coupled with full moist physics.*

*It is recognized that hydrostatic models tend to simulate more intense vertical velocity than non-hydrostatic models (Tag and Rosmound 1980; Kato and Sato 1995,; Qi et al.,2018). Hydrostatic model does not have a term that decelerates upward motion through non-hydrostatic, perturbation pressure. In other words, the work required for a rising air parcel to push out the environmental air from its way (Davies-Jones 2003; Pauluis and Garner 2006; Morrison 2016; Jeevanjee and Romps 2016). This environmental response becomes non-linearly stronger as horizontal scale of the parcel, or grid column, becomes smaller(Jeevanjee and Romps 2016), meaning that neglection of this term in hydrostatic models become increasingly problematic at smaller scales. This limitation of the hydrostatic formulation is depicted by Pauluis et al., 2006 as follows,*
*"... In the atmospheric sciences, the system of equations in which the vertical momentum equation (1) has been replaced by the hydrostatic balance (6) is known as the primitive equations. ... One of the main limitations of the primitive equations lies in how they handle the kinetic energy of convective motions. ... For planetary- or synoptic-scale circulations, the vertical velocity is much smaller than the horizontal velocity, and its contribution to the kinetic energy is negligible. This is not the case for convective motions for which the use of the primitive equations leads to unrealistic values of the vertical velocity. In fact, analytic solutions of the primitive equations in a convectively unstable atmosphere yield infinite growth rate for short horizontal wavelengths."*

*Please note that models do response to forcing in the spatial scales smaller than the effective resolutions, usually not in a physically realistic way, and associated precipitation is termed as "grid-point" storm or "truncation-scale" storms (Held et al., 2007; Williamson 2013; section 7 in Gross et al., 2018; Marquet et al., 2019). Unphysical vertical wind is often accompanied by strong moisture convergence, condensation, and positive buoyancy production, which is mainly represented by the cloud microphysics in the CAM model. That Yang et al. (2017) and Liu et al. (2022; their section 4) showed the hydrostatic and non-hydrostatic difference become much larger with moist physics seems relevant to this limitation of the hydrostatic model. Also for this concern of grid-scale storms, the model effective resolution and its spatial scale (as used by some studies) are not relevant.*

*3) The present authors seem to underestimate the implication of the result from Yang et al. (2017) First, Yang et al. (2017) reported that the difference between hydrostatic and nonhydrostatic simulations is statistically significant in the mid-latitude case, but with much less magnitude than over the tropics. Second, while the maximum height of the mountain in Yang et al. (2017) is comparable to the topographic relief over Tibetan Plateau, the slope is far smaller than those over Tibetan Plateau. The idealized mountain in Yang et al., has the following parameters:*

*max height (h0) = 2km*
*half width (a) = 60km and 120km.*

*These parameters lead to the slopes of 1.9 degree with a = 60 km and 0.6 degree with a = 120km. On the other hand, slopes of > 10 degrees are common over the Tibetan Plateau (Liu-Zeng et al., 2007). Although the model input topography is usually smoothed, it is still likely that vertical wind forced by the impinging horizontal winds is stronger over the realistic Tibetan Plateau topography than over the idealized mountain in Yang et al., (2017).*

*I do not think the authors need to provide this much background information in their manuscript. But again please consider the above suggestions to caution the readers about the CAM configuration used in this study and introduce relevant, on-going development in CAM that enable better configuration for the scales considered in this study. Justification for using the current CAM configuration should be made in accordance with focus of the study, model development status, and robustness of the main findings regardless of the atmospheric forcing.*
* * *
*other reference:*

*Davies-Jones, R. (2003). An expression for effective buoyancy in surroundings with horizontal density gradients. Journal of the Atmospheric Sciences, 60(23), 2922–2925. https://doi.org/10.1175/1520-0469(2003)060<2922:AEFEBI>2.0.CO;2*

*Dudhia, J. (1992). A Nonhydrostatic Version of the Penn State–NCAR Mesoscale Model: Validation Tests and Simulation of an Atlantic Cyclone. Monthly Weather Review. https://doi.org/10.1175/1520-0493(1993)121<1493:ANVOTP>2.0.CO;2*

*Gross, M., Wan, H., Rasch, P. J., Caldwell, P. M., Williamson, D. L., Klocke, D., et al. (2018). Physics–Dynamics Coupling in weather, climate and Earth system models: Challenges and recent progress. Monthly Weather Review, MWR-D-17-0345.1. https://doi.org/10.1175/MWR-D-17-0345.1*

*Held, I. M., Zhao, M., & Wyman, B. (2007). Dynamic radiative-convective equilibria using GCM column physics. Journal of the Atmospheric Sciences, 64(1), 228–238. https://doi.org/10.1175/JAS3825.11*

*Huang, X., Gettelman, A., Skamarock, W. C., Lauritzen, P. H., Curry, M., Herrington, A., et al. (2022). Advancing Precipitation Prediction Using a New Generation Storm-resolving Model Framework -- SIMA-MPAS (V1.0): a Case Study over the Western United States. Geoscientific Model Development Discussions, 2022(June), 1–23. Retrieved from https://gmd.copernicus.org/preprints/gmd-2022-111/*

*Liu-Zeng, J., Tapponnier, P., Gaudemer, Y., & Ding, L. (2008). Quantifying landscape differences across the Tibetan plateau: Implications for topographic relief evolution. Journal of Geophysical Research: Earth Surface, 113(4), 1–26. https://doi.org/10.1029/2007JF000897*

*Kato, T., & Saito, K. (1995). Hydrostatic and Non-Hydrostatic Simulations of Moist Convection. Journal of the Meteorological Society of Japan. Ser. II, 73(1), 59–77. https://doi.org/10.2151/jmsj1965.73.1_59\Y*

*Jang, J., & Hong, S. Y. (2016). Comparison of simulated precipitation over East Asia in two regional models with hydrostatic and nonhydrostatic dynamical cores. Monthly Weather Review, 144(10), 3579–3590. https://doi.org/10.1175/MWR-D-15-0428.1*

*Jang, J., Skamarock, W. C., Park, S., Zarzycki, C. M., Sakaguchi, K., & Leung, L. R. (2022). Effect of the Grell-Freitas Deep Convection Scheme in Quasi-uniform and Variable-resolution Aquaplanet CAM Simulations. Journal of Advances in Modeling Earth Systems. https://doi.org/10.1029/2020ms002459*

*Jeevanjee, N., & Romps, D. M. (2016). Effective buoyancy at the surface and aloft. Quarterly Journal of the Royal Meteorological Society, 142(695), 811–820. https://doi.org/10.1002/qj.2683*

*Marquet, P., Descamps, L., & Bouyssel, F. (2019). A new " grid-point storm control " scheme in the ARPEGE NWP model. In Research Activities in Earth System Modelling (Vol. 49, pp. 4–11). Geneva: WMO Working Group on Numerical Experimentation. Retrieved from https://wgne.net/publications/*

*Martin, C. L., & Pielke, R. A. (1983). The Adequacy of the Hydrostatic Assumption in Sea Breeze Modeling over Flat Terrain. Journal of the Atmospheric Sciences, 40(6), 1472–1481. https://doi.org/10.1175/1520-0469(1983)040<1472:TAOTHA>2.0.CO;2*

*Qiang Sun, Y., Rotunno, R., & Zhang, F. (2017). Contributions of moist convection and internal gravity waves to building the atmospheric -5/3 kinetic energy spectra. Journal of the Atmospheric Sciences, 74(1), 185–201. https://doi.org/10.1175/JAS-D-16-0097.1*

*Pauluis, O., & Garner, S. (2006). Sensitivity of Radiative Convective Equilibrium Simulations to Horizontal Resolution. Journal of the Atmospheric Sciences, 63(7), 1910–1923. https://doi.org/10.1175/JAS3705.1*

*Pauluis, O., Frierson, D. M. W., Garner, S. T., Held, I. M., & Vallis, G. K. (2006). The hypohydrostatic rescaling and its impacts on modeling of atmospheric convection. Theoretical and Computational Fluid Dynamics, 20(5–6), 485–499. https://doi.org/10.1007/s00162-006-0026-x*

*Tag, P. M., & Rosmound, T. E. (1980). Accuracy and Energy Conservation in a Three-Dimensional Anelastic Model. Journal of the Atmospheric Sciences, 37(10), 2150–2168. https://doi.org/10.1175/1520-0469(1980)037<2150:AAECIA>2.0.CO;2*

*Williamson, D. L. (2013). The effect of time steps and time-scales on parameterization suites. Quarterly Journal of the Royal Meteorological Society, 139(January), 548–560. https://doi.org/10.1002/qj.1992*

We thank the reviewer for his/her evaluation of the revised manuscript. We appreciate the reviewer for carefully explaining the reasoning for a general consensus within the mesoscale modeling community that dx~10 km grid spacing is the limit at which hydrostatic dynamics are no longer valid. It is clear that decades of research have gone into developing this consensus. As our discussion has expanded such that we are debating a variety of issues, such as whether the effective resolution provides a constraint on permitted updrafts and grid-scale storms in a hydrostatic model, we'd like to focus specifically on this consensus to streamline the review process. The reviewer states:

"The general agreement within the modeling communities, especially those who focus on moist convection with meso-scale and convection permitting resolution, is that grid spacing smaller than ~ 10km should use non-hydrostatic model."

We remain skeptical on how rigorously this dx~10 km cut-off is intended to be adhered to, and in particular for deeming grid resolutions in its immediate vicinity as inappropriate (e.g., dx=7 km) or appropriate (e.g., dx=13 km) for a hydrostatic model. It is inconceivable that the actual cut-off grid-spacing in which hydrostatic dynamics are inappropriate would not be model dependent, a point the reviewer mentions ('Difference between hydrostatic and non-hydrostatic models are somewhat model-dependent'). Similarly, studies seeking out the true cut-off grid-spacing rely on model's with a non-hydrostatic/hydrostatic "switch." These non-hydrostatic configurations are not identical to the hydrostatic version other than the non-hydrostatic terms; the model structure necessarily changes to accommodate a new prognostic variable, e.g., introducing additional filters to remove errors associated with a different vertical discretization. That different modeling studies find that different grid spacings are important for non-hydrostatic dynamics we think speaks to the importance of these model dependent structural changes.

There are studies that indicate non-hydrostatic dynamics are only important for grid spacings finer than 10 km. The modeling study of Jeevanjee (2017) illustrated that in FV3, the non-hydrostatic and hydrostatic vertical velocities only begin to diverge at dx=2 km and finer (see their Figure 4). Jang and Hong (2016) highlight prior studies indicating that non-hydrostatic dynamics are 'weak' at grid spacing courser than 5 km. Similarly, a study of the IFS model using hydrostatic and non-hydrostatic versions found very little sensitivity down to 5 km (Wedi et al. 2010):

"The tests performed ranged from seasonal climate runs at T159 (125 km) resolution to medium-range forecasts up to and including T2047 (10 km) to assess the performance of the non-hydrostatic model in the hydrostatic regime, all the way to ultra-high resolution simulations in the non-hydrostatic regime (Wedi et al., 2009). Experiments with the T2047 horizontal resolution indicate that the differences between the hydrostatic and the non-hydrostatic simulations are still not significant at this resolution. Even the highest horizontal resolution at which the IFS can be run to date (T3999, 5 km) is still too coarse to fully resolve non-hydrostatic phenomena."

We do not want to insist, based on these studies, that our configuration is optimal. As the reviewer highlights, there are studies that find non-hydrostatic dynamics important at grid spacings significantly coarser than 10 km. So even though studies with FV3 and IFS suggest that nonhydrostatic effects are of secondary importance at grid-spacings of 5-10 km, we agree that caution is warranted. Nevertheless, we believe that the spread in the literature of the grid-spacing at which non-hydrostatic dynamics becomes important indicates there is insufficient basis to deem our model configuration inappropriate. We have made adjustments in Section 2.2 *HMA-VR Grid and Performance* to emphasize this point, and that we are not insisting on our configuration as appropriate. We've also added some language in Section 3.7 *Future Directions*, using this uncertainty to motivate a future study using the same grid resolution, but with the MPAS-A non-hydrostatic dycore that was recently ported to CESM2.3.X.

The adjusted paragraph in Section 2.2 *HMA-VR Grid and Performance* (L213-233) is shown below:

"The spectral-element dynamical core used by CESM is currently based on the hydrostatic approximation; non-hydrostatic vertical acceleration terms are neglected, which are important for the representation of deep convection, gravity waves, and flow over topography (Jeevanjee, 2017; Liu et al., 2022). Conventionally, the horizontal scales at which non-hydrostatic terms become important are assumed to be O(10 km), the vertical scale of the atmosphere (e.g., Wedi and Smolarkiewicz, 2009). This could raise the question whether it is appropriate or not to use a 7 km regionally refined grid in combination with a hydrostatic model. In literature, there is, however, a spread of the grid spacing at which non-hydrostatic dynamics becomes important. There are, for instance, studies that indicate non-hydrostatic dynamics are only important for grid spacings finer than 10 km. Jeevanjee (2017) illustrated that in the FV3 model, the non-hydrostatic and hydrostatic vertical velocities only begin to diverge at dx = 2km or finer. Jang and Hong (2016) highlighted prior studies (e.g., Dudhia, 1993, 2014; Kato, 1996; Janjic et al., 2001) indicating that non-hydrostatic dynamics are 'weak' at grid spacings coarser than 5 km. Similarly, a study of the IFS model using hydrostatic and non-hydrostatic versions found very little sensitivity down to 5 km (Wedi et al., 2010). Other studies indicate that non-hydrostatic dynamics are also important at grid spacings coarser than 10 km, which is suggested by the studies of Yang et al. (2017) and Liu et al. (2022), who show that non-hydrostatic terms are important at grid spacings up to 25 km, particularly in its representation of tropical convective systems. Due to the inherent difficulty in testing the null hypothesis (Liu et al., 2022), and the spread in literature of the grid-spacing at which non-hydrostatic dynamics becomes important, we believe there is insufficient basis to deem our model configuration inappropriate. However, we are aware that using the hydrostatic version in combination with a 7 km regionally refined grid could propagate model-related uncertainties into model outputs, such as precipitation. Nonetheless, the ability of the hydrostatic VR-CESM to simulate mountainous climate with a 7 km regionally refined grid has successfully been shown over the mountain ranges of western USA (Rhoades et al., 2018)."

The newly added paragraph in Section 3.7 *Future Directions* (L591-598) is shown below:

"In this study, the spectral-element dynamical core used by CESM is based on the hydrostatic approximation, which means non-hydrostatic vertical acceleration terms are neglected. Although we believe there is an insufficient basis to deem our model configuration inappropriate, we are aware that the hydrostatic version in combination with a 7 km regionally refined grid could propagate model-related uncertainties into model outputs, such as precipitation, wind, and cloud cover, which eventually also could affect the SEB and SMB. To address these model-related uncertainties, one potential solution could be to apply a similar regionally refined grid in combination with the newly developed MPAS (Model for Prediction Across Scales) non-hydrostatic dynamical core, which has recently successfully been applied over the western US mountain ranges (Huang et al., 2022)"

**References**

Dudhia, J.: A nonhydrostatic version of the Penn State-NCAR mesoscale model: validation tests and simulation of an Atlantic cyclone and cold front, Mon Weather Rev, 121, https://doi.org/10.1175/1520-0493(1993)121<1493:ANVOTP>2.0.CO;2, 1993.

Dudhia, J.: A history of mesoscale model development, Asia Pac J Atmos Sci, 50, https://doi.org/10.1007/s13143-014-0031-8, 2014.

Huang, X., Gettelman, A., Skamarock, W. C., Lauritzen, P. H., Curry, M., Herrington, A., Truesdale, J. T., and Duda, M.: Advancing precipitation prediction using a new-generation storm-resolving model framework - SIMA-MPAS (V1.0): a case study over the western United States, Geosci Model Dev, 15, https://doi.org/10.5194/gmd-15-8135-2022, 2022.

Jang, J. and Hong, S. Y.: Comparison of simulated precipitation over East Asia in two regional models with hydrostatic and nonhydrostatic dynamical cores, Mon Weather Rev, 144, https://doi.org/10.1175/MWR-D-15-0428.1, 2016.

Janjic, Z. I., Gerrity, J., and Nickovic, S.: An alternative approach to nonhydrostatic modeling, Mon Weather Rev, 129, https://doi.org/10.1175/1520-0493(2001)129<1164:AAATNM>2.0.CO;2, 2001.

Jeevanjee, N.: Vertical Velocity in the Gray Zone, J Adv Model Earth Syst, 9, https://doi.org/10.1002/2017MS001059, 2017.

Kato, T.: Hydrostatic and non-hydrostatic simulations of the 6 August 1993 Kagoshima torrential rain, Journal of the Meteorological Society of Japan, 74, https://doi.org/10.2151/jmsj1965.74.3_355, 1996.

Liu, W., Ullrich, P. A., Guba, O., Caldwell, P. M., and Keen, N. D.: An Assessment of Nonhydrostatic and Hydrostatic Dynamical Cores at Seasonal Time Scales in the Energy Exascale Earth System Model (E3SM), J Adv Model Earth Syst, 14, https://doi.org/10.1029/2021MS002805, 2022.

Rhoades, A. M., Ullrich, P. A., Zarzycki, C. M., Johansen, H., Margulis, S. A., Morrison, H., Xu, Z., and Collins, W. D.: Sensitivity of Mountain Hydroclimate Simulations in Variable-Resolution CESM to Microphysics and Horizontal Resolution, J Adv Model Earth Syst, 10, 1357–1380, https://doi.org/10.1029/2018MS001326, 2018.

Wedi, N., Benard, P., Yessad, K., Untch, A., Malardel, S., Hamrud, M., Mozdzynski, G., Fisher, M., and Smolarkiewicz, P.: Non-hydrostatic modeling with IFS: current status, https://www.ecmwf.int/en/elibrary/78642-non-hydrostatic-modeling-ifs-current-status, November 2010.

Wedi, N. P. and Smolarkiewicz, P. K.: A framework for testing global non-hydrostatic models, Quarterly Journal of the Royal Meteorological Society, 135, https://doi.org/10.1002/qj.377, 2009.

Yang, Q., Leung, L. R., Lu, J., Lin, Y. L., Hagos, S., Sakaguchi, K., and Gao, Y.: Exploring the effects of a nonhydrostatic dynamical core in high-resolution aquaplanet simulations, J Geophys Res, 122, https://doi.org/10.1002/2016JD025287, 2017.